# Neural Collapse Under MSE Loss: Proximity to and Dynamics on the Central Path

**X.Y. Han**[*]
Cornell University
xh332@cornell.edu

**Vardan Papyan**[*]
University of Toronto
vardan.papyan@utoronto.ca

**David L. Donoho**
Stanford University
donoho@stanford.edu

## Abstract

The recently discovered Neural Collapse (NC) phenomenon occurs pervasively in today's deep net training paradigm of driving cross-entropy (CE) loss towards zero. During NC, last-layer features collapse to their class-means, both classifiers and class-means collapse to the same Simplex Equiangular Tight Frame, and classifier behavior collapses to the nearest-class-mean decision rule. Recent works demonstrated that deep nets trained with mean squared error (MSE) loss perform comparably to those trained with CE. As a preliminary, we empirically establish that NC emerges in such MSE-trained deep nets as well through experiments on three canonical networks and five benchmark datasets. We provide, in a Google Colab notebook, PyTorch code for reproducing MSE-NC and CE-NC: here. The analytically-tractable MSE loss offers more mathematical opportunities than the hard-to-analyze CE loss, inspiring us to leverage MSE loss towards the theoretical investigation of NC. We develop three main contributions: (I) We show a new decomposition of the MSE loss into (A) terms directly interpretable through the lens of NC and which assume the last-layer classifier is exactly the least-squares classifier; and (B) a term capturing the deviation from this least-squares classifier. (II) We exhibit experiments on canonical datasets and networks demonstrating that term-(B) is negligible during training. This motivates us to introduce a new theoretical construct: the *central path*, where the linear classifier stays MSE-optimal for feature activations throughout the dynamics. (III) By studying renormalized gradient flow along the central path, we derive exact dynamics that predict NC.

## 1 Introduction

Modern deep learning includes paradigmatic procedures that are commonly adopted, but not entirely understood. Some examples are multi-layered architectures, stochastic gradient descent, batch normalization, cross-entropy (CE) loss, and training past *zero error* towards *zero loss*. Analyzing the properties of these practices is an important research task.

In this work, we theoretically investigate behavior of last-layer features in classification deep nets. In particular, we consider the training of deep neural networks on datasets containing images from $C$ different classes with $N$ examples in each class. After passing the $i$-th example in the $c$-th class through all layers except the last-layer of the network, the network outputs some last-layer features $\boldsymbol{h}_{i,c} \in \mathbb{R}^P$. The last-layer of the network—which, for each class $c$, possesses a classifier $\boldsymbol{w}_c \in \mathbb{R}^P$ and bias $\boldsymbol{b}_c \in \mathbb{R}$—then predicts a label for the example using the rule $\arg\max_{c'} (\langle \boldsymbol{w}_{c'}, \boldsymbol{h}_{i,c} \rangle + b_{c'})$.

The network's performance is evaluated by calculating the *error* defined by

$$\text{Error} = \underset{i,c}{\text{Ave}}\ \mathbf{1}\{c \neq \arg\max_{c'} (\langle \boldsymbol{w}_{c'}, \boldsymbol{h}_{i,c} \rangle + b_{c'})\},$$

while the weights, biases, and other parameters of the network (that determine the behavior of the layers before the last layer) in the network are updated by minimizing the CE loss defined by

$$\text{CE} = -\underset{i,c}{\text{Ave}} \log \frac{\exp\{\langle \boldsymbol{w}_c, \boldsymbol{h}_{i,c} \rangle + b_c\}}{\sum_{c'=1}^{C} \exp\{\langle \boldsymbol{w}_{c'}, \boldsymbol{h}_{i,c} \rangle + b_{c'}\}},$$

---

[*]Equal contribution. Listed alphabetically.

where $\mathrm{Ave}$ is the operator that averages over its subscript indices.

Prior works such as Zhang et al. (2016); Belkin et al. (2019) have shown that overparameterized classifiers (such as deep nets) can "memorize" their training set without harming performance on unseen test data. Moreover, works such as Soudry et al. (2018) have further shown that continuing to train networks past memorization can still lead to performance improvements[1,2]. Papyan, Han, and Donoho (2020) recently examined this setting, referring to the phase during which one trains past zero-error towards zero-CE-loss as the **Terminal Phase of Training** (TPT). During TPT, they exposed a phenomenon called **Neural Collapse** (NC).

## 1.1 NEURAL COLLAPSE

NC is defined relative to the feature global mean,

$$\boldsymbol{\mu}_G = \mathrm{Ave}_{i,c}\, \boldsymbol{h}_{i,c},$$

the feature class-means,

$$\boldsymbol{\mu}_c = \mathrm{Ave}_{i}\, \boldsymbol{h}_{i,c}, \quad c = 1, \ldots, C,$$

the feature within-class covariance,

$$\boldsymbol{\Sigma}_W = \mathrm{Ave}_{i,c}(\boldsymbol{h}_{i,c} - \boldsymbol{\mu}_c)(\boldsymbol{h}_{i,c} - \boldsymbol{\mu}_c)^\top, \quad (1)$$

and the feature between-class covariance,

$$\boldsymbol{\Sigma}_B = \mathrm{Ave}_{c}(\boldsymbol{\mu}_c - \boldsymbol{\mu}_G)(\boldsymbol{\mu}_c - \boldsymbol{\mu}_G)^\top.$$

It is characterized by the following four limiting behaviors where limits take place with increasing training epoch $t$:

**(NC1) Within-class variability collapse[3]:**

$$\boldsymbol{\Sigma}_B^\dagger \boldsymbol{\Sigma}_W \to \boldsymbol{0},$$

where $\dagger$ denotes the Moore-Penrose pseudoinverse.

**(NC2) Convergence to Simplex ETF:**

$$\frac{\langle \boldsymbol{\mu}_c - \boldsymbol{\mu}_G, \boldsymbol{\mu}_{c'} - \boldsymbol{\mu}_G \rangle}{\|\boldsymbol{\mu}_c - \boldsymbol{\mu}_G\|_2 \|\boldsymbol{\mu}_{c'} - \boldsymbol{\mu}_G\|_2} \to \begin{cases} 1, & c = c' \\ \frac{-1}{C-1}, & c \neq c' \end{cases}$$

$$\|\boldsymbol{\mu}_c - \boldsymbol{\mu}_G\|_2 - \|\boldsymbol{\mu}_{c'} - \boldsymbol{\mu}_G\|_2 \to 0 \quad \forall c \neq c'$$

**(NC3) Convergence to self-duality:**

$$\frac{\boldsymbol{w}_c}{\|\boldsymbol{w}_c\|_2} - \frac{\boldsymbol{\mu}_c - \boldsymbol{\mu}_G}{\|\boldsymbol{\mu}_c - \boldsymbol{\mu}_G\|_2} \to 0$$

**(NC4): Simplification to nearest class center:**

$$\arg\max_{c'} \langle \boldsymbol{w}_{c'}, \boldsymbol{h} \rangle + b_{c'} \to \arg\min_{c'} \|\boldsymbol{h} - \boldsymbol{\mu}_{c'}\|_2$$

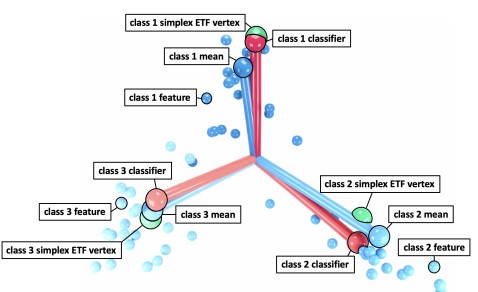

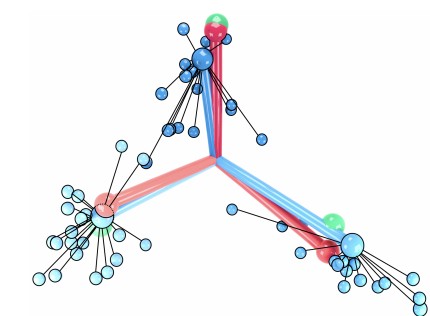

Figure 1: *Portrait of Neural Collapse.* Top figure depicts the last-layer features, class-means, and classifiers with which NC is defined—as well as the Simplex ETF to which they all converge with training. Bottom figure shows the deviations of features from their corresponding class-means. *Reproduced and modified from Figure 1 of Papyan, Han, and Donoho (2020).*

---

[1] A similar phenomenon had been previously observed in boosting (Bartlett et al., 1998).

[2] An alternative line of work on "early stopping" (Prechelt, 1998; Li et al., 2020; Rice et al., 2020) advocates for terminating the training process early and shows that it could be beneficial when training on noisy data or small datasets. Such datasets are out of the scope of this paper.

[3] Our characterization of **(NC1)** is more precise than that given by Papyan, Han, and Donoho (2020), which only states $\boldsymbol{\Sigma}_W \to \boldsymbol{0}$ for rhetorical simplicity. The convergence of the trace-norm of $\boldsymbol{\Sigma}_B^\dagger \boldsymbol{\Sigma}_W$ to zero is the actual quantity measured by Figure 6 of Papyan, Han, and Donoho (2020) when demonstrating **(NC1)**. $\boldsymbol{\Sigma}_B^\dagger \boldsymbol{\Sigma}_W$

The **(NC2)** property captures convergence to a simple geometric structure called an *Equiangular Tight Frame* (ETF). An ETF is a collection of vectors $\{\boldsymbol{v}_c\}_{c=1}^{C}$ having equal lengths and equal, maximally separated pair-wise angles. In classification deep nets, last-layer features are of higher dimension than the number of classes i.e. $P>C$. In this setting, the maximal angles are given by

$$\frac{\langle \boldsymbol{v}_c, \boldsymbol{v}_{c'} \rangle}{\|\boldsymbol{v}_c\|_2 \|\boldsymbol{v}_{c'}\|_2} = \left\{ \begin{array}{ll} 1, & \text{for } c = c' \\ -\frac{1}{C-1}, & \text{for } c \neq c' \end{array} \right. ,$$

and the ETF is called a Simplex ETF[4]. Observe that as $C$ increases, the ETF approaches a (partial) orthogonal matrix. Thus, when $C$ is large, this translates to the intuitive notion that classifiers and class-means tend with training to (near) orthogonality.

## 1.2 DEEP NET CLASSIFICATION WITH MSE LOSS

While classification deep nets are typically trained with CE loss, Demirkaya et al. (2020) and Hui & Belkin (2020) recently reported deep nets trained with squared error (MSE) loss,

$$\mathcal{L}(\boldsymbol{W}, \boldsymbol{b}, \boldsymbol{H}) = \frac{1}{2} \operatorname*{Ave}_{i,c} \|\boldsymbol{W}\boldsymbol{h}_{i,c} + \boldsymbol{b} - \boldsymbol{y}_{i,c}\|_2^2 + \frac{\lambda}{2}(\|\boldsymbol{W}\|_F^2 + \|\boldsymbol{b}\|_2^2) \tag{2}$$

$$= \frac{1}{2CN}\|\boldsymbol{W}\boldsymbol{H} + \boldsymbol{b}\mathbb{1}_{CN}^\top - \boldsymbol{Y}\|_F^2 + \frac{\lambda}{2}(\|\boldsymbol{W}\|_F^2 + \|\boldsymbol{b}\|_2^2),$$

achieve comparable test performance as those trained with CE loss. Above, $\boldsymbol{H} \in \mathbb{R}^{P \times CN}$ and $\boldsymbol{Y} \in \mathbb{R}^{C \times CN}$ are matrices resulting from stacking[5] together the feature vectors $\boldsymbol{h}_{i,c}$ and one-hot vectors $\boldsymbol{y}_{i,c}$ as their respective columns; $\boldsymbol{W} \in \mathbb{R}^{C \times P}$ is the matrix resulting from the stacking of classifiers $\boldsymbol{w}_c$ as rows; $\boldsymbol{b} \in \mathbb{R}^C$ is the vector resulting from concatenating the scalars $\{b_c\}_{c=1}^C$; and $\mathbb{1}_{CN}$ is the length-$CN$ vector of ones. Table 1 in Appendix A shows supplementary measurements affirming the findings of Hui & Belkin (2020); Demirkaya et al. (2020), i.e. the measurements affirm that MSE-trained networks indeed achieve accuracies on testing data comparable to those of CE-trained networks (cf. the analogous Table 1 in Papyan, Han, and Donoho (2020)). The analytically-tractable MSE loss offers more mathematical opportunities than the hard-to-analyze CE loss and inspires this paper's main theoretical contributions explicitly characterizing MSE-NC.

## 1.3 CONTRIBUTIONS

Our main contributions are as follows:

- We propose a new *decomposition of the MSE loss*: $\mathcal{L} = \mathcal{L}_{\text{NC1}} + \mathcal{L}_{\text{NC2/3}} + \mathcal{L}_{\text{LS}}^{\perp}$. The terms $\mathcal{L}_{\text{NC1}}$ and $\mathcal{L}_{\text{NC2/3}}$ possess interpretations attributable to NC phenomena and assume the classifier $\boldsymbol{W}$ is exactly the least-squares optimal classifier $\boldsymbol{W}_{\text{LS}}$ (relative to the given $\boldsymbol{H}$); and $\mathcal{L}_{\text{LS}}^{\perp}$ captures the deviation of $\boldsymbol{W}$ from $\boldsymbol{W}_{\text{LS}}$. (Section 2)
- We provide empirical measurements of our decomposition (Figure 2) on realistic dataset-network combinations showing that $\mathcal{L}_{\text{LS}}^{\perp}$ becomes negligible during training—leading us to define the *central path* where $\mathcal{L}_{\text{LS}}^{\perp} = zero$. (Section 2)
- We reveal a key *invariance property* on the central path. The invariance motivates the examination of a representative set of features, $\boldsymbol{X} = \boldsymbol{\Sigma}_W^{-\frac{1}{2}} \boldsymbol{H}$, that we call *renormalized features*. (Sections 3.1-3.2)
- We study the gradient flow of those renormalized features along the central path and derive *exact, closed-form dynamics* that imply NC. The dynamics are explicit in terms of the

---

captures the subtlety that the size of the within-class "noise" (captured by $\boldsymbol{\Sigma}_W$) should be viewed relative to the size of the class-means (captured by $\boldsymbol{\Sigma}_B$).

[4]Traditional research on ETFs (cf. Strohmer & Heath Jr (2003); Fickus & Mixon (2015)) examines the $P \leq C$ setting where $C$ ETF-vectors span their ambient $\mathbb{R}^P$ space. However, in the $P > C$ setting of classification deep nets, the vectors can not span $\mathbb{R}^P$. Following the precedent of Papyan, Han, and Donoho (2020), we interpret ETFs as equal-length and maximally-equiangular vectors that are not necessarily spanning.

[5]Assume the stacking is performed in *i-then-c order*: the first column is $(i, c) = (1, 1)$, the second is $(i, c) = (2, 1),...,$ the $N$-th is $(i, c) = (N, 1)$, the $(N + 1)$-st is $(i, c) = (1, 2)$, and so on. This matters for formalizing Equation 11 in Corollary 2.

*singular value decomposition* of the *renormalized feature class-means* at initialization. (Section 3.3)

Additionally, we complement this paper with new, extensive measurements on five benchmark datasets—in particular, the MNIST, FashionMNIST, CIFAR10, SVHN, and STL10 datasets[6] (Deng et al., 2009; Krizhevsky & Hinton, 2009; LeCun et al., 2010; Xiao et al., 2017)—and three canonical deep nets—in particular, the VGG, ResNet, and DenseNet networks (He et al., 2016; Huang et al., 2017; Simonyan & Zisserman, 2014)—that verify the empirical reality of MSE-NC i.e. they show **(NC1)-(NC4)** indeed occur for networks trained with MSE loss. These experiments establish that theoretical modeling of MSE-NC is empirically well-motivated. They are lengthy—together spanning four pages with seven figures and a table—so we collectively defer them to Appendix A.

## 2 DECOMPOSITION OF MSE LOSS

Inspired by the community's interest in the role of the MSE loss in deep net training (Demirkaya et al., 2020; Hui & Belkin, 2020; Mixon et al., 2020; Poggio & Liao, 2020a;b), we derive a new decomposition of the MSE loss that gives insights into the NC phenomenon. First, absorb the bias vector into the weight matrix—by defining the extended weight matrix $\widetilde{W} = [W, b] \in \mathbb{R}^{C \times (P+1)}$ and the extended feature vector $\widetilde{h}_{i,c} = [h_{i,c}; 1] \in \mathbb{R}^{P+1}$—so that Equation 2 can be rewritten as

$$\mathcal{L}(\widetilde{W}, \widetilde{H}) = \frac{1}{2} \underset{i,c}{\mathrm{Ave}} \|\widetilde{W}\widetilde{h}_{i,c} - y_{i,c}\|_2^2 + \frac{\lambda}{2} \|\widetilde{W}\|_F^2. \tag{3}$$

Using $\left\{\widetilde{h}_{i,c}\right\}$, define the entities $\widetilde{\mu}_c$, $\widetilde{\mu}_G$, $\widetilde{H}$, and $\widetilde{\Sigma}_W$ analogously to those in Section 1.1. We further define the extended total covariance and extended class-means matrices, respectively, as

$$\widetilde{\Sigma}_T = \underset{i,c}{\mathrm{Ave}}(\widetilde{h}_{i,c} - \widetilde{\mu}_G)(\widetilde{h}_{i,c} - \widetilde{\mu}_G)^\top \in \mathbb{R}^{(P+1)\times(P+1)}$$

$$\widetilde{M} = [\widetilde{\mu}_1, \dots, \widetilde{\mu}_C] \in \mathbb{R}^{(P+1)\times C}.$$

Next, we reformulate, with weight decay incorporated, a classic result of Webb & Lowe (1990):

**Proposition 1** (**Webb & Lowe (1990) with Weight Decay**). *For fixed extended features $\widetilde{H}$, the optimal classifier minimizing the MSE loss $\mathcal{L}(\widetilde{W}, \widetilde{H})$ is*

$$\widetilde{W}_{LS} = \frac{1}{C}\widetilde{M}^\top(\widetilde{\Sigma}_T + \widetilde{\mu}_G\widetilde{\mu}_G^\top + \lambda I)^{-1},$$

*where $I$ is the identity matrix. Note that $\widetilde{W}_{LS}$ depends on $\widetilde{H}$ only.*

Note that we can interpret $\widetilde{W}_{\mathrm{LS}}$ as a function of $\widetilde{H}$, leading us to identify the following decomposition of $\mathcal{L}$ where one term depends on $\widetilde{H}$ only.

**Theorem 1.** *(Decomposition of MSE Loss; Proof in Appendix B) The MSE loss, $\mathcal{L}(\widetilde{W}, \widetilde{H})$, can be decomposed into two terms, $\mathcal{L}(\widetilde{W}, \widetilde{H}) = \mathcal{L}_{LS}(\widetilde{H}) + \mathcal{L}_{LS}^{\perp}(\widetilde{W}, \widetilde{H})$, where*

$$\mathcal{L}_{LS}(\widetilde{H}) = \frac{1}{2}\underset{i,c}{\mathrm{Ave}}\|\widetilde{W}_{LS}\widetilde{h}_{i,c} - y_{i,c}\|_2^2 + \frac{\lambda}{2}\|\widetilde{W}_{LS}\|_F^2,$$

*and*

$$\mathcal{L}_{LS}^{\perp}(\widetilde{W}, \widetilde{H}) = \frac{1}{2}\mathrm{tr}\Big\{(\widetilde{W} - \widetilde{W}_{LS})\left(\widetilde{\Sigma}_T + \widetilde{\mu}_G\widetilde{\mu}_G^\top + \lambda I\right)(\widetilde{W} - \widetilde{W}_{LS})^\top\Big\}.$$

In the above, $\mathcal{L}_{\mathrm{LS}}(\widetilde{H})$ is independent of $\widetilde{W}$. Intuitively, it is the MSE-performance of the optimal classifiers $\widetilde{W}_{\mathrm{LS}}$ (rather than the "real classifiers" $\widetilde{W}$) on input $\widetilde{H}$.

---

[6]CIFAR100 and ImageNet are omitted because Demirkaya et al. (2020); Hui & Belkin (2020) showed that they require an additional scaling-heuristic on top of the traditional MSE loss. Rigorous investigation into this scaling-heuristic would have demanded mathematical analysis outside the scope of this paper as well as experimentation that would have consumed expensive amounts of computational resources.

The component $\mathcal{L}_{\text{LS}}^{\perp}(\widetilde{W}, \widetilde{H})$ is non-negative[7] and is zero only when $\widetilde{W} = \widetilde{W}_{\text{LS}}$. Therefore, $\mathcal{L}_{\text{LS}}^{\perp}(\widetilde{H})$ quantifies the distance of $\widetilde{W}$ from $\widetilde{W}_{\text{LS}}$. In short, the *least-squares* component, $\mathcal{L}_{\text{LS}}(\widetilde{H})$, captures the behavior of the network when the classifier possesses optimal, least squares behavior. Meanwhile, the *deviation* component, $\mathcal{L}_{\text{LS}}^{\perp}(\widetilde{W}, \widetilde{H})$, captures the divergence from that behavior.

We can further decompose $\mathcal{L}_{\text{LS}}(\widetilde{H})$ into two terms, one capturing activation collapse **(NC1)** and the other capturing convergence to Simplex ETF of both features and classifiers (**(NC2)** and **(NC3)**).

**Theorem 2.** *(Decomposition of Least-Squares Component; Proof in Appendix C) The least-squares component, $\mathcal{L}_{LS}(\widetilde{H})$, of the MSE decomposition in Theorem 1 can be further decomposed into $\mathcal{L}_{LS}(\widetilde{H}) = \mathcal{L}_{NC1}(\widetilde{H}) + \mathcal{L}_{NC2/3}(\widetilde{H})$, where*

$$\mathcal{L}_{NC1}(\widetilde{H}) = \frac{1}{2}\operatorname{tr}\left\{\widetilde{W}_{LS}\left[\widetilde{\Sigma}_W + \lambda I\right]\widetilde{W}_{LS}^{\top}\right\},$$

$$\mathcal{L}_{NC2/3}(\widetilde{H}) = \frac{1}{2C}\|\widetilde{W}_{LS}\widetilde{M} - I\|_F^2.$$

Inspection of these terms is revealing. First, observe that $\mathcal{L}_{\text{NC2/3}}$ is a function of the class-means and MSE-optimal classifiers. Minimizing $\mathcal{L}_{\text{NC2/3}}$ will push the (unextended) class-means and classifiers towards the same Simpex ETF matrix[8] i.e. **(NC2)-(NC3)**. Next, note that the within-class variation is independent of the means. Thus, despite the fact that classifiers are converging towards some (potentially large) ETF matrix, we can always reduce $\mathcal{L}_{\text{NC1}}$ by pushing $\Sigma_W$ towards zero, which corresponds to **(NC1)** .

Figure 2 measures the empirical values of the above decomposition terms on five canonical datasets and three prototypical networks. It shows that $\mathcal{L}_{\text{LS}}^{\perp}(\widetilde{W}, \widetilde{H})$ becomes negligible compared to $\mathcal{L}_{\text{LS}}(\widetilde{H})$ when training canonical deep nets on benchmark datasets. In other words, $\mathcal{L}(\widetilde{W}, \widetilde{H}) \approx \mathcal{L}_{\text{LS}}(\widetilde{H})$ starting from an early epoch in training and persists into TPT.

This motivates us to formulate the *central path*:

$$\mathcal{P} = \left\{(\widetilde{W}_{\text{LS}}(\widetilde{H}), \widetilde{H}) \mid \widetilde{H} \in \mathbb{R}^{(P+1) \times CN}\right\}, \tag{4}$$

where the notation $\widetilde{W}_{\text{LS}}(\cdot)$ makes explicit the fact that $\widetilde{W}_{\text{LS}}$ is a function of $\widetilde{H}$ only. Intuitively, for a classifier-features pair to lie on the central path, i.e. $(\widetilde{W}, \widetilde{H}) \in \mathcal{P}$, means that the "real classifier" $\widetilde{W}$ will exactly equal $\widetilde{W}_{\text{LS}}(\widetilde{H})$—the optimal classifier that would result from fixing $\widetilde{H}$ and minimizing $\mathcal{L}$ w.r.t. just the classifier. Combined with Theorem 1, we see $(\widetilde{W}, \widetilde{H}) \in \mathcal{P}$ if and only if $\mathcal{L}(\widetilde{W}, \widetilde{H}) = \mathcal{L}_{\text{LS}}(\widetilde{H})$. Figure 2 shows that classifier-features pairs lie approximately on the central path during TPT, allowing us to shift focus from $\mathcal{L}$ to $\mathcal{L}_{\text{LS}}$.

## 3 EXACT CLOSED-FORM ANALYSIS ON CENTRAL PATH

Consider $\lambda = 0$ in this section. For the subsequent theory, we adopt the *unconstrained features* (Mixon et al., 2020) or *layer-peeled* (Fang et al., 2021) modeling perspective[9]. We consider continuous training dynamics with gradient flow where *time-of-training* is denoted by the variable[10] $t \geq 0$. Within this model, we analyze the dynamics of the *renormalized features* $X = \Sigma_W^{-\frac{1}{2}} H$ on the central path

$$\frac{\mathrm{d}}{\mathrm{d}t}X = -\Pi_{T_X \mathcal{X}}\left(\nabla_X \mathcal{L}_{\text{LS}}(X)\right), \tag{5}$$

---

[7]The middle term is the sum of positive-semidefinite matrices—one of which is positive-definite.

[8]Appendices C.1-C.2 elaborates upon this interpretation of $\mathcal{L}_{\text{NC2/3}}$ in more detail.

[9]Both the unconstrained features model (Mixon et al., 2020) and the (1-)layer-peeled model (Fang et al., 2021) advocate for the mathematical analysis of deep nets through variants of gradient flow directly on the last-layer features and classifiers. The layer-peeled model is so-named because it proposes to analyze deep nets by "peeling away" one layer at a time. The unconstrained features model is so-name because features are not constrained to be the output of a deep net forward pass but are rather free variables that can be optimized directly: This captures the nearly unlimited flexibility afforded to feature engineering transformations in modern deep nets by the many nonlinear, overparameterized layers.

[10]Intuitively, one should interpret $t=0$ as the time-of-training when feature-classifier pairs effectively enter the central path.

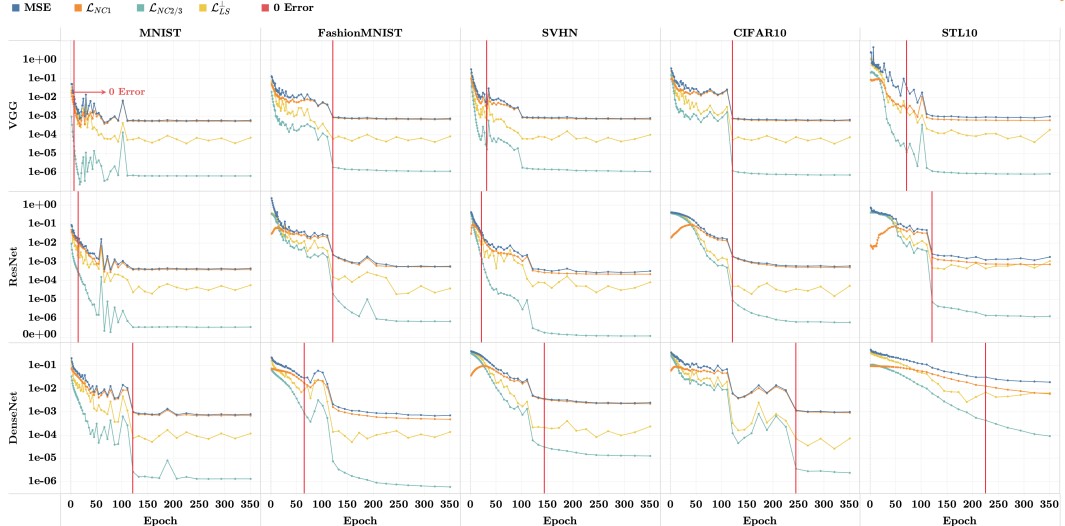

Figure 2: *Decomposition of MSE loss:* Each array column shows a benchmark image classification dataset while each row shows a canonical deep net architecture trained with MSE loss. The red vertical line indicates the epoch at which zero training error was achieved. In each array cell, we plot terms of the MSE loss decomposition $\mathcal{L}(\widetilde{W}, \widetilde{H}) = \mathcal{L}_{\mathrm{NC1}}(\widetilde{H}) + \mathcal{L}_{\mathrm{NC2/3}}(\widetilde{H}) + \mathcal{L}_{\mathrm{LS}}^{\perp}(\widetilde{W}, \widetilde{H})$ from Section 2. Starting from an early epoch in training, $\mathcal{L}_{\mathrm{LS}}^{\perp}(\widetilde{W}, \widetilde{H})$ becomes negligible compared to the dominant term, $\mathcal{L}_{\mathrm{NC1}}(\widetilde{H})$, implying $\mathcal{L}_{\mathrm{LS}}^{\perp}(\widetilde{W}, \widetilde{H}) \ll \mathcal{L}_{\mathrm{LS}}(\widetilde{H}) = \mathcal{L}_{\mathrm{NC1}}(\widetilde{H}) + \mathcal{L}_{\mathrm{NC2/3}}(\widetilde{H})$, i.e. the features and classifiers are *effectively on the central path* during TPT. Note that $\mathcal{L}_{\mathrm{NC2/3}}(\widetilde{H})$ diminishes the fastest among all the terms: Intuitively, this shows that the network primarily focuses on distributing the feature class-means into a "uniform" Simplex ETF configuration **(NC1)-(NC2)** early on and, from there, compresses the activations towards their class-means, i.e. **(NC1)**, as much as possible. Further experimental details are in Appendix A. Outlier behavior is discussed in Appendix A.7.

where the operator $\Pi_{T_{x}\mathcal{X}}$ projects the gradient onto the tangent space of the manifold[11], $\mathcal{X}$, of all identity-covariance features. For $\Sigma_W^{-\frac{1}{2}} H$ to be well-defined, we assume that $\Sigma_W$ *remains full-rank*[12] during training. We call Equation 5 the *continually renormalized gradient flow* and will motivate it in more detail in later subsections. We now state our main theorem:

**Theorem 3.** *(NC Under Continually Renormalized Gradient Flow) Consider the continually renormalized gradient flow (Equation 5) in which the dynamics are restricted to the central path, and the features are renormalized to identity within-class covariance. Assume the features are initialized with zero global mean. Then, Neural Collapse emerges on the renormalized features with explicit dynamics of the flow given in Proposition 2 as well as Corollaries 1 and 2 later in this paper.*

Three assumptions standout which we discuss below:

**(A1) Restriction to central path:** Figure 2 shows that—in practice, during TPT—$\mathcal{L}_{\mathrm{LS}}^{\perp}(\widetilde{W}, \widetilde{H})$ is observed to be negligible compared to the dominant term, $\mathcal{L}_{\mathrm{LS}}(\widetilde{H})$. This indicates $(\widetilde{W}, \widetilde{H})$ is close to the central path (Equation 4) where $\mathcal{L}(\widetilde{W}, \widetilde{H}) = \mathcal{L}_{\mathrm{LS}}(\widetilde{H})$. One the central path, one is effectively training with $\mathcal{L}_{\mathrm{LS}}$ as the loss.

**(A2) Zero global mean:** Proximity to the central path implies that the last-layer biases become close to $b_{\mathrm{LS}}$. Having biases $b = b_{\mathrm{LS}}$ is effectively performing classification with no bias term (i.e. using only a linear classifier $W$) on *zero-mean data*. To see this, define the globally-centered features,

---

[11] $\mathcal{X}$ is a variant of the well-known *generalized Stiefel manifold* (Absil et al., 2009, Chapter 3.4).

[12] Given the definition in Equation 1, the positive-definiteness of $\Sigma_W$ is unsurprising in deep net contexts because $CN \gg P$ i.e. the number of examples is larger than the width of the last-layer. Note this does not contradict **(NC1)**. **(NC1)** is characterized by $\Sigma_W$ *approaching* the zero-matrix with training (relative to $\Sigma_B$). In practice (for example, in our supplementary Appendix A experiments), we observed that the zero-matrix is never actually achieved during the training dynamics and $\Sigma_W$ indeed remains full-rank.

globally-centered means, and total-covariance matrix in unextended coordinates:

$$\overline{H} = H - \mu_G \mathbb{1}_{CN}^\top; \quad \overline{M} = [\mu_1 - \mu_G, \dots, \mu_C - \mu_G]; \quad \Sigma_T = \frac{1}{CN} \overline{H} \overline{H}^\top.$$

Proposition 1 then reduces to the following forms for the unextended weights and biases:

$$\widetilde{W}_{\mathrm{LS}} = [W_{\mathrm{LS}}, \; b_{\mathrm{LS}}] = \left[ C^{-1} \overline{M}^\top \Sigma_T^{-1}, \quad C^{-1} \mathbb{1}_C - C^{-1} \overline{M}^\top \Sigma_T^{-1} \mu_G \right]. \tag{6}$$

For an arbitrary activation-target pair $(h, y)$, the prediction error obeys

$$W_{\mathrm{LS}} h + b_{\mathrm{LS}} - y = C^{-1} \overline{M}^\top \Sigma_T^{-1} h + C^{-1} \mathbb{1}_C - C^{-1} \overline{M}^\top \Sigma_T^{-1} \mu_G - y$$
$$= W_{\mathrm{LS}}(h - \mu_G) - (y - C^{-1} \mathbb{1}_C).$$

The last line exhibits two terms in parentheses, both traceable to the action of the bias $b_{\mathrm{LS}}$. The term $h - \mu_G$ demonstrates that the bias induces a global-mean subtraction on $h$, while the term $y - C^{-1} \mathbb{1}$ demonstrates that the bias also induces a global-mean subtraction on the one-hot targets.

**(A3) Renormalization to identity covariance:** Renormalization is ubiquitous in the empirical deep learning literature through works such batch normalization and its variants (Ioffe & Szegedy, 2015; Salimans & Kingma, 2016; Wu & He, 2018; Ulyanov et al., 2016; Ba et al., 2016; Krizhevsky et al., 2012). It is also ideologically precedented in the theoretical ML literature through works such as Douglas et al. (1998); Banburski et al. (2019); Poggio & Liao (2020a;b). Within this paper, the covariance-renormalization is inspired by an invariance property of the loss as well as the intuitive concept of the signal-to-noise ratio from mathematical statistics (discussed later in Sections 3.1-3.2).

In the subsequent sections, we motivate and introduce the theoretical constructs necessary for proving Theorem 3 and for presenting the explicit dynamics of the flow.

## 3.1 INVARIANCE PROPERTY

By Assumption **(A2)**, classification on the central path is equivalent to classification on globally-centered features using only the linear classifier $W_{\mathrm{LS}}$. Then, using Equation 6, we can equivalently represent the central path (Equation 4) as follows:

**Definition 1 (Zero Global Mean Central Path).**

$$\overline{\mathcal{P}} = \left\{ (W_{LS}, \overline{H}) \; \middle| \; W_{LS} = C^{-1} \overline{M}^\top \Sigma_T^{-1} \right\}.$$

An *invariance property* holds on the central path. Let $A$ denote a symmetric, full-rank matrix. Then, the explicit form in Equation 6 for $W_{\mathrm{LS}}$ implies

$$W_{\mathrm{LS}}(A\overline{H}) A\overline{H} = C^{-1} \left( A\overline{M} \right)^\top \left[ \left( A\overline{H} \right) \left( A\overline{H} \right)^\top \right]^{-1} A\overline{H}$$
$$= C^{-1} \overline{M}^\top A \left[ A^{-1} \left( \overline{H} \overline{H}^\top \right)^{-1} A^{-1} \right] A\overline{H} \tag{7}$$
$$= C^{-1} \overline{M}^\top \left( \overline{H} \overline{H}^\top \right)^{-1} \overline{H} = C^{-1} \overline{M}^\top \Sigma_T^{-1} \overline{H} = W_{\mathrm{LS}} \left( \overline{H} \right) \overline{H},$$

where the notation $W_{\mathrm{LS}}(\cdot)$ makes explicit that $W_{\mathrm{LS}}$ is a function of a set of input features. Thus, the actual predictions made by the least squares classifier are invariant to choice of the coordinates in which we express $\overline{H}$, i.e. all features $A\overline{H}$ are *equivalently performing*. Among those coordinate systems, we will prefer the one in which the "noise" is "whitened" or "sphered." Recall that we assume $\Sigma_W(\overline{H})$—where the notation $\Sigma_W(\cdot)$ expresses the within-class covariance as a function of the features—is positive-definite. Consider the coordinate transformation $\overline{H} \mapsto A\overline{H}$ with $A = \Sigma_W^{-\frac{1}{2}}(\overline{H})$. In these coordinates, the features are "renormalized" to spherical covariance,

$$\Sigma_W(A\overline{H}) = A\Sigma_W(\overline{H})A = I,$$

so we will call $A\overline{H}$ the *renormalized features*. The class-means of $A\overline{H}$ are

$$\overline{M}(A\overline{H}) = A\overline{M}(\overline{H}) = \Sigma_W^{-\frac{1}{2}}(\overline{H})\overline{M}(\overline{H}),$$

where the notation $\overline{M}(\cdot)$ expresses that the means are a function of features. Thus, Equation 5 describes continuous training dynamics where we continually "renormalize" features to their equivalently-performing representer.

## 3.2 SIGNAL-TO-NOISE RATIO MATRIX ON THE CENTRAL PATH

The $\boldsymbol{\Sigma}_W^{-\frac{1}{2}}\overline{\boldsymbol{M}}$ term evokes the canonical idea of a signal-to-noise ratio from statistics. To see this, note that the classifier in Equation 6 can be rewritten as

$$\boldsymbol{W}_{\text{LS}} = C^{-1}\overline{\boldsymbol{M}}^{\top}\left(C^{-1}\overline{\boldsymbol{M}}\,\overline{\boldsymbol{M}}^{\top} + \boldsymbol{\Sigma}_W\right)^{-1}.$$

The combined presence of the terms $\overline{\boldsymbol{M}}\,\overline{\boldsymbol{M}}^{\top}$ and $\boldsymbol{\Sigma}_W$ in the denominator, along with $\overline{\boldsymbol{M}}$ in the numerator, signals to us the presence of the informative *signal-to-noise (SNR) ratio matrix*:

$$\text{SNR} \equiv \boldsymbol{\Sigma}_W^{-\frac{1}{2}}\overline{\boldsymbol{M}}. \qquad (8)$$

Why this terminology? The globally-centered class-means represent the overall "signal" indicating that the class-means are separated from each other: if they were non-separated—e.g. all equal—$\overline{\boldsymbol{M}}$ would be zero. However, a classifier decision must keep in view noise—captured by $\boldsymbol{\Sigma}_W$—which might confuse the classifier. So, one wishes the signal be large compared to the noise. It may help to consider the mental model—which is not at all necessary to the correctness of our analysis—under which the activations are normally distributed with $\boldsymbol{h}_{i,c} \sim \mathcal{N}(\boldsymbol{\mu}_c, \boldsymbol{\Sigma}_W)$. Under that model, a linear classifier will indeed get confused if the norm $\|\boldsymbol{\mu}_c - \boldsymbol{\mu}_G\|_2$ is "small compared" to $\boldsymbol{\Sigma}_W$. Replacing this somewhat vague statement by a discussion of quantitative properties of the SNR matrix is well understood by mathematical statisticians to be decisive for understanding classification performance in the normal case. Additionally, observe that $\text{SNR} = \overline{\boldsymbol{M}}(\boldsymbol{\Sigma}_W^{-\frac{1}{2}}\overline{\boldsymbol{H}})$. Thus, connecting back to Section 3.1, *the SNR matrix is simply the class-means matrix of the renormalized features.*

The SNR can be further understood through its singular value decomposition (SVD).

**Definition 2** (**SVD of SNR Matrix**). *Denote the SVD of the SNR matrix (Equation 8) as follows:*

$$\text{SNR} = \boldsymbol{U}\boldsymbol{\Omega}\boldsymbol{V}^{\top} = \sum_{j=1}^{C-1}\omega_j\boldsymbol{u}_j\boldsymbol{v}_j^{\top}.$$

*Here, $\{\omega_c\}_{c=1}^{C-1}$ are the non-zero singular values of $\text{SNR}$; $\boldsymbol{\Omega} = \text{diag}\left(\{\omega_c\}_{c=1}^{C-1}, 0\right) \in \mathbb{R}^{C\times C}$; and the left and right singular-vectors, $\boldsymbol{U} \in \mathbb{R}^{P\times C}$ and $\boldsymbol{V} \in \mathbb{R}^{C\times C}$, are partial orthogonal and orthogonal matrices, respectively, with columns $\{\boldsymbol{u}_j\}_{j=1}^{C}$ and $\{\boldsymbol{v}_j\}_{j=1}^{C}$. The SNR matrix is rank $C-1$ since $\boldsymbol{\Sigma}_W$ is assumed full-rank and $\overline{\boldsymbol{M}}$ is rank $C-1$ due to global-mean subtraction.*

The non-zero singular values, $\{\omega_j\}_{j=1}^{C-1}$, are decisive for understanding the separation performance of the least-squares linear classifier. Good performance demands that the singular values be large. Works such as Fukunaga (1972, Chapter 10) and Hastie & Tibshirani (1996) show that the magnitude of each singular value of the SNR matrix is the size of the class separation in the direction of the corresponding singular vector. In this sense, the smallest singular value captures the smallest separation between classes. Consequently, driving-larger the smallest non-zero singular value makes the task of linear separation more immune to (spherical) noise and to Euclidean-norm-constrained adversarial noise. As we will see in the next section, examining the dynamics of these singular-values when the features undergo gradient flow in renormalized coordinates leads to precise characterizations of Neural Collapse[13].

## 3.3 DYNAMICS OF SNR

For any matrix $\boldsymbol{Z}$, we use $\boldsymbol{Z}_t$ in this section to denote the state of that matrix at time $t$. Similarly, we denote with $\{\omega_j(t)\}_{j=1}^{C-1}$ the state of the non-zero SNR singular values at $t$. We now present Proposition 2 and Corollaries 1-2 that provide the explicit dynamics referenced by Theorem 3.

**Proposition 2** (**Dynamics of Singular Values of SNR Matrix; Proof in Appendix D.4**). *Continually renormalized gradient flow on the central path (Equation 5) induces the following closed-form dynamics on the SNR singular values (Definition 2):*

$$c_1\log(\omega_j(t)) + c_2\omega_j^2(t) + c_3\omega_j^4(t) = a_j + t, \quad t \geq 0, \quad \text{for all } j = 1,\ldots,C-1. \qquad (9)$$

*$c_1$, $c_2$, and $c_3$ are positive constants independent of $j$, and $a_j$ is a constant depending on $\omega_j(0)$.*

---

[13]For additional, supplementary intuitions about invariance, renormalization, and continually renormalized gradient flow, see the exposition in Appendices D.1-D.3.

**Corollary 1** (**Properties of SNR Singular Values; Proof in Appendix D.5**). *SNR singular values (Definition 2) following the Equation 9 dynamics satisfy the following limiting behaviors:*

1. $\lim_{t \to \infty} \omega_j(t) = \infty$ *and* $\lim_{t \to \infty} \frac{\omega_j(t)}{\sqrt[4]{t/c_3}} = 1$, *for all* $j = 1, \ldots, C - 1$.

2. $\lim_{t \to \infty} \frac{\max_j \omega_j(t)}{\min_j \omega_j(t)} = 1$.

The first fact shows that all non-zero singular values of the SNR matrix, $\mathbf{\Sigma}_W^{-\frac{1}{2}} \overline{\mathbf{M}}$, grow to infinity at an asymptotic rate of $\sqrt[4]{t/c_3}$. Intuitively, this shows that the "signal" becomes infinitely large compared to the "noise", i.e. **(NC1)**. The second fact shows that the singular values of $\mathbf{\Sigma}_W^{-\frac{1}{2}} \overline{\mathbf{M}}$ (which has zero-mean columns) approach equality—implying that the limiting matrix is a Simplex ETF (Lemma 7, Appendix D.6) i.e. **(NC2)**. Finally, Papyan, Han, and Donoho (2020, Theorem 1) show that **(NC1-2)** imply **(NC3-4)** on the central path. Corollary 2 formalizes these intuitions.

**Corollary 2** (**Neural Collapse Under MSE Loss; Proof in Appendix D.6**). *Under continually renormalized gradient flow (Equation 5), the SNR matrix (Equation 8) converges to*

$$\lim_{t \to \infty} \frac{1}{\omega_{\max}(t)} \mathrm{SNR}_t = \widehat{\mathbf{U}}_0 \widehat{\mathbf{V}}_0^\top, \tag{10}$$

*where* $\widehat{\mathbf{U}}_0 \in \mathbb{R}^{P \times (C-1)}$ *and* $\widehat{\mathbf{V}}_0 \in \mathbb{R}^{C \times (C-1)}$ *are the left and right singular vectors of the SNR matrix (Definition 2) at $t=0$ corresponding to the non-zero singular values; and $\omega_{\max}(t)$ is the largest singular value at time t. Furthermore, Corollary 1 implies the occurrence of (NC1)-(NC4) i.e. renormalized gradient flow on the central path leads to Neural Collapse.*

*Moreover, denoting the Kronecker product with $\otimes$, the renormalized features matrix converges to*

$$\lim_{t \to \infty} \frac{1}{\omega_{\max}(t)} \mathbf{\Sigma}_{W,t}^{-\frac{1}{2}} \overline{\mathbf{H}}_t = (\widehat{\mathbf{U}}_0 \widehat{\mathbf{V}}_0^\top) \otimes \mathbb{1}_N^\top. \tag{11}$$

## 4 RELATED WORKS

Since the publication of Papyan, Han, and Donoho (2020), several works (Mixon et al., 2020; Lu & Steinerberger, 2020; E & Wojtowytsch, 2020; Poggio & Liao, 2020a;b; Fang et al., 2021; Ergen & Pilanci, 2020; Zhu et al., 2021) proposed mathematical frameworks ratifying Neural Collapse.

Among them, Mixon et al. (2020) and Poggio & Liao (2020a;b) also examine MSE-NC. Our contributions are *distinct* from the MSE-NC analyses of Mixon et al. (2020) and Poggio & Liao (2020a;b): The work of Mixon et al. (2020) relies on a *linearization* of the unconstrained features model ODE while Poggio & Liao (2020a;b)—in a special section co-authored with Andrzej Banburski—examine *homogeneous, weight-normalized* networks. In contrast, this paper considers the dynamics of the renormalized gradient flow on the central path (motivated by the discussions in Sections 2-3). Neither Mixon et al. (2020) nor Poggio & Liao (2020a;b) provide exact, closed-form dynamics as we do.

Outside the MSE setting, Lu & Steinerberger (2020); E & Wojtowytsch (2020); Ergen & Pilanci (2020); Fang et al. (2021); Zhu et al. (2021) examine the emergence of the NC properties under variants of the unconstrained features/layer-peeled model trained with CE loss. These works focus on characterizing the loss landscape or the global minima without describing the dynamics.

We provide a detailed survey of all above-mentioned papers in Appendix E.

## 5 CONCLUSION

In this paper, after verifying that NC occurs when prototypical classification deep nets are trained with MSE loss on canonical datasets, we then derive and measure a novel decomposition of the MSE loss. We observed that the last-layer classifier tends to the least-squares classifier—motivating us to define the central path on which the classifier behaves exactly as optimal least-squares classifier. On the central path, we showed invariance properties that inspired us to examine the renormalized features and their corresponding continually renormalized gradient flow. This flow induces closed-form dynamics that imply the occurrence of Neural Collapse.

## REPRODUCIBILITY STATEMENT

This paper is reproducible. Experimental details about all empirical results described in this paper are provided in Appendix A. Additionally, we provide PyTorch (Paszke et al., 2019) code for reproducing NC under both MSE and CE—as well as the Figure 2 MSE decomposition—in the following Google Colaboratory notebook: here. Experimental measurements used to generated the plots in this paper have been deposited in the Stanford Digital Repository, here. All datasets and networks used in our experiments originate from the PyTorch Model Zoo with pre-processing details described in Appendix A. Formal statements and proofs of all our theoretical results are provided in Appendices B-D.

## ACKNOWLEDGEMENTS

Some of the computing for this project was performed on the Sherlock cluster. We would like to thank Stanford University and the Stanford Research Computing Center for providing computational resources and support that contributed to these research results.

We acknowledge the support of the Natural Sciences and Engineering Research Council of Canada (NSERC), [funding reference number 512236 and 512265]. This research was funded by a Connaught New Researcher Award at the University of Toronto. This research was enabled in part by support provided by Compute Ontario (www.computeontario.ca) and Compute Canada (www.computecanada.ca). This work was also partially supported by NSF Division of Mathematical Sciences Grants 1407813, 1418362, and 1811614 and by private donors.

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

APPENDIX

# A    MSE Neural Collapse experiments

The experiments in this section examine the properties of Neural Collapse (NC) on deep nets trained using MSE loss. The direct MSE-analogues to the cross-entropy (CE) loss table and figures in Papyan, Han, and Donoho (2020) are in Table 1 and Figures 3-9 here. Furthermore, Figures 10-11 compares the MSE-NC experiments observed in this paper with the CE-NC behaviors in Papyan, Han, and Donoho (2020). Experimental descriptions and discussions are within the captions. Subsections A.1-A.5 describe formatting and experimental details. Then, Subsection A.6 collectively presents all experimental figures for this section. Subsection A.7 addresses outlier behaviors observed in the plots. Finally, Subsection A.8 discusses open questions inspired by NC.

## A.1    Datasets

We consider the MNIST, FashionMNIST, CIFAR10, SVHN, and STL10 datasets with pre-processing the same as in Papyan, Han, and Donoho (2020).

### A.1.1    Choice of datasets

Demirkaya et al. (2020) and Hui & Belkin (2020) showed multi-class classification deep nets trained with MSE loss—sometimes with a heuristical-scaling—demonstrate comparable test-error performance to those trained with CE loss. Even without additional scaling, i.e. with vanilla MSE loss, we found this true for ten class classification datasets—such as MNIST, FashionMNIST, SVHN, CIFAR10, and STL10—so we focus on these five datasets. For more than ten classes (i.e. datasets such as CIFAR100 and ImageNet), both Demirkaya et al. (2020) and Hui & Belkin (2020) showed that the additional scaling-heuristic does need to be applied to the loss before comparable test-performance can be achieved. In informal, exploratory experiments not reported here, we were able to reproduce their results on datasets with more than ten classes. But, we feel these scaling-heuristics merit further scientific investigation of their own and are beyond the scope of this article[14], so we do not include datasets that require scaling-heuristic tuning.

## A.2    Networks

We train VGG, ResNet, and DenseNet models with the same depth-selection procedures and architecture-specification choices as in Papyan, Han, and Donoho (2020). In the MSE setting, the final chosen depths were as follows:

| Dataset | VGG | ResNet | DenseNet |
|---|---|---|---|
| MNIST | VGG11 | ResNet18 | DenseNet40 |
| FashionMNIST | VGG11 | ResNet18 | DenseNet250 |
| SVHN | VGG13 | ResNet34 | DenseNet40 |
| CIFAR10 | VGG11 | ResNet50 | DenseNet100 |
| STL10 | VGG11 | ResNet18 | DenseNet201 |

## A.3    Optimization methodology

The optimization algorithm, parameters, and hyperparameter tuning are the same as in Papyan, Han, and Donoho (2020).

## A.4    Computational resources

The experiments were run on Stanford University's Sherlock computing cluster. Each dataset-network combination was trained on a single GPU attached to a CPU with at least 32GB of RAM—the specific types of the CPUs/GPUs vary according to whichever was first assigned to us by the HPC cluster scheduler.

---

[14]Thorough experimentation with this scaling-heuristic would also consume prohibitively expensive amounts of computational resources.

## A.5 FORMATTING

The coloring and formatting of the plots are the same as in Papyan, Han, and Donoho (2020).

## A.6 EXPERIMENTAL RESULTS

Table 1: Table comparing test-accuracy at moment 0-error is achieved vs. at the end of training. Analogous to Table 1 of Papyan, Han, and Donoho (2020). The median improvement is 0.962 percentage points; the mean is 1.833 percentage points.

| DATASET | NET | ACC. 0-ERROR | ACC. FINAL |
|---------|-----|--------------|------------|
| MNIST | VGG | 99.23 | 99.59 |
| | RESNET | 99.16 | 99.70 |
| | DENSENET | 99.62 | 99.70 |
| FASHION | VGG | 92.76 | 92.95 |
| | RESNET | 93.55 | 93.76 |
| | DENSENET | 90.56 | 92.95 |
| SVHN | VGG | 89.35 | 93.91 |
| | RESNET | 85.18 | 92.65 |
| | DENSENET | 95.61 | 95.23 |
| CIFAR10 | VGG | 83.13 | 84.54 |
| | RESNET | 75.43 | 76.39 |
| | DENSENET | 91.77 | 91.78 |
| STL10 | VGG | 60.43 | 67.24 |
| | RESNET | 59.35 | 60.56 |
| | DENSENET | 58.74 | 60.41 |

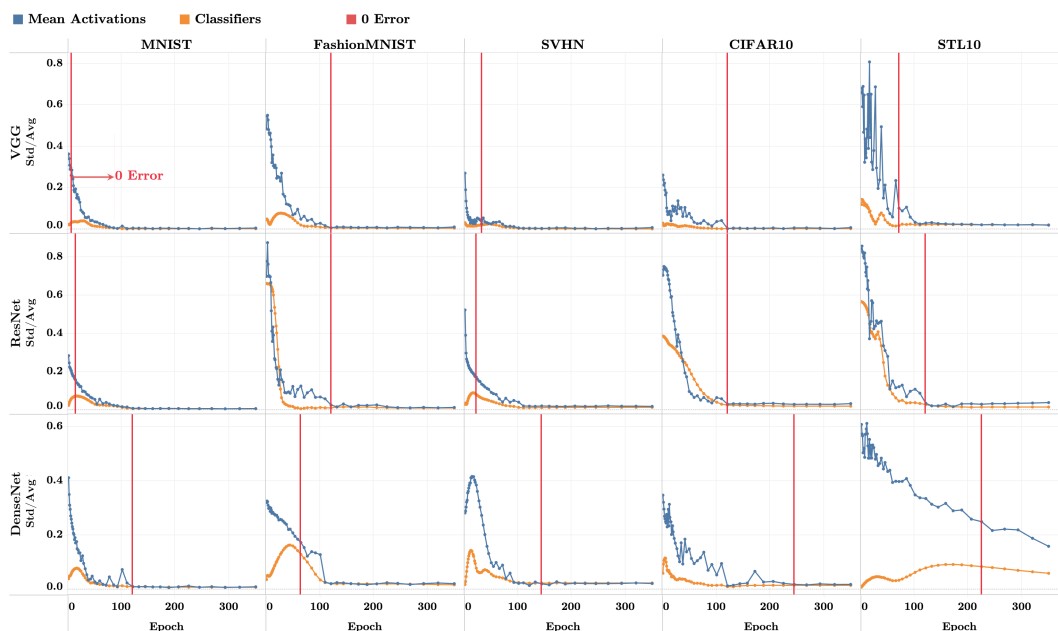

Figure 3: Plots analogous to Figure 2 in Papyan et al. (2020), but on networks trained with MSE Loss. Results demonstrate that last-layer features and classifiers approach *equinormness*.

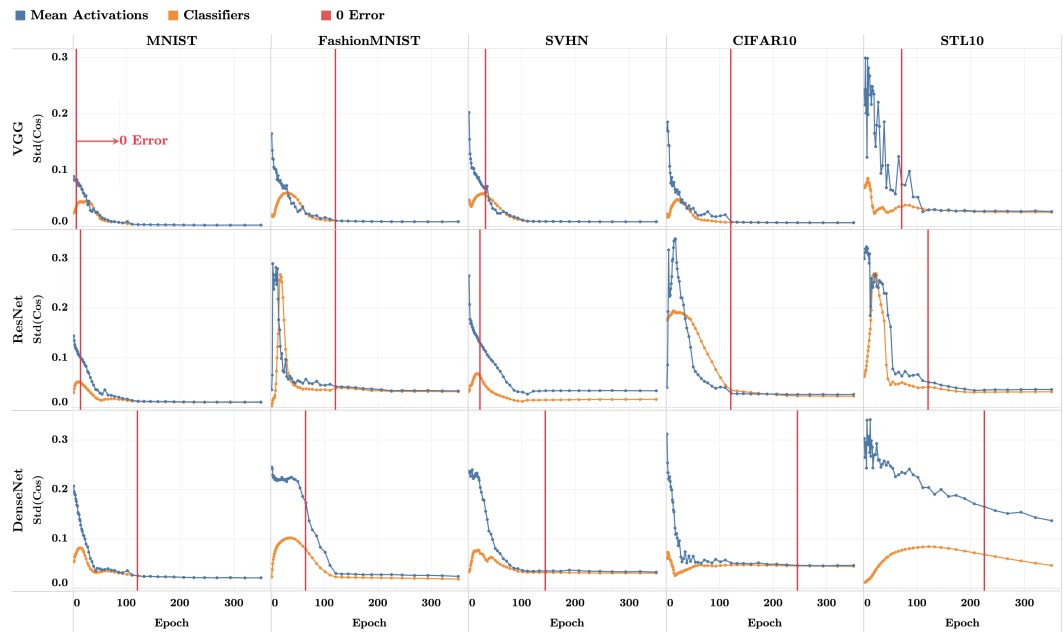

Figure 4: Plots analogous to Figure 3 in Papyan et al. (2020), but on networks trained with MSE Loss. Results demonstrate that last-layer features and classifiers approach *equiangularity*.

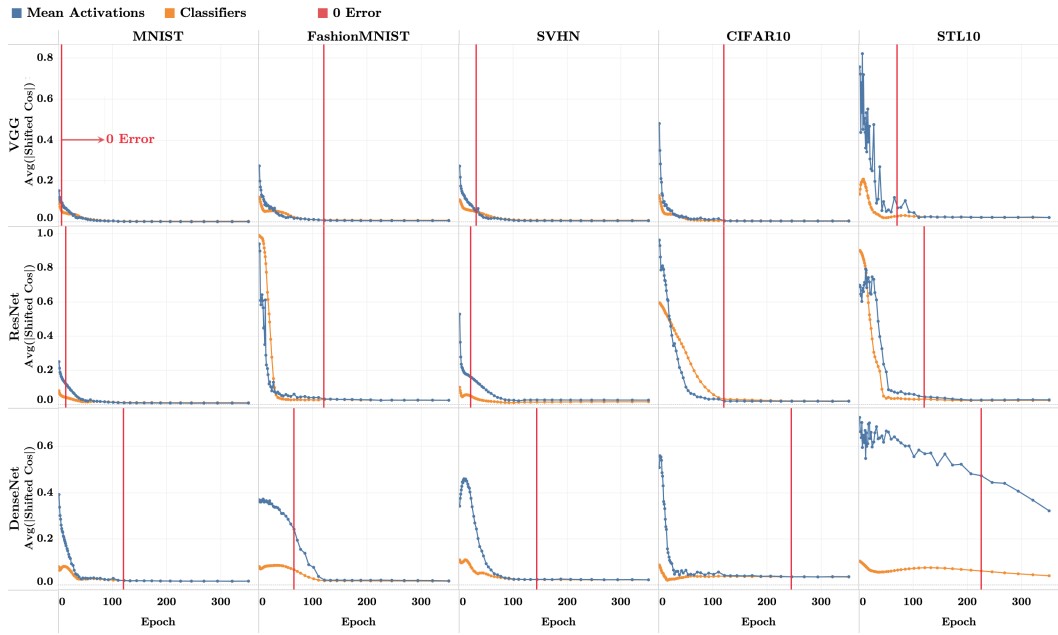

Figure 5: Plots analogous to Figure 4 in Papyan et al. (2020), but on networks trained with MSE Loss. Results demonstrate that last-layer features and classifiers approach *maximal-equiangularity*.

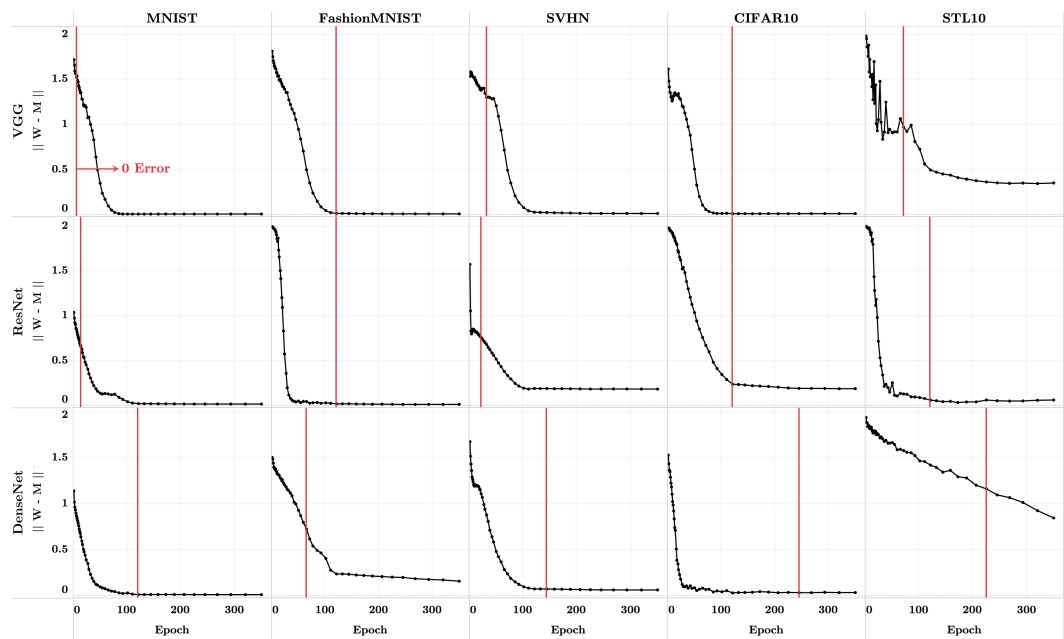

Figure 6: Plots analogous to Figure 5 in Papyan et al. (2020), but on networks trained with MSE Loss. Results demonstrate that last-layer features and classifiers approach *self-duality*.

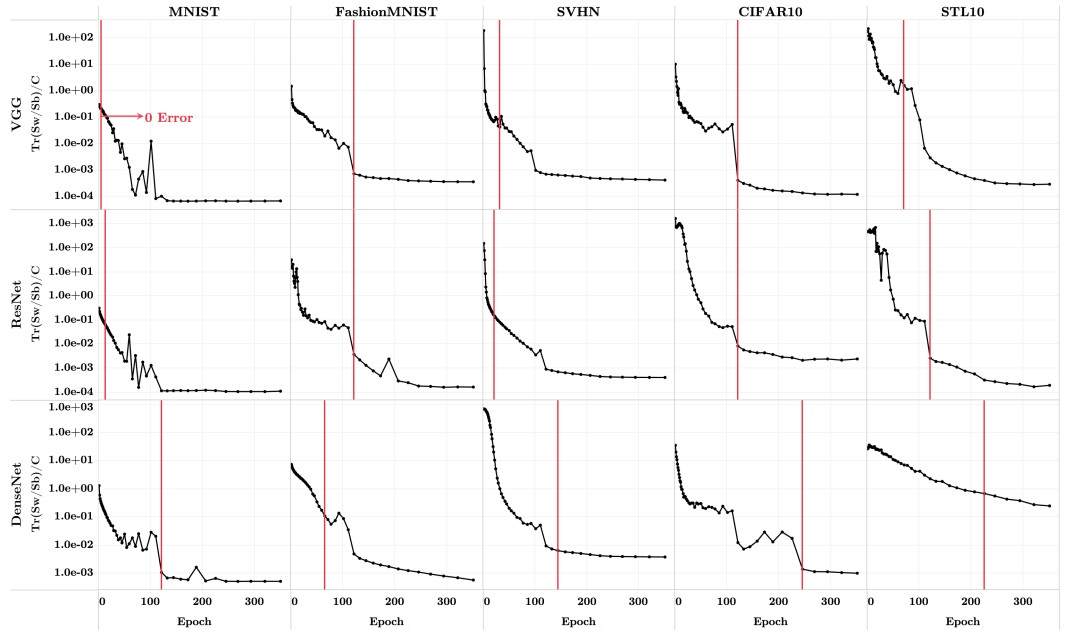

Figure 7: Plots analogous to Figure 6 in Papyan et al. (2020), but on networks trained with MSE Loss. Results demonstrate that last-layer features undergo *variability collapse*.

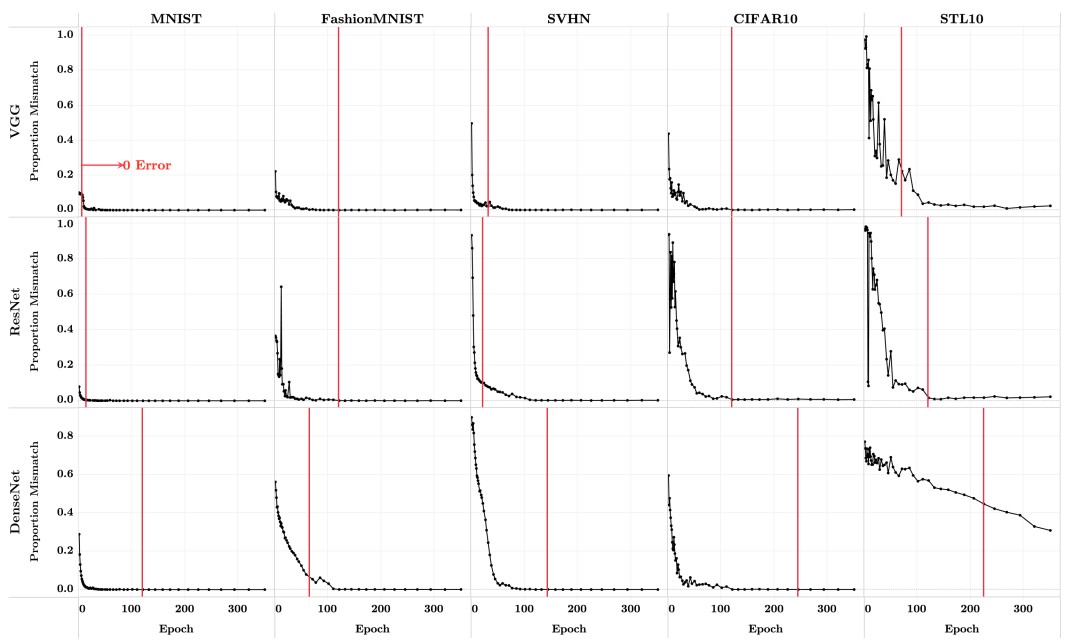

Figure 8: Plots analogous to Figure 7 in Papyan et al. (2020), but on networks trained with MSE Loss. Results demonstrate that classifier decisions converge to those of the *nearest class center decision rule*.

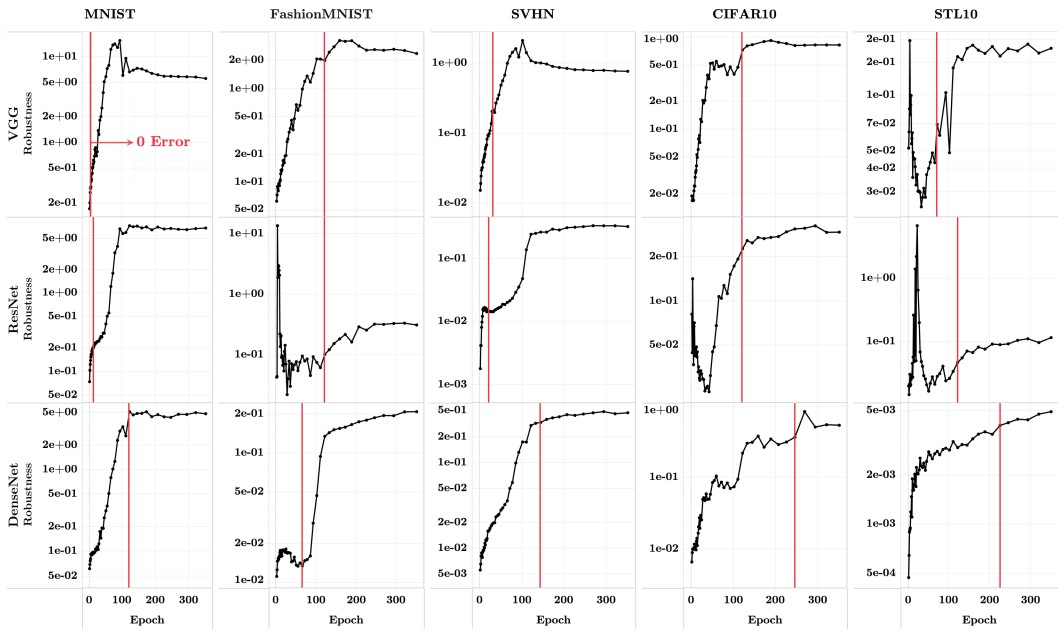

Figure 9: Plots analogous to Figure 8 in Papyan et al. (2020), but on networks trained with MSE Loss. Results demonstrate that networks *become more robust when trained beyond 0-error*. The median improvement in the robustness measure in the last epoch over the first epoch achieving zero training error is 0.1762; the mean improvement is 0.9278.

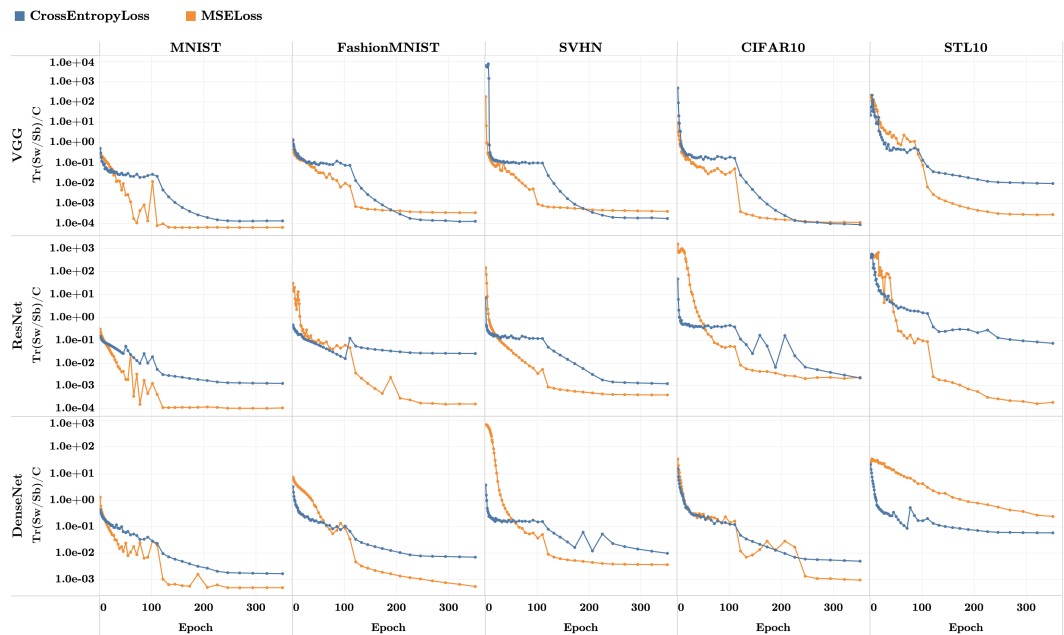

Figure 10: *Activation collapse under MSE loss vs. CE loss:* Comparison of activation collapse observed in this paper for networks trained under MSE loss (Figure 7) with that observed in Papyan et al. (2020) under CE loss. Networks trained with MSE loss tend to achieve faster activation collapse than those trained with CE.

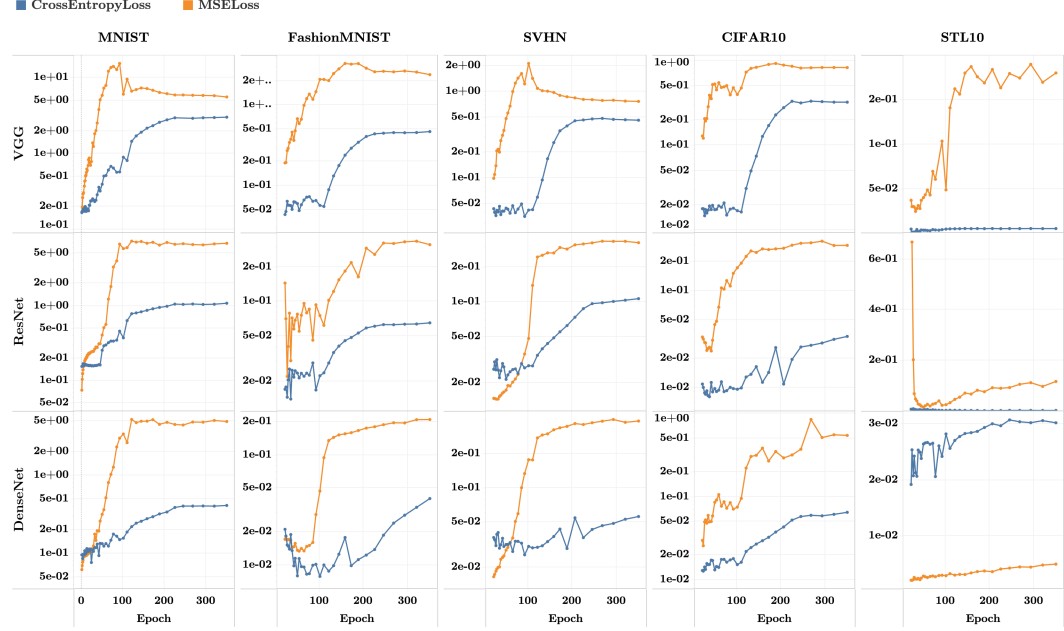

Figure 11: *Adversarial robustness under MSE loss vs. CE loss:* Comparison of adversarial robustness observed in this paper for networks trained under MSE loss (Figure 9) with that observed in Papyan et al. (2020) under CE loss. Robustness tends to be better—sometimes several magnitudes better—when the networks are trained with MSE loss than with CE loss.

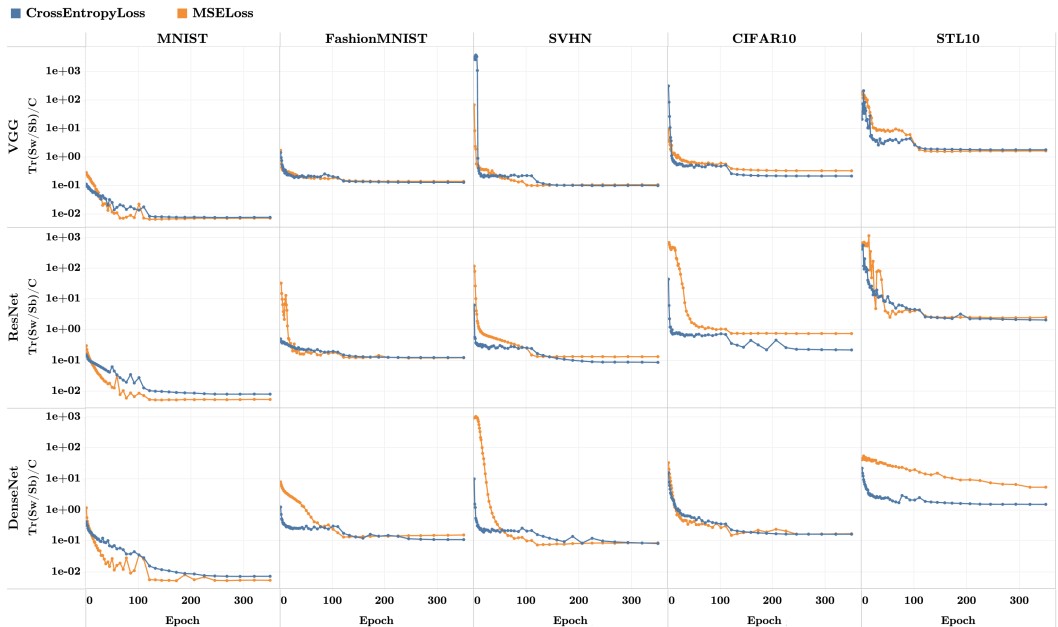

Figure 12: *Activation collapse on test data for both losses:* Activation collapse observed on *test data* for models trained with MSE loss (from this current paper) and CE loss (posted by Papyan, Han, and Donoho (2020) on Stanford Data Repository). On the test data, activation collapse still visibly occurs on multiple dataset-network combinations: Albeit the rate of collapse is much slower on the test data compared to that on the train data, and the plotted measure (described in Figure 6 of Papyan, Han, and Donoho (2020)) at the last epoch is larger than that on the train data. Also interesting is that the value at the last epoch is roughly monotonic with the difficulty of the dataset. See discussion in Section A.8.

## A.7 EXCEPTIONS: STL10-RESNET AND STL10-DENSENET

The STL10-ResNet and STL10-DenseNet dataset-network pairs stand out among the experiments described in this section—sometimes displaying outlier behavior either by converging more slowly to Neural Collapse or by exhibiting trends inconsistent with those found in other dataset-network pairs.

STL10 stands apart from the other canonical datasets for multiple reasons[15] and is a "less-typical" benchmark compared to more "compulsory" datasets like CIFAR10. In this particular case, we hypothesize that its outlier behavior might be caused by the size of the STL10 images ($96 \times 96$ compared to the $32 \times 32$, after padding, of the other datasets)—leading to a higher-dimensional problem, which, in turn, induces a harder non-convex optimization problem, which might make SGD less likely to converge to a useful optimum or might make the convergence much slower and harder to observe under a fixed computational budget.

It was previously noted in Hui & Belkin (2020) and Demirkaya et al. (2020) that more challenging classification problems—in those projects, problems with more classes to be labeled—may require modifications to the MSE loss. We decided that such modifications are unnecessary in the ten class problems examined in this paper's experiments. But perhaps the outlier nature of the STL10-ResNet and STL10-DenseNet combinations signal that the MSE loss modifications proposed by Hui & Belkin (2020) and Demirkaya et al. (2020) ought to have been used.

## A.8 OPEN QUESTIONS: WEIGHT-DECAY, BATCH NORMALIZATION, SGD, GENERALIZATION, TEST DATA

The experiments in this paper as well as those in Papyan, Han, and Donoho (2020) were conducted under the canonical setting where deep nets were trained using SGD with weight-decay and batch normalization—leading one to wonder about the roles that these particular ingredients play on the NC phenomena. Additionally, most observables focus on the train data—raising the question of how the NC phenomena behave on test data as well as their relationship to generalization. We find these questions intriguing, but feel each deserves careful experimentation outside the scope of this paper.

Nonetheless, we note here that several existing papers have already begun insightful investigations in these directions. For example, weight-decay, weight-normalization (as a proxy for batch-normalization), and SGD play key roles in the analyses of Banburski et al. (2019); Poggio & Liao (2020a;b) that, along other things, lead to the prediction of NC in homogeneous deep nets. Banburski et al. (2021) also explores the connection between NC and margin distributions with generalization.

Another example is Zhu et al. (2021) in which the authors conduct several experiments on ResNets trained on CIFAR10. On their models, the authors examine NC-related properties relative to the train data, test data, and randomly labeled data; they also conducted ablation studies varying control parameters (weight-decay, width, etc.) and the optimization algorithm (SGD, ADAM, L-BFGS). Based on their results, Zhu et al. (2021) propose thought-provoking conjectures on the role of each of these components (see Section E.7 for a brief survey of their findings).

As a preliminary exploration, we include Figure 12 showing the variability collapse (**NC1**) behavior on *test data* for the networks trained with MSE loss (used in this paper) as well as those trained with CE loss (released[16] by Papyan, Han, and Donoho (2020)). From Figure 12, we see that variability collapse occurs much slower on the test data than on the train data. Although not shown here, the other NC phenomena behave similarly.

In this Appendix, we show Figure 12 since it is concise, within reach, and may be of interest to readers who share our curiosity. We refrain from presenting the entire series of multi-figure NC measurements on test data because it is outside the scope of this paper and would significantly lengthen this appendix. We also refrain from speculating on any underlying explanations until more meticulous experiments are conducted.

---

[15]For example, one sense in which STL-10 is clearly different from the other datasets is the comparatively small number of labeled training examples per class.

[16]The CE test data results were released by Papyan, Han, and Donoho (2020) on the paper's corresponding Stanford Data Repository entry.

## B  THEOREM 1

**Theorem 1.** *(Decomposition of MSE Loss) The MSE loss, $\mathcal{L}(\widetilde{\boldsymbol{W}}, \widetilde{\boldsymbol{H}})$, can be decomposed into two terms, $\mathcal{L}(\widetilde{\boldsymbol{W}}, \widetilde{\boldsymbol{H}}) = \mathcal{L}_{LS}(\widetilde{\boldsymbol{H}}) + \mathcal{L}_{LS}^{\perp}(\widetilde{\boldsymbol{W}}, \widetilde{\boldsymbol{H}})$, where*

$$\mathcal{L}_{LS}(\widetilde{\boldsymbol{H}}) = \frac{1}{2} \operatorname*{Ave}_{i,c} \|\widetilde{\boldsymbol{W}}_{LS}\widetilde{\boldsymbol{h}}_{i,c} - \boldsymbol{y}_{i,c}\|_2^2 + \frac{\lambda}{2}\|\widetilde{\boldsymbol{W}}_{LS}\|_F^2,$$

*and*

$$\mathcal{L}_{LS}^{\perp}(\widetilde{\boldsymbol{W}}, \widetilde{\boldsymbol{H}}) = \frac{1}{2}\operatorname{tr}\left\{(\widetilde{\boldsymbol{W}} - \widetilde{\boldsymbol{W}}_{LS})\left(\widetilde{\boldsymbol{\Sigma}}_T + \widetilde{\boldsymbol{\mu}}_G\widetilde{\boldsymbol{\mu}}_G^{\top} + \lambda\boldsymbol{I}\right)(\widetilde{\boldsymbol{W}} - \widetilde{\boldsymbol{W}}_{LS})^{\top}\right\}.$$

*Proof.* Recall the MSE loss:

$$\mathcal{L}(\widetilde{\boldsymbol{W}}, \widetilde{\boldsymbol{H}}) = \frac{1}{2}\operatorname*{Ave}_{i,c}\|\widetilde{\boldsymbol{W}}\widetilde{\boldsymbol{h}}_{i,c} - \boldsymbol{y}_{i,c}\|_2^2 + \frac{\lambda}{2}\|\widetilde{\boldsymbol{W}}\|_F^2. \tag{12}$$

Consider a specific extended activations matrix $\widetilde{\boldsymbol{H}}$; the least-squares solution $\widetilde{\boldsymbol{W}}$ minimizing $\mathcal{L}(\widetilde{\boldsymbol{W}}, \widetilde{\boldsymbol{H}})$ *with $\widetilde{\boldsymbol{H}}$ kept fixed* must obey the following first-order optimality condition:

$$\operatorname*{Ave}_{i,c}(\widetilde{\boldsymbol{W}}_{\mathrm{LS}}\widetilde{\boldsymbol{h}}_{i,c} - \boldsymbol{y}_{i,c})\widetilde{\boldsymbol{h}}_{i,c}^{\top} + \lambda\widetilde{\boldsymbol{W}}_{\mathrm{LS}} = 0. \tag{13}$$

From Proposition 1 in the main manuscript, the solution to the above is given by

$$\widetilde{\boldsymbol{W}}_{\mathrm{LS}} = C^{-1}\widetilde{\boldsymbol{M}}^{\top}(\widetilde{\boldsymbol{\Sigma}}_T + \widetilde{\boldsymbol{\mu}}_G\widetilde{\boldsymbol{\mu}}_G^{\top} + \lambda\boldsymbol{I})^{-1}.$$

The loss can be rewritten as

$$\frac{1}{2}\operatorname*{Ave}_{i,c}\|\widetilde{\boldsymbol{W}}_{\mathrm{LS}}\widetilde{\boldsymbol{h}}_{i,c} - \boldsymbol{y}_{i,c} + (\widetilde{\boldsymbol{W}} - \widetilde{\boldsymbol{W}}_{\mathrm{LS}})\widetilde{\boldsymbol{h}}_{i,c}\|_2^2 + \frac{\lambda}{2}\|\widetilde{\boldsymbol{W}}\|_F^2.$$

Combining the above with Equation 13 gives, after rearranging:

$$\frac{1}{2}\operatorname*{Ave}_{i,c}\|\widetilde{\boldsymbol{W}}_{\mathrm{LS}}\widetilde{\boldsymbol{h}}_{i,c} - \boldsymbol{y}_{i,c}\|_2^2 + \frac{1}{2}\operatorname*{Ave}_{i,c}\|(\widetilde{\boldsymbol{W}} - \widetilde{\boldsymbol{W}}_{\mathrm{LS}})\widetilde{\boldsymbol{h}}_{i,c}\|_2^2 - \lambda\operatorname{tr}\left\{\widetilde{\boldsymbol{W}}_{\mathrm{LS}}(\widetilde{\boldsymbol{W}} - \widetilde{\boldsymbol{W}}_{\mathrm{LS}})^{\top}\right\} + \frac{\lambda}{2}\|\widetilde{\boldsymbol{W}}\|_F^2,$$

which is equivalent to

$$\frac{1}{2}\operatorname*{Ave}_{i,c}\|\widetilde{\boldsymbol{W}}_{\mathrm{LS}}\widetilde{\boldsymbol{h}}_{i,c} - \boldsymbol{y}_{i,c}\|_2^2 + \frac{1}{2}\operatorname*{Ave}_{i,c}\|(\widetilde{\boldsymbol{W}} - \widetilde{\boldsymbol{W}}_{\mathrm{LS}})\widetilde{\boldsymbol{h}}_{i,c}\|_2^2 + \frac{\lambda}{2}\|\widetilde{\boldsymbol{W}}_{\mathrm{LS}}\|_F^2 + \frac{\lambda}{2}\|\widetilde{\boldsymbol{W}}_{\mathrm{LS}} - \widetilde{\boldsymbol{W}}\|_F^2.$$

Using the above, the loss can indeed be decomposed as

$$\mathcal{L}(\widetilde{\boldsymbol{W}}, \widetilde{\boldsymbol{H}}) = \mathcal{L}_{\mathrm{LS}}(\widetilde{\boldsymbol{H}}) + \mathcal{L}_{\mathrm{LS}}^{\perp}(\widetilde{\boldsymbol{W}}, \widetilde{\boldsymbol{H}}),$$

where

$$\mathcal{L}_{\mathrm{LS}}(\widetilde{\boldsymbol{H}}) \equiv \frac{1}{2}\operatorname*{Ave}_{i,c}\|\widetilde{\boldsymbol{W}}_{\mathrm{LS}}\widetilde{\boldsymbol{h}}_{i,c} - \boldsymbol{y}_{i,c}\|_2^2 + \frac{\lambda}{2}\|\widetilde{\boldsymbol{W}}_{\mathrm{LS}}\|_F^2 \tag{14}$$

and

$$\begin{aligned}\mathcal{L}_{\mathrm{LS}}^{\perp}(\widetilde{\boldsymbol{W}}, \widetilde{\boldsymbol{H}}) &\equiv \frac{1}{2}\operatorname*{Ave}_{i,c}\|(\widetilde{\boldsymbol{W}} - \widetilde{\boldsymbol{W}}_{\mathrm{LS}})\widetilde{\boldsymbol{h}}_{i,c}\|_2^2 + \frac{\lambda}{2}\|\widetilde{\boldsymbol{W}}_{\mathrm{LS}} - \widetilde{\boldsymbol{W}}\|_F^2 \\ &= \frac{1}{2}\operatorname{tr}\left\{(\widetilde{\boldsymbol{W}} - \widetilde{\boldsymbol{W}}_{\mathrm{LS}})\left(\widetilde{\boldsymbol{\Sigma}}_T + \widetilde{\boldsymbol{\mu}}_G\widetilde{\boldsymbol{\mu}}_G^{\top} + \lambda I\right)(\widetilde{\boldsymbol{W}} - \widetilde{\boldsymbol{W}}_{\mathrm{LS}})^{\top}\right\}.\end{aligned}$$

This completes the proof. $\qquad\square$

## C  THEOREM 2

**Theorem 2.** *(Decomposition of Least-Squares Component) The least-squares component, $\mathcal{L}_{LS}(\widetilde{\boldsymbol{H}})$, of the MSE decomposition in Theorem 1 can be further decomposed into $\mathcal{L}_{LS}(\widetilde{\boldsymbol{H}}) = \mathcal{L}_{NC1}(\widetilde{\boldsymbol{H}}) + \mathcal{L}_{NC2/3}(\widetilde{\boldsymbol{H}})$, where*

$$\mathcal{L}_{NC1}(\widetilde{\boldsymbol{H}}) = \frac{1}{2}\operatorname{tr}\left\{\widetilde{\boldsymbol{W}}_{LS}\left[\widetilde{\boldsymbol{\Sigma}}_W + \lambda\boldsymbol{I}\right]\widetilde{\boldsymbol{W}}_{LS}^{\top}\right\},$$

$$\mathcal{L}_{NC2/3}(\widetilde{\boldsymbol{H}}) = \frac{1}{2C}\|\widetilde{\boldsymbol{W}}_{LS}\widetilde{\boldsymbol{M}} - \boldsymbol{I}\|_F^2.$$

*Proof.* Under Euclidean distance, perturbations to the extended activations matrix $\widetilde{H}$ which affect only the class-means or global-mean are *orthogonal* to perturbations which affect only the within-class covariance. Using this, $\mathcal{L}_{\mathrm{LS}}(\widetilde{H})$ can be further decomposed via the Pythagorean theorem:

$$\mathcal{L}_{\mathrm{LS}}(\widetilde{H}) = \frac{1}{2} \operatorname*{Ave}_{i,c} \|\widetilde{W}_{\mathrm{LS}}(\widetilde{h}_{i,c} - \widetilde{\mu}_c)\|_2^2 + \frac{\lambda}{2}\|\widetilde{W}_{\mathrm{LS}}\|_F^2 + \frac{1}{2} \operatorname*{Ave}_c \|\widetilde{W}_{\mathrm{LS}}\widetilde{\mu}_c - y_{i,c}\|_2^2.$$

The first and second terms above merge into $\mathcal{L}_{\mathrm{NC1}}(\widetilde{H})$, while the third term becomes $\mathcal{L}_{\mathrm{NC2/3}}(\widetilde{H})$:

$$\mathcal{L}_{\mathrm{NC1}}(\widetilde{H}) = \frac{1}{2}\operatorname{tr}\left\{\widetilde{W}_{\mathrm{LS}}\left(\widetilde{\Sigma}_W + \lambda I\right)\widetilde{W}_{\mathrm{LS}}^\top\right\} \tag{15}$$

$$\mathcal{L}_{\mathrm{NC2/3}}(\widetilde{H}) = \frac{1}{2C}\|\widetilde{W}_{\mathrm{LS}}\widetilde{M} - I\|_F^2.$$

This completes the proof. $\qquad\square$

## C.1 Intuitions for Theorem 2 in unextended coordinates

For intuition, consider when $\lambda = 0$, i.e. the no weight-decay case. Proposition 1 in the main text gives

$$W_{\mathrm{LS}} = C^{-1}\overline{M}^\top \Sigma_T^{-1}, \tag{16}$$

where $\overline{M} \in \mathbb{R}^{P \times C}$ is the matrix with columns $\mu_c - \mu_G$. Returning to the unextended coordinates, $\mathcal{L}_{\mathrm{NC1}}(\overline{H})$ simplifies to

$$\mathcal{L}_{\mathrm{NC1}}(\overline{H}) = \frac{1}{2}\operatorname{tr}\left\{W_{\mathrm{LS}}\Sigma_W W_{\mathrm{LS}}^\top\right\}, \tag{17}$$

where $\overline{H} \in \mathbb{R}^{P \times CN}$ has columns $h_{i,c} - \mu_G$. The term $\mathcal{L}_{\mathrm{NC2/3}}(\overline{H})$ also simplifies instructively to

$$\mathcal{L}_{\mathrm{NC2/3}}(\overline{H}) = \frac{1}{2C}\|W_{\mathrm{LS}}\overline{M} - \Phi\|_F^2, \tag{18}$$

where $\Phi \in \mathbb{R}^{C \times C}$ is the standard Simplex ETF:

$$\Phi = I - \frac{1}{C}\mathbb{1}\mathbb{1}^\top.$$

## C.2 Additional intuitions for Theorem 2 terms

The expressions in Equation 15 and Equation 17 further evoke the intuition that

> $\mathcal{L}_{\mathrm{NC1}}(\overline{H})$ *is a variance term that goes to zero only under activation collapse* **(NC1)**
.

More specifically, we see that, *with the class-means held constant*, the only way to have $\mathcal{L}_{\mathrm{NC1}}(\overline{H}) \to 0$ is for $\Sigma_W \to 0$. Similarly, by examining Equation 18, we see that

> $\mathcal{L}_{\mathrm{NC2/3}}(\overline{H})$ quantifies deviations of $W_{\mathrm{LS}}\overline{M}$—i.e. the matrix of class-mean predictions—from the standard Simplex ETF.

Under **(NC2)** and **(NC3)**, both $W_{\mathrm{LS}}$ and $\overline{M}$ tend to jointly aligned ETF's, possibly in an alternate pose; so $W_{\mathrm{LS}} \to \Phi U^T$ and $\overline{M} \to U\Phi$ for a partial orthogonal matrix $U$ satisfying $U^T U = I_C$. Since $\Phi^2 = \Phi$, **(NC2)** and **(NC3)** together demand $W_{\mathrm{LS}}\overline{M} \to \Phi$. Thus, $\mathcal{L}_{\mathrm{NC2/3}}(\overline{H})$ can be interpreted as a semi-metric reflecting the distance from achieving **(NC2)** and **(NC3)**.

## D Theorem 3

### D.1 Implications of invariance

In this section, we will discuss in more detail how the invariance observed Section 3.1 leads the the continually renormalized gradient flow (Equation 5). In particular, we have seen in Section 3.1 of the main text that, on the central path, both the predictions $W_{\mathrm{LS}}(\overline{H})\overline{H}$ and the MSE loss

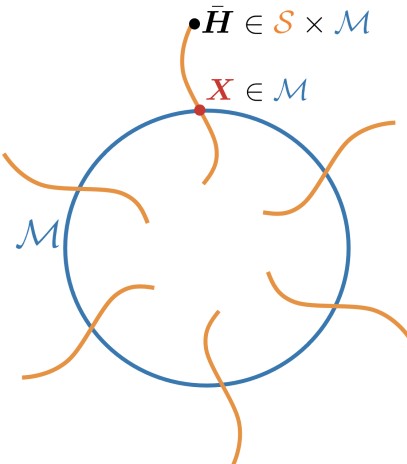

Figure 13: *Fiber bundle.* Any full-rank features matrix, $\overline{H}$, has a representative element, $X$, on $\mathcal{X}$. For any $X \in \mathcal{X}$, a *fiber* is the set $\mathcal{S} \times X$ i.e. $\{AX : A \in \mathcal{S}\}$ (the Minkowski set product), where $\mathcal{S}$ is the set of symmetric positive-definite matrices. Features on the same fiber generate the *same* class predictions and the *same* MSE loss. The optimization described in Sections D.1-D.3 moves *fiber-to-fiber*.

$\mathcal{L}(W_{\mathrm{LS}}(\overline{H}), \overline{M}(\overline{H}))$ are invariant under the transformation $\overline{H} \mapsto A\overline{H}$ when $A$ is a invertible matrix. Since our interest mainly lies with $A = \Sigma_W^{-\frac{1}{2}}$, we will restrict the intuitive discussion in this section to positive-definite $A$.

Now, consider the set $\mathcal{H}$ of features $\overline{H}$ with with non-singular within-class covariances. We view $\mathcal{H}$ as a fiber bundle[17], where—on each fiber—live (apparently) different activations, $\overline{H}$, that (in fact) generate the exact *same* class predictions and the exact *same* MSE loss (see Figure 13). The base space—we will denote it $\mathcal{X}$—can be taken to be the collection of $X \in \mathcal{H}$ where $\Sigma_W(X) = I$. Every $\overline{H} \in \mathcal{H}$ is representable as $\overline{H} = AX$ where $A = \Sigma_W^{-\frac{1}{2}}(\overline{H})$ and $\Sigma_W(X) = I$. The activation matrices $X$ in the base space might be variously called "pre-whitened," "normalized," "standardized," or "sphered." We shall call $\mathcal{X}$ a *normalized features manifold*[18].

Invoking invariance, we can make the following observations about our optimization task:

1. If, at a specific $t$, $H_t$ happens to be a normalized set of activations (i.e. $H_t \in \mathcal{X}$ is currently located in the base space $\mathcal{X}$), there is no performance benefit to leaving the base space. There is also *no performance benefit* to moving *along* a fiber; only by moving from *fiber-to-fiber* can we improve performance.

2. In some sense, we waste time *except* when jumping from fiber-to-fiber; we might prefer to stay in the base space all the time by forcing the dynamics to jump from fiber-to-fiber.

On the other hand, we started with the viewpoint of studying gradient flow of the original $H$. So, we consider the following natural model for the application of gradient descent in $X$:

1. For a given initial feature activations, $\overline{H}_0 \in \mathcal{H}$, we renormalize—obtaining a starting point $X_0 \in \mathcal{X}$; here $X_0 = \Sigma_W^{-\frac{1}{2}}(\overline{H}_0)\overline{H}_0$. Class predictions and MSE loss performance for the MSE-optimal classifier do not change from this renormalization.

2. We compute the usual gradient of MSE loss, at $X_0$, obtaining a step $\Delta X_0 = -\eta \nabla_X \mathcal{L}_{\mathrm{LS}}(X_0)$, where $\eta$ is a step size.

---

[17]An even larger space could be considered, where $\Sigma_W(\overline{H})$ is not necessarily full rank, and then other components of this space having covariances of rank $C$, $C - 1$,...,1 could be considered. We just discuss the full-rank case here.

[18]This is sometimes called the *Generalized Stiefel Manifold* and will be defined more formally in Section D.2.

3. We take a step, going to
$$\overline{\boldsymbol{H}}_1 = \boldsymbol{X}_0 + \Delta \boldsymbol{X}_0.$$

4. $\overline{\boldsymbol{H}}_1$ will not necessarily be normalized. We map back along whatever fiber we have landed upon, obtaining a corresponding point in the base space $\mathcal{X}$, call this $\boldsymbol{X}_1$. Here,
$$\boldsymbol{X}_1 = \boldsymbol{\Sigma}_W^{-\frac{1}{2}}(\overline{\boldsymbol{H}}_1)\overline{\boldsymbol{H}}_1.$$
Class predictions and MSE loss performance do not change from this renormalization.

5. Repeat steps 2, 3, 4 at $\boldsymbol{X}_1$, obtaining thereby $\boldsymbol{X}_2$; and so on.

This process might be described as *gradient descent with continual renormalization*:

1. Renormalize the initial features activation matrix;

2. Compute the ordinary gradient of MSE loss and take the gradient descent step;

3. Renormalize again after each such step; and

4. Repeat steps 2 and 3.

While the continual renormalization process differs from usual gradient flow, it is both intuitively understandable and sensible. See discussion of assumption **(A3)** in Section 3

## D.2 ALIGNED SNR COORDINATES

Analysis of continually renormalized gradient flow can be simplified, without loss of generality, by applying a change of basis in both row and column space:

**Definition 3** (**Transformation into Aligned SNR Coordinates**). *Consider the SVD decomposition of the SNR from Definition 2 outputting left singular vectors $\boldsymbol{U} \in \mathbb{R}^{P \times C}$ and right singular vectors $\boldsymbol{V} \in \mathbb{R}^{C \times C}$. For any matrix $\boldsymbol{Z} \in \mathbb{R}^{P \times C}$, define the transformation $\boldsymbol{Z} \to \boldsymbol{U}^\top \boldsymbol{Z} \boldsymbol{V}$ as the **transformation into aligned-SNR coordinates**.*

The aligned-SNR coordinates is so-named because it diagonalizes i.e. aligns the SNR matrix:

**Definition 4** (**Aligned SNR Matrix**). *Consider the SVD decomposition of the SNR from Definition 2. Combined with Equation 8, observe that*
$$\boldsymbol{\Omega} = \mathrm{diag}\left(\{\omega_c\}_{c=1}^{C-1}, 0\right) = \boldsymbol{U}^\top \boldsymbol{\Sigma}_W^{-\frac{1}{2}} \overline{\boldsymbol{M}} \boldsymbol{V}$$
*We call $\boldsymbol{\Omega}$ the **aligned SNR matrix**.*

The following facts about the aligned SNR matrix becomes useful later in our derivations:

**Observation 1** (**SVD of Aligned SNR Matrix**). *The SVD of the aligned SNR matrix itself is simply*
$$\boldsymbol{\Omega} = \sum_{j=1}^{C-1} \omega_j \boldsymbol{e}_j \boldsymbol{e}_j^\top,$$
*with the canonical basis $\{\boldsymbol{e}_j\}_{j=1}^{C}$ as its singular vectors.*

As discussed in Section 3.2, the $\{\omega_c\}_{c=1}^{C-1}$ that comprise $\boldsymbol{\Omega}$ are decisive for understanding separation performance. In fact, when $\lambda=0$, *the MSE loss can be entirely characterized by the SNR singular values on the central path*:

**Lemma 1** (**Spectral Representation of MSE Loss**). *When $\lambda = 0$, MSE loss obeys*
$$\mathcal{L}(\boldsymbol{W}_{LS}(\overline{\boldsymbol{H}}), \overline{\boldsymbol{H}}) = \mathcal{L}_{LS}(\boldsymbol{W}_{LS}(\overline{\boldsymbol{H}}), \overline{\boldsymbol{H}}) = \frac{1}{2} \sum_{j=1}^{C-1} \frac{1}{\omega_j^2 + C} = \mathcal{L}\left(\{\omega_j\}_{j=1}^{C-1}\right),$$
*on the central path, and its decomposition obeys*
$$\mathcal{L}_{NC1}(\overline{\boldsymbol{H}}) = \frac{1}{2} \sum_{j=1}^{C-1} \frac{\omega_j^2}{(C + \omega_j^2)^2} = \mathcal{L}_{NC1}\left(\{\omega_j\}_{j=1}^{C-1}\right)$$
$$\mathcal{L}_{NC2/3}(\overline{\boldsymbol{H}}) = \frac{1}{2} \sum_{j=1}^{C-1} \frac{1}{C} \left(\frac{\omega_j^2}{\omega_j^2 + C} - 1\right)^2 = \mathcal{L}_{NC2/3}\left(\{\omega_j\}_{j=1}^{C-1}\right).$$

*Proof.* Using Equation 16, Equation 17, and Equation 18, the loss on the central path equals

$$\mathcal{L}(\boldsymbol{W}_{\text{Ls}}, \overline{\boldsymbol{M}}) = \frac{1}{2C} \|C^{-1} \overline{\boldsymbol{M}}^\top \boldsymbol{\Sigma}_T^{-1} \overline{\boldsymbol{M}} - \boldsymbol{\Phi}\|_F^2 + \frac{1}{2} \operatorname{tr}\{C^{-1} \overline{\boldsymbol{M}}^\top \boldsymbol{\Sigma}_T^{-1} \boldsymbol{\Sigma}_W \boldsymbol{\Sigma}_T^{-1} \overline{\boldsymbol{M}} C^{-1}\}. \quad (19)$$

Observe $\overline{\boldsymbol{M}}^\top \boldsymbol{\Sigma}_T^{-1} \overline{\boldsymbol{M}}$ and $\boldsymbol{\Phi}$ are simultaneously diagonalizable since they share the same $[1, \ldots, 1]^\top$ null space and $\boldsymbol{\Phi}$ has $C - 1$ equal eigenvalues. Also, observe that

$$\boldsymbol{\Sigma}_T = \boldsymbol{\Sigma}_W + \frac{1}{C} \overline{\boldsymbol{M}}\, \overline{\boldsymbol{M}}^T.$$

Let $\widehat{\boldsymbol{\Omega}}$ be submatrix of the aligned-SNR matrix $\boldsymbol{\Omega}$ corresponding to the non-zero singular values i.e. $\widehat{\boldsymbol{\Omega}} = \operatorname{diag}\left(\{\omega_j\}_{c=1}^{C-1}\right)$. After taking simultaneous-diagonalizations and canceling partial orthogonal matrices, Equation 19 can be written solely in terms of $\widehat{\boldsymbol{\Omega}}$:

$$\mathcal{L}(\widehat{\boldsymbol{\Omega}}) = \frac{1}{2C} \left\| \widehat{\boldsymbol{\Omega}}^\top \left(\widehat{\boldsymbol{\Omega}}\widehat{\boldsymbol{\Omega}}^\top + C\boldsymbol{I}_{C-1}\right)^{-1} \widehat{\boldsymbol{\Omega}} - \boldsymbol{I}_{C-1} \right\|_F^2 + \frac{1}{2} \operatorname{tr}\left\{\widehat{\boldsymbol{\Omega}}^\top (\widehat{\boldsymbol{\Omega}}\widehat{\boldsymbol{\Omega}}^\top + C\boldsymbol{I}_{C-1})^{-2}\widehat{\boldsymbol{\Omega}}\right\}.$$

This can be further simplified into

$$\mathcal{L}(\{\omega_j\}_{c=1}^{C-1}) = \frac{1}{2} \sum_{j=1}^{C-1} \frac{1}{C} \left(\frac{\omega_j^2}{\omega_j^2 + C} - 1\right)^2 + \frac{\omega_j^2}{(C + \omega_j^2)^2}$$

$$= \frac{1}{2} \sum_{j=1}^{C-1} \frac{1}{\omega_j^2 + C}. \quad (20)$$

This completes the proof for Lemma 1. $\qquad \square$

Thus, the aligned SNR matrix, $\boldsymbol{U}^\top \boldsymbol{\Sigma}_W^{-\frac{1}{2}} \overline{\boldsymbol{M}} \boldsymbol{V}$, maximally simplifies analysis by reducing the vanilla SNR matrix to a diagonal matrix (Definition 4) whose entries determine the MSE loss (Lemma 1).

We introduce the alignment of features into their own notion of aligned SNR coordinates:

**Definition 5** (**Renormalization to aligned SNR Coordinates**). *Consider $\overline{\boldsymbol{H}} \in \mathbb{R}^{P \times CN}$ with corresponding SNR left and right singular vectors $\boldsymbol{U}$ and $\boldsymbol{V}$ (Definition 2). Then, define the SNR-aligned renormalization of the features as*

$$\boldsymbol{X} = \boldsymbol{U}^\top \left(\boldsymbol{\Sigma}_W\right)^{-\frac{1}{2}} \overline{\boldsymbol{H}}(\boldsymbol{V} \otimes \boldsymbol{I}_N) = \boldsymbol{U}^\top \left(\overline{\boldsymbol{H}} C \overline{\boldsymbol{H}}^\top\right)^{-\frac{1}{2}} \overline{\boldsymbol{H}}(\boldsymbol{V} \otimes \boldsymbol{I}_N) \in \mathbb{R}^{C \times NC},$$

*where $\otimes$ is the Kronecker product and $\boldsymbol{C}$ is a class-centering matrix defined as*

$$\boldsymbol{C} = \frac{1}{CN}\left(\boldsymbol{I}_{CN} - \frac{1}{N}\boldsymbol{Y}^\top \boldsymbol{Y}\right). \quad (21)$$

As discussed in Section 3.1, the transformation $\overline{\boldsymbol{H}} \to \boldsymbol{\Sigma}_W^{-\frac{1}{2}} \overline{\boldsymbol{H}}$ does not change the predictions of the least squares classifier. Hence, it does not change the MSE loss. It is easy to check that this still holds for $\boldsymbol{X}$ i.e. invariance is preserved by the change into SNR-aligned coordinates.

It is also easy to check that the sphericity of the within-class covariance is preserved as well:

**Observation 2** (**Sphericity of Within-Class Covariance of SNR Coordinates**).

$$\boldsymbol{X} \boldsymbol{C} \boldsymbol{X}^\top = \boldsymbol{I}_C$$

Finally, routine applications of Definitions and canceling out of (partial) orthogonal matrix multiplications in aligned SNR coordinates lead to the following simplified representation for $\boldsymbol{\Omega}$ and its differential:

**Observation 3** (**Relation Between SNR Coordinates and SNR Matrix**). *The aligned SNR matrix, $\boldsymbol{\Omega}$ (Definition 4) can be expressed as a function of the features in aligned SNR-coordinates $\boldsymbol{X}$ (Definition 5) as simply*

$$\boldsymbol{\Omega}(\boldsymbol{X}) = \frac{1}{N} \boldsymbol{X} \boldsymbol{Y}^\top.$$

*Moreover, let* $\mathrm{d}\boldsymbol{\Omega}_{ij}(\cdot): T_{\boldsymbol{X}}\mathcal{X} \to \mathbb{R}$ *be the differential 1-form*[19] *associated with the* $ij$-*th entry of* $\boldsymbol{\Omega}$. *Define* $\mathrm{d}\boldsymbol{\Omega}(\cdot): T_{\boldsymbol{X}}\mathcal{X} \to \mathbb{R}^{C \times C}$ *such that, for some* $\boldsymbol{Z} \in T_{\boldsymbol{X}}\mathcal{X}$, *each entry matrix of the matrix* $\mathrm{d}\boldsymbol{\Omega}(\boldsymbol{Z})$ *is* $\mathrm{d}\boldsymbol{\Omega}_{i,j}(\boldsymbol{Z})$. *Then, it follows that*

$$\mathrm{d}\boldsymbol{\Omega}(\boldsymbol{Z}) = \frac{1}{N}\boldsymbol{Z}\boldsymbol{Y}^{\top}, \quad \forall \boldsymbol{Z} \in T_{\boldsymbol{X}}\mathcal{X}.$$

From a geometric perspective, the transformation in Definition 5 maps the features, $\boldsymbol{H}$, to $\boldsymbol{X}$ belonging to a *Normalized Features Manifold*:

**Definition 6 (Normalized Features Manifold).**

$$\mathcal{X} = \{\boldsymbol{X} \in \mathbb{R}^{C \times CN} \mid \boldsymbol{X}\boldsymbol{C}\boldsymbol{X}^{\top} = \boldsymbol{I}_C\}$$

To understand the dynamics of the singular values of $\boldsymbol{\Omega}$—which is connected to the dynamics of the features through Lemma 1—we will analyze the gradient flow *on this manifold*. To this end, we first identify its tangent space at a particular $\boldsymbol{X}$ as well as the projection operator onto that tangent space.

**Proposition 3 (Tangent Space of Normalized Features Manifold).**

$$T_{\boldsymbol{X}}\mathcal{X} = \{\boldsymbol{Z} \in \mathbb{R}^{C \times CN} \mid \boldsymbol{X}\boldsymbol{C}\boldsymbol{Z}^{\top} + \boldsymbol{Z}\boldsymbol{C}\boldsymbol{X} = \boldsymbol{0}\}$$

**Proposition 4 (Projection Onto Tangent Space of Normalized Features Manifold).**

$$\Pi_{T_{\boldsymbol{X}}\mathcal{X}}(\boldsymbol{Z}) = \boldsymbol{Z} - \frac{1}{2}(\boldsymbol{X}\boldsymbol{C}\boldsymbol{Z}^{\top} + \boldsymbol{Z}\boldsymbol{C}\boldsymbol{X}^{\top})\boldsymbol{X}$$

### D.3 CONTINUALLY RENORMALIZED GRADIENT FLOW

In fact, gradient flow on the normalized features manifold (Definition 6) is the continuous analogue of the intuitive, discrete algorithm in Subsection D.1. In particular, the intuitive algorithm consists of a gradient step, $\overline{\boldsymbol{H}}_1 = \boldsymbol{X}_0 + \Delta\boldsymbol{X}_0$ followed by a mapping to the base space of the fiber bundle, $\boldsymbol{X}_1 = \boldsymbol{\Sigma}(\overline{\boldsymbol{H}}_1)^{-\frac{1}{2}}\overline{\boldsymbol{H}}_1$. For small $\Delta\boldsymbol{X}_0$, these two steps are in fact equivalent (up to a negligible term) to taking a $T_{\boldsymbol{X}}\mathcal{X}$-projected gradient step on the manifold $\mathcal{X}$ as proven below:

**Lemma 2.** *Assuming* $\boldsymbol{X}_0 \in \mathcal{X}$, *a renormalized gradient step,*

$$\boldsymbol{X}_1 = \boldsymbol{\Sigma}_W^{-\frac{1}{2}}(\overline{\boldsymbol{H}}_1)\overline{\boldsymbol{H}}_1 = \boldsymbol{\Sigma}_W^{-\frac{1}{2}}(\boldsymbol{X}_0 + \Delta\boldsymbol{X}_0) \cdot (\boldsymbol{X}_0 + \Delta\boldsymbol{X}_0),$$

*is equivalent, up to an* $O\left(\|\Delta\boldsymbol{X}_0\|^2\right)$ *term, to a* $T_{\boldsymbol{X}}\mathcal{X}$-*projected gradient step, i.e.*

$$\boldsymbol{X}_1 = \boldsymbol{X}_0 + \Pi_{T_{\boldsymbol{X}}\mathcal{X}}(\Delta\boldsymbol{X}_0) + O\left(\|\Delta\boldsymbol{X}_0\|^2\right).$$

*Proof.* Notice that

$$\begin{aligned}
\boldsymbol{X}_1 &= \boldsymbol{\Sigma}_W^{-\frac{1}{2}}(\overline{\boldsymbol{H}}_1)\overline{\boldsymbol{H}}_1 \\
&= \left((\boldsymbol{X}_0 + \Delta\boldsymbol{X}_0)\boldsymbol{C}(\boldsymbol{X}_0 + \Delta\boldsymbol{X}_0)^{\top}\right)^{-\frac{1}{2}}(\boldsymbol{X}_0 + \Delta\boldsymbol{X}_0) \\
&= \left(\boldsymbol{I} + \Delta\boldsymbol{X}_0\boldsymbol{C}\boldsymbol{X}_0^{\top} + \boldsymbol{X}_0\boldsymbol{C}\Delta\boldsymbol{X}_0^{\top} + \Delta\boldsymbol{X}_0\boldsymbol{C}\Delta\boldsymbol{X}_0^{T}\right)^{-\frac{1}{2}}(\boldsymbol{X}_0 + \Delta\boldsymbol{X}_0),
\end{aligned}$$

where in the last step we used our assumption that $\boldsymbol{X}_0 \in \mathcal{X}$, i.e., $\boldsymbol{X}_0\boldsymbol{C}\boldsymbol{X}_0 = \boldsymbol{I}$. By Taylor's Theorem,

$$(\boldsymbol{I} + \boldsymbol{A})^{-\frac{1}{2}} = \boldsymbol{I} - \frac{1}{2}\boldsymbol{A} + O(\|\boldsymbol{A}\|^2).$$

Therefore, we get

$$\begin{aligned}
\boldsymbol{X}_1 &= \left(\boldsymbol{I} - \frac{1}{2}\left(\Delta\boldsymbol{X}_0\boldsymbol{C}\boldsymbol{X}_0^{\top} + \boldsymbol{X}_0\boldsymbol{C}\Delta\boldsymbol{X}_0^{\top} + \Delta\boldsymbol{X}_0\boldsymbol{C}\Delta\boldsymbol{X}_0^{T}\right) + O\left(\|\Delta\boldsymbol{X}_0\|^2\right)\right)(\boldsymbol{X}_0 + \Delta\boldsymbol{X}_0) \\
&= \boldsymbol{X}_0 + \Delta\boldsymbol{X}_0 - \frac{1}{2}\left(\Delta\boldsymbol{X}_0\boldsymbol{C}\boldsymbol{X}_0^{\top} - \boldsymbol{X}_0\boldsymbol{C}\Delta\boldsymbol{X}_0^{\top}\right)\boldsymbol{X}_0 + O\left(\|\Delta\boldsymbol{X}_0\|^2\right) \\
&= \boldsymbol{X}_0 + \Pi_{T_{\boldsymbol{X}}\mathcal{X}}(\Delta\boldsymbol{X}_0) + O\left(\|\Delta\boldsymbol{X}_0\|^2\right),
\end{aligned}$$

where the last step follows from Proposition 4. $\qquad\square$

As we transition from a discrete gradient descent to a continuous gradient flow, the step size $\eta \to 0$, and the residual $O\left(\|\Delta \boldsymbol{X}_0\|^2\right)$ in the above lemma becomes negligible, since $\Delta \boldsymbol{X}_0 = -\eta \nabla_{\boldsymbol{X}} \mathcal{L}_{\mathrm{LS}}(\boldsymbol{X}_0)$. This motivates us to study, in the next section, the continually renormalized gradient flow (as defined in Equation 5 of the main text) of $\boldsymbol{X}$ on $\mathcal{X}$:

$$\frac{\mathrm{d}}{\mathrm{d}t} \boldsymbol{X} = -\Pi_{T_{\boldsymbol{X}}\mathcal{X}} \left(\nabla_{\boldsymbol{X}} \mathcal{L}_{\mathrm{LS}}(\boldsymbol{X})\right).$$

### D.4 PROOF OF PROPOSITION 2 (GRADIENT FLOW IN ALIGNED SNR COORDINATES)

To prove Proposition 2, we first set up some auxiliary notation and lemmas. Denote the $j$-th row (transposed) of $\boldsymbol{X}$ and $\boldsymbol{Y}$, respectively, by

$$\boldsymbol{x}_j = \boldsymbol{X}^\top \boldsymbol{e}_j \tag{22}$$

and

$$\boldsymbol{y}_j = \boldsymbol{Y}^\top \boldsymbol{e}_j.$$

Recall that—when differentiating SVDs—the derivative of the $j$-th singular value, $\omega_j$, only depends on the $j$-th singular subspace (see Equation 17 of Townsend (2016)). As a consequence, we will see in the following lemma that, for any differential $\mathrm{d}\boldsymbol{X}$, the corresponding $\mathrm{d}\omega_j$ only depends on $\mathrm{d}\boldsymbol{x}_j$.

**Lemma 3 (Derivative of Singular Values With Respect to SNR Coordinates).** *Given the differential 1-form* $\mathrm{d}\omega_j\left(\cdot\right) : T_{\boldsymbol{X}}\mathcal{X} \to \mathbb{R}$, *the following holds for all*[19] $\mathrm{d}\boldsymbol{X} \in \mathbb{R}^{C \times CN}$:

$$\mathrm{d}\omega_j\left(\Pi_{T_{\boldsymbol{X}}\mathcal{X}}(\mathrm{d}\boldsymbol{X})\right) = \left(\frac{1}{N}\boldsymbol{y}_j^\top - \omega_j \boldsymbol{x}_j^\top \boldsymbol{C}\right) \mathrm{d}\boldsymbol{x}_j, \quad \forall j = 1, \ldots, C.$$

*where (consistent with Definition 2) we adopt the convention that* $\omega_C = 0$ *is the $C$-th singular value of the SNR matrix.*

*Proof.* Without loss of generality, assume that $\boldsymbol{X}$ is represented in aligned SNR coordinates (Section D.2). Using Observation 1 and the differential of the singular value decomposition (see Equation 17 of Townsend (2016)):

$$\mathrm{d}\omega_j\left(\Pi_{T_{\boldsymbol{X}}\mathcal{X}}(\mathrm{d}\boldsymbol{X})\right) = \boldsymbol{e}_j^\top \mathrm{d}\boldsymbol{\Omega}\left(\Pi_{T_{\boldsymbol{X}}\mathcal{X}}(\mathrm{d}\boldsymbol{X})\right) \boldsymbol{e}_j, \tag{23}$$

where $\mathrm{d}\boldsymbol{\Omega}\left(\cdot\right)$ is defined as in Observation 3. Next, Observation 3 implies

$$\mathrm{d}\boldsymbol{\Omega}\left(\Pi_{T_{\boldsymbol{X}}\mathcal{X}}(\mathrm{d}\boldsymbol{X})\right) = \frac{1}{N}\Pi_{T_{\boldsymbol{X}}\mathcal{X}}(\mathrm{d}\boldsymbol{X}) \boldsymbol{Y}^\top. \tag{24}$$

Using the projection onto the tangent space, given in Proposition 4 above,

$$\Pi_{T_{\boldsymbol{X}}\mathcal{X}}(\mathrm{d}\boldsymbol{X}) = \mathrm{d}\boldsymbol{X} - \frac{1}{2}(\mathrm{d}\boldsymbol{X}\,\boldsymbol{C}\boldsymbol{X}^\top + \boldsymbol{X}\boldsymbol{C}\,\mathrm{d}\boldsymbol{X}^\top)\boldsymbol{X}. \tag{25}$$

Combining Equation 23, Equation 24, and Equation 25, we obtain:

$$\begin{aligned}
\mathrm{d}\omega_j\left(\Pi_{T_{\boldsymbol{X}}\mathcal{X}}(\mathrm{d}\boldsymbol{X})\right) &= \boldsymbol{e}_j^\top \mathrm{d}\boldsymbol{\Omega}\left(\Pi_{T_{\boldsymbol{X}}\mathcal{X}}(\mathrm{d}\boldsymbol{X})\right) \boldsymbol{e}_j \\
&= \frac{1}{N}\boldsymbol{e}_j^\top \Pi_{T_{\boldsymbol{X}}\mathcal{X}}(\mathrm{d}\boldsymbol{X})\, \boldsymbol{Y}^\top \boldsymbol{e}_j \\
&= \frac{1}{N}\boldsymbol{e}_j^\top \left(\mathrm{d}\boldsymbol{X} - \frac{1}{2}\left(\mathrm{d}\boldsymbol{X}\,\boldsymbol{C}\boldsymbol{X}^\top + \boldsymbol{X}\boldsymbol{C}\,\mathrm{d}\boldsymbol{X}^\top\right)\boldsymbol{X}\right) \boldsymbol{Y}^\top \boldsymbol{e}_j.
\end{aligned} \tag{26}$$

---

[19]Notation: For real-valued functions $f$, we use $\mathrm{d}f\left(\cdot\right) : T_{\boldsymbol{X}}\mathcal{X} \to \mathbb{R}$ to denote the associated differential 1-form i.e. the function outputting the directional derivative of $f$ in the direction of the argument; For matrices $\boldsymbol{Z} \in \mathbb{R}^{m \times n}$, we use $\mathrm{d}\boldsymbol{Z} \in \mathbb{R}^{m \times n}$—without succeeding parenthesis—to denote the matrix differential (cf. Townsend (2016)). These notions are equivalent: $\mathrm{d}f\left(\cdot\right)$ can be represented as a vector $\mathrm{d}f$ in the dual-space of $T_{\boldsymbol{X}}\mathcal{X}$; When $\boldsymbol{Z}$ is a function of some vector $\boldsymbol{v}$, $\mathrm{d}\boldsymbol{Z}$ can be represented as a collection of 1-forms by defining $\mathrm{d}\boldsymbol{Z}_{ij}\left(\cdot\right) = \left\langle \frac{\mathrm{d}\boldsymbol{Z}_{ij}}{\mathrm{d}\boldsymbol{v}}, \cdot \right\rangle$ for each entry $\boldsymbol{Z}_{ij}$.

Moreover, using Observation 1 and 3, we can simplify these expressions into

$$\frac{1}{N}\boldsymbol{e}_j^\top \,\mathrm{d}\boldsymbol{X}\,\boldsymbol{Y}^\top \boldsymbol{e}_j = \frac{1}{N}\boldsymbol{y}_j^\top \,\mathrm{d}\boldsymbol{x}_j$$

$$\frac{1}{N}\boldsymbol{e}_j^\top \,\mathrm{d}\boldsymbol{X}\,\boldsymbol{C}\boldsymbol{X}^\top \boldsymbol{X}\boldsymbol{Y}^\top \boldsymbol{e}_j = \omega_j \boldsymbol{e}_j^\top \,\mathrm{d}\boldsymbol{X}\,\boldsymbol{C}\boldsymbol{X}^\top \boldsymbol{e}_j = \omega_j \boldsymbol{x}_j^\top \boldsymbol{C}\,\mathrm{d}\boldsymbol{x}_j$$

$$\frac{1}{N}\boldsymbol{e}_j^\top \boldsymbol{X}\boldsymbol{C}\,\mathrm{d}\boldsymbol{X}^\top \,\boldsymbol{X}\boldsymbol{Y}^\top \boldsymbol{e}_j = \omega_j \boldsymbol{e}_j^\top \boldsymbol{X}\boldsymbol{C}\,\mathrm{d}\boldsymbol{X}^\top \,\boldsymbol{e}_j = \omega_j \boldsymbol{x}_j^\top \boldsymbol{C}\,\mathrm{d}\boldsymbol{x}_j \,.$$

Finally, substituting the above three expressions into Equation 26, we get:

$$\mathrm{d}\omega_j\left(\Pi_{T_{\boldsymbol{X}}\mathcal{X}}(\mathrm{d}\boldsymbol{X})\right) = \left(\frac{1}{N}\boldsymbol{y}_j^\top - \omega_j \boldsymbol{x}_j^\top \boldsymbol{C}\right)\mathrm{d}\boldsymbol{x}_j \,,$$

which proves the claim. □

Using Lemma 3, we can now obtain the dynamics of $\boldsymbol{x}_j$ under gradient flow:

**Lemma 4.** *Under continually renormalized gradient flow (Equation 5),*

$$\frac{\mathrm{d}}{\mathrm{d}t}\boldsymbol{x}_j = \frac{\omega_j}{(C+\omega_j^2)^2}\left(\frac{1}{N}\boldsymbol{y}_j - \omega_j \boldsymbol{C}\boldsymbol{x}_j\right), \quad \forall j = 1, \ldots, C,$$

*where (consistent with Definition 2) we adopt the convention that $\omega_C = 0$ is the $C$-th singular value of the SNR matrix.*

*Proof.* Recall $\boldsymbol{x}_j \in \mathbb{R}^{CN}$ is the $j$-th row of $\boldsymbol{X} \in \mathbb{R}^{C \times CN}$, which we assume to be in aligned SNR coordinates without loss of generality (Section D.2). Applying the flow definition in Equation 5 to the $i$-th element of $\boldsymbol{x}_j$ gives

$$\begin{aligned}
\frac{\mathrm{d}}{\mathrm{d}t}x_{ji} &= -\boldsymbol{e}_j^\top \Pi_{T_{\boldsymbol{X}}\mathcal{X}}\left(\frac{\mathrm{d}\mathcal{L}_{\mathrm{LS}}}{\mathrm{d}\boldsymbol{X}}\right)\boldsymbol{e}_i \\
&= -\left\langle \Pi_{T_{\boldsymbol{X}}\mathcal{X}}\left(\frac{\mathrm{d}\mathcal{L}_{\mathrm{LS}}}{\mathrm{d}\boldsymbol{X}}\right), \boldsymbol{e}_j \boldsymbol{e}_i^\top \right\rangle_F \\
&= -\left\langle \frac{\mathrm{d}\mathcal{L}_{\mathrm{LS}}}{\mathrm{d}\boldsymbol{X}}, \Pi_{T_{\boldsymbol{X}}\mathcal{X}}(\boldsymbol{e}_j \boldsymbol{e}_i^\top) \right\rangle_F,
\end{aligned}$$

where $\frac{\mathrm{d}\mathcal{L}_{\mathrm{LS}}}{\mathrm{d}\boldsymbol{X}} \in \mathbb{R}^{C \times CN}$ is the standard derivative within the ambient $\mathbb{R}^{C \times CN}$-space, $\boldsymbol{e}_i \in \mathbb{R}^{CN}$ and $\boldsymbol{e}_j \in \mathbb{R}^C$ are the canonical basis vectors, and $\langle \cdot, \cdot \rangle_F$ is the Frobenius matrix inner-product. Next, observe that the differential 1-form $\mathrm{d}\mathcal{L}_{\mathrm{LS}}(\cdot) : T_{\boldsymbol{X}}\mathcal{X} \to \mathbb{R}$ satisfies the following:

$$\mathrm{d}\mathcal{L}_{\mathrm{LS}}(\boldsymbol{Z}) = \left\langle \frac{\mathrm{d}\mathcal{L}_{\mathrm{LS}}}{\mathrm{d}\boldsymbol{X}}, \boldsymbol{Z}\right\rangle_F, \quad \forall \boldsymbol{Z} \in T_{\boldsymbol{X}}\mathcal{X}.$$

Taking $\boldsymbol{Z} = \Pi_{T_{\boldsymbol{X}}\mathcal{X}}(\boldsymbol{e}_j \boldsymbol{e}_i^\top)$ then leads to

$$\begin{aligned}
\frac{\mathrm{d}}{\mathrm{d}t}x_{ji} &= -\mathrm{d}\mathcal{L}_{\mathrm{LS}}\left(\Pi_{T_{\boldsymbol{X}}\mathcal{X}}(\boldsymbol{e}_j \boldsymbol{e}_i^\top)\right) \\
&= -\sum_{k=1}^{C-1}\frac{\mathrm{d}\mathcal{L}_{\mathrm{LS}}}{\mathrm{d}\omega_k}\,\mathrm{d}\omega_k\left(\Pi_{T_{\boldsymbol{X}}\mathcal{X}}(\boldsymbol{e}_j \boldsymbol{e}_i^\top)\right) \\
&= -\frac{\mathrm{d}\mathcal{L}_{\mathrm{LS}}}{\mathrm{d}\omega_j}\,\mathrm{d}\omega_j\left(\Pi_{T_{\boldsymbol{X}}\mathcal{X}}(\boldsymbol{e}_j \boldsymbol{e}_i^\top)\right)
\end{aligned}$$

where the second step follows by the chain rule, and the last step follows from Lemma 3 in that $\mathrm{d}\omega_k\left(\Pi_{T_{\boldsymbol{X}}\mathcal{X}}(\boldsymbol{e}_j \boldsymbol{e}_i^\top)\right) = 0$ if $k \neq j$. Moreover, Lemma 3 also gives

$$\mathrm{d}\omega_j\left(\Pi_{T_{\boldsymbol{X}}\mathcal{X}}(\boldsymbol{e}_j \boldsymbol{e}_i^\top)\right) = \left(\frac{1}{N}\boldsymbol{y}_j^\top - \omega_j \boldsymbol{x}_j^\top \boldsymbol{C}\right)\boldsymbol{e}_i.$$

It then follows that

$$\frac{\mathrm{d}}{\mathrm{d}t}\boldsymbol{x}_j = -\frac{\mathrm{d}\mathcal{L}_{\mathrm{LS}}}{\mathrm{d}\omega_j}\left(\frac{1}{N}\boldsymbol{y}_j - \omega_j \boldsymbol{C}\boldsymbol{x}_j\right)$$
$$= \frac{\omega_j}{(C+\omega_j^2)^2}\left(\frac{1}{N}\boldsymbol{y}_j - \omega_j \boldsymbol{C}\boldsymbol{x}_j\right),$$

where the last step follows from differentiating the expression in Lemma 1. □

The gradient flows of $\boldsymbol{x}_j$ induce the dynamics of $\omega_j$, which are described below.

**Lemma 5** (**Induced Dynamics on SVD of SNR**). *Under continually renormalized gradient flow (Equation 5), the dynamics for the $j$-th non-zero SNR singular value (Definition 2) is*

$$\frac{\mathrm{d}}{\mathrm{d}t}\omega_j = \frac{1}{N}\frac{\omega_j}{(C+\omega_j^2)^2}, \quad \forall j = 1,\ldots,C-1. \tag{27}$$

*Proof.* Without loss of generality, assume $\boldsymbol{X}$ is represented in aligned SNR coordinates (Section D.2). Observation 1, the differential of the singular value decomposition (see Equation 17 of Townsend (2016)), and Observation 3 collectively imply that for any[20] $\mathrm{d}\boldsymbol{X} \in T_{\boldsymbol{X}}\mathcal{X}$,

$$\mathrm{d}\omega_j\,(\mathrm{d}\boldsymbol{X}) = \boldsymbol{e}_j^\top\,\mathrm{d}\boldsymbol{\Omega}\,(\mathrm{d}\boldsymbol{X})\,\boldsymbol{e}_j = \frac{1}{N}\boldsymbol{e}_j^\top\,\mathrm{d}\boldsymbol{X}\,\boldsymbol{Y}^\top\boldsymbol{e}_j = \frac{1}{N}\boldsymbol{y}_j^\top\,\mathrm{d}\boldsymbol{x}_j\,. \tag{28}$$

Recall that the differential of $\omega_j$ only depends on the differential of $\boldsymbol{x}_j$ (see Lemma 3). Then, we can apply the chain rule:

$$\frac{\mathrm{d}}{\mathrm{d}t}\omega_j = \frac{\mathrm{d}\omega_j}{\mathrm{d}\boldsymbol{x}_j}\frac{\mathrm{d}\boldsymbol{x}_j}{\mathrm{d}t} = \frac{1}{N}\frac{\omega_j}{(C+\omega_j^2)^2}\boldsymbol{y}_j^\top\left(\frac{1}{N}\boldsymbol{y}_j - \omega_j \boldsymbol{C}\boldsymbol{x}_j\right), \tag{29}$$

where the last equality results substituting-in Lemma 4 and Equation 28. Since $\boldsymbol{Y}\boldsymbol{Y}^\top = N\boldsymbol{I}$,

$$\frac{1}{N^2}\boldsymbol{y}_j^\top\boldsymbol{y}_j = \frac{1}{N}.$$

Applying the same relation and Equation 21:

$$\boldsymbol{x}_j^\top \boldsymbol{C}\boldsymbol{y}_j = \boldsymbol{e}_j^\top \boldsymbol{X}\boldsymbol{C}\boldsymbol{Y}^\top\boldsymbol{e}_j = \frac{1}{CN}\boldsymbol{e}_j^\top \boldsymbol{X}\left(\boldsymbol{I} - \frac{1}{N}\boldsymbol{Y}^\top\boldsymbol{Y}\right)\boldsymbol{Y}^\top\boldsymbol{e}_j = 0.$$

Combining all the above equations, we obtain

$$\frac{\mathrm{d}}{\mathrm{d}t}\omega_j = \frac{1}{N}\frac{\omega_j}{(C+\omega_j^2)^2}.$$

□

The closed-form given in Proposition 2 now directly follows.

**Proposition 2** (**Dynamics of Singular Values of SNR Matrix**). *Continually renormalized gradient flow on the central path (Equation 5) induces the following closed-form dynamics on the SNR singular values (Definition 2):*

$$c_1 \log(\omega_j(t)) + c_2\omega_j^2(t) + c_3\omega_j^4(t) = a_j + t, \quad t \geq 0, \quad \text{for all } j = 1,\ldots,C-1. \tag{9}$$

$c_1$, $c_2$, *and* $c_3$ *are positive constants independent of* $j$, *and* $a_j$ *is a constant depending on* $\omega_j(0)$.

*Proof.* Follows from symbolically solving the ODE in Lemma 5 with routine methods. The constants are $c_1 = C^2 N$, $c_2 = CN$, and $c_3 = \frac{N}{4}$. □

---

[20]For our goal of applying the chain rule in Equation 29, we can restrict our attention to just $\mathrm{d}\boldsymbol{X} \in T_{\boldsymbol{X}}\mathcal{X}$: The definition of renormalized gradient flow (Equation 5) ensures that $\frac{\mathrm{d}\boldsymbol{X}}{\mathrm{d}t}$ will always be in $T_{\boldsymbol{X}}\mathcal{X}$.

## D.5 PROOF OF COROLLARY 1

**Corollary 1** (**Properties of SNR Singular Values**). *SNR singular values (Definition 2) following the Equation 9 dynamics satisfy the following limiting behaviors:*

1. $\lim_{t\to\infty} \omega_j(t) = \infty$ *and* $\lim_{t\to\infty} \frac{\omega_j(t)}{\sqrt[4]{t/c_3}} = 1$, *for all* $j = 1, \ldots, C-1$.

2. $\lim_{t\to\infty} \frac{\max_j \omega_j(t)}{\min_j \omega_j(t)} = 1$.

*Proof.* As $t$ tends to infinity, the right-hand side of Equation 9 diverges to infinity and therefore so does the left-hand side (LHS). As the LHS approaches infinity, the logarithmic terms become negligible compared to the dominant quartic term, implying $\omega_j^4(t) \to \infty$. Since $\omega_j(t)$ are singular values, they must be non-negative—implying $\omega_j(t) \to \infty$. Based on the same argument, observe that $\lim_{t\to\infty} \frac{\omega_j(t)}{\sqrt[4]{\frac{t}{c_3}}} = 1$ for all $j$. Since the constant $c_3$ is independent of $j$, it follows that $\lim_{t\to\infty} \frac{\max_j \omega_j(t)}{\min_j \omega_j(t)} = 1$. $\qquad\square$

## D.6 PROOF OF COROLLARY 2

**Lemma 6.** *Under continually renormalized gradient flow (Equation 5), the left and right singular vectors of the SNR matrix remain constant i.e. they are independent of t.*

*Proof.* Without loss of generality, assume $\boldsymbol{X}$ is in represented in aligned SNR coordinates (Section D.2). Recall that, in this coordinate system, the corresponding SNR matrix $\boldsymbol{\Omega} = N^{-1}\boldsymbol{X}\boldsymbol{Y}^\top$ is diagonal, and the left and right singular vectors are simply partial identity matrices.

To show that the singular values of the SNR remain constant, it then suffices to show that $\frac{d\boldsymbol{\Omega}}{dt} = \frac{d\boldsymbol{X}}{dt}\boldsymbol{Y}$ is a diagonal matrix as well[21]. Lemma 4 gives

$$\frac{d}{dt}\boldsymbol{X} = \frac{1}{N}\boldsymbol{D}_1\boldsymbol{Y} - \boldsymbol{D}_2\boldsymbol{X}\boldsymbol{C}$$

where $\boldsymbol{D}_1$ and $\boldsymbol{D}_2$ are $C \times C$ diagonal matrices with the diagonal entries $\left\{\frac{\omega_j}{(C+\omega_j^2)^2}\right\}_{c=1}^C$ and $\left\{\frac{\omega_j^2}{(C+\omega_j^2)^2}\right\}_{c=1}^C$, respectively. Then,

$$\frac{d}{dt}\boldsymbol{\Omega} = \left(\frac{d}{dt}\boldsymbol{X}\right)\boldsymbol{Y}^\top = \frac{1}{N}\boldsymbol{D}_1\boldsymbol{Y}\boldsymbol{Y}^\top - \boldsymbol{D}_2\boldsymbol{X}\boldsymbol{C}\boldsymbol{Y}^\top = \boldsymbol{D}_1,$$

where, in the last equality, we used the easy-to-check identities that $\boldsymbol{Y}\boldsymbol{Y}^\top = N\boldsymbol{I}_C$ and $\boldsymbol{C}\boldsymbol{Y} = \boldsymbol{0}$. In more detail, the first identity follows from the fact that $\boldsymbol{Y}$ is the one-hot label vectors stacked as columns; the second identity follows from the facts that $\boldsymbol{C}$ is the class-centering matrix, and the one-hot label is the same for examples from the same class. Thus, we have shown $\frac{d\boldsymbol{\Omega}}{dt}$ is diagonal, which concludes our proof. $\qquad\square$

**Lemma 7.** *A matrix $\boldsymbol{E} \in \mathbb{R}^{P \times C}$ is a Simplex ETF if and only if*

1. *$\boldsymbol{E}$ has exactly $C-1$ non-zero singular values, which are all equal.*

2. *$\boldsymbol{E}$ has a rank-1 nullspace spanned by the ones vector, i.e., $\boldsymbol{E}\mathbb{1}_C = \boldsymbol{0}$. In other words, $\boldsymbol{E}$ has zero-mean columns.*

*Proof.* First, recall from Papyan, Han, and Donoho (2020, Definition 1) that a $P \times C$ matrix $\boldsymbol{E}$ is called a Simplex ETF if it satisfies

$$\boldsymbol{E}^\top\boldsymbol{E} = \alpha\left(\frac{C}{C-1}\boldsymbol{I} - \frac{1}{C-1}\mathbb{1}_C\mathbb{1}_C^\top\right)$$

---

[21]Intuitively, changing the diagonal SNR matrix by a diagonal matrix increment implies the updated SNR matrix is still diagonal. This, in turn, implies the left and right singular vectors of the updated SNR matrix are still the left and right partial identity matrices, respectively.

for some scaling $\alpha > 0$. We now prove the equivalence.

Simplex ETF implies 1-2: Consider the SVD decomposition $\boldsymbol{E} = \boldsymbol{U}_E \boldsymbol{S}_E \boldsymbol{V}_E^\top$, where $\boldsymbol{U}_E$ is a $P \times (C-1)$ partial orthogonal matrix satisfying $\boldsymbol{U}_E^\top \boldsymbol{U}_E = \boldsymbol{I}$, $\boldsymbol{S}_E$ is the $(C-1) \times (C-1)$ diagonal matrix of singular values, and $\boldsymbol{V}_E^\top$ is a $(C-1) \times C$ partial orthogonal matrix satisfying $\boldsymbol{V}_E^\top \boldsymbol{V}_E = \boldsymbol{I}$. Since $\boldsymbol{E}$ is a Simplex ETF, by definition,

$$\boldsymbol{E}^\top \boldsymbol{E} = \alpha \left( \frac{C}{C-1} \boldsymbol{I} - \frac{1}{C-1} \mathbb{1}_C \mathbb{1}_C^\top \right),$$

and according to the SVD decomposition

$$\boldsymbol{E}^\top \boldsymbol{E} = \boldsymbol{V}_E \boldsymbol{S}_E \boldsymbol{U}_E^\top \boldsymbol{U}_E \boldsymbol{S}_E \boldsymbol{V}_E^\top = \boldsymbol{V}_E \boldsymbol{S}_E^2 \boldsymbol{V}_E^\top.$$

Therefore,

$$\boldsymbol{V}_E \boldsymbol{S}_E^2 \boldsymbol{V}_E^\top = \alpha \left( \frac{C}{C-1} \boldsymbol{I} - \frac{1}{C-1} \mathbb{1}_C \mathbb{1}_C^\top \right).$$

Notice that the right-hand-side has $C-1$ equal singular values. This implies $\boldsymbol{S}_E$—and, thus, $\boldsymbol{E}$ as well—has $C-1$ equal singular values. In other words, $\boldsymbol{S}_E$ is a $(C-1) \times (C-1)$ diagonal matrix equal to $\mathrm{diag}(s, \ldots, s)$ for some scalar $s > 0$.

Next, notice that on the one hand

$$\boldsymbol{E}^\top \boldsymbol{E} \mathbb{1}_C = \alpha \left( \frac{C}{C-1} \boldsymbol{I} - \frac{1}{C-1} \mathbb{1}_C \mathbb{1}_C^\top \right) \mathbb{1}_C = \boldsymbol{0}.$$

On the other hand,

$$\boldsymbol{E}^\top \boldsymbol{E} \mathbb{1}_C = \boldsymbol{V}_E \boldsymbol{S}_E \boldsymbol{U}_E^\top \boldsymbol{U}_E \boldsymbol{S}_E \boldsymbol{V}_E^\top \mathbb{1}_C = \boldsymbol{V}_E \boldsymbol{S}_E^2 \boldsymbol{V}_E^\top \mathbb{1}_C = s^2 \boldsymbol{V}_E \boldsymbol{V}_E^\top \mathbb{1}_C.$$

Combining the above, we get $\boldsymbol{V}_E \boldsymbol{V}_E^\top \mathbb{1}_C = \boldsymbol{0}$. Since $\boldsymbol{V}_E \boldsymbol{V}_E^\top$ is a $C \times C$ matrix of rank $C-1$, we deduce that the rank-1 nullspace of $\boldsymbol{V}_E^\top$ is spanned by the $C$-dimensional ones vector, i.e. $\boldsymbol{V}_E^\top \mathbb{1}_C = \boldsymbol{0}$. Consequently, $\boldsymbol{E}$ has zero-mean columns since $\boldsymbol{E} \mathbb{1}_C = \boldsymbol{U}_E \boldsymbol{S}_E \boldsymbol{V}_E^\top \mathbb{1}_C = \boldsymbol{0}$.

1-2 implies Simplex ETF: Assume $\boldsymbol{E}$ has exactly $C-1$ non-zero, equal singular values as well as a rank-1 nullspace spanned by the ones-vector, i.e., $\boldsymbol{E} \mathbb{1}_C = \boldsymbol{0}$. Then, $\boldsymbol{E} = \boldsymbol{U}_E \boldsymbol{S}_E \boldsymbol{V}_E^\top$ where

- $\boldsymbol{U}_E$ is a $P \times (C-1)$ partial orthogonal matrix satisfying $\boldsymbol{U}_E^\top \boldsymbol{U}_E = \boldsymbol{I}$;

- $\boldsymbol{S}_E = \mathrm{diag}(s, ..., s)$ is a $(C-1) \times (C-1)$ diagonal matrix for some scalar $s > 0$; and

- $\boldsymbol{V}_E^\top$ is a $(C-1) \times C$ partial orthogonal matrix satisfying $\boldsymbol{V}_E^\top \boldsymbol{V}_E = \boldsymbol{I}$ and also satisfying $\boldsymbol{V}_E^\top \mathbb{1}_C = \boldsymbol{0}$.

Therefore,

$$\boldsymbol{E}^\top \boldsymbol{E} = \boldsymbol{V}_E \boldsymbol{S}_E \boldsymbol{U}_E^\top \boldsymbol{U}_E \boldsymbol{S}_E \boldsymbol{V}_E^\top = \boldsymbol{V}_E \boldsymbol{S}_E^2 \boldsymbol{V}_E^\top = s^2 \boldsymbol{V}_E \boldsymbol{V}_E^\top.$$

Using our assumptions on $\boldsymbol{V}_E$,

$$\boldsymbol{V}_E \boldsymbol{V}_E^\top + \frac{1}{C} \mathbb{1}_C \mathbb{1}_C^\top = \boldsymbol{I},$$

where we divide each of the ones-vectors by $\sqrt{C}$ to create a unit vector. Thus, we conclude

$$\boldsymbol{E}^\top \boldsymbol{E} = s^2 \left( \boldsymbol{I} - \frac{1}{C} \mathbb{1}_C \mathbb{1}_C^\top \right) = \alpha \left( \frac{C}{C-1} \boldsymbol{I} - \frac{1}{C-1} \mathbb{1}_C \mathbb{1}_C^\top \right),$$

where $\alpha = s^2 \frac{C-1}{C}$. Hence, by definition, $\boldsymbol{E}$ is a Simplex ETF. $\qquad\square$

**Corollary 2** (**Neural Collapse Under MSE Loss**). *Under continually renormalized gradient flow (Equation 5), the SNR matrix (Equation 8) converges to*

$$\lim_{t \to \infty} \frac{1}{\omega_{\max}(t)} \mathrm{SNR}_t = \widehat{\boldsymbol{U}}_0 \widehat{\boldsymbol{V}}_0^\top, \tag{10}$$

*where $\widehat{\boldsymbol{U}}_0 \in \mathbb{R}^{P \times (C-1)}$ and $\widehat{\boldsymbol{V}}_0 \in \mathbb{R}^{C \times (C-1)}$ are the left and right singular vectors of the SNR matrix (Definition 2) at $t=0$ corresponding to the non-zero singular values; and $\omega_{\max}(t)$ is the*

*largest singular value at time t. Furthermore, Corollary 1 implies the occurrence of (NC1)-(NC4) i.e. renormalized gradient flow on the central path leads to Neural Collapse.*

*Moreover, denoting the Kronecker product with $\otimes$, the renormalized features matrix converges to*

$$\lim_{t \to \infty} \frac{1}{\omega_{\max}(t)} \mathbf{\Sigma}_{W,t}^{-\frac{1}{2}} \overline{\mathbf{H}}_t = (\widehat{\mathbf{U}}_0 \widehat{\mathbf{V}}_0^\top) \otimes \mathbb{1}_N^\top. \tag{11}$$

*Proof.* Derivation of Equation 10: Corollary 1 proves the singular values $j = 1, \ldots, C-1$ of $\mathrm{SNR}_t$ diverge to infinity and that their ratio tends to one. By Lemma 6, the renormalized gradient flow will not change the singular vectors of the SNR matrix, and—combined with the fact that the singular values converge to equality (second fact of Corollary 1)—we get the limit in Equation 10.

Derivation of **(NC1)**: Let $s_j(\cdot)$ denote the $j$-th singular value of its argument. Then, observe that

$$
\begin{aligned}
\mathrm{tr}\left(\mathbf{\Sigma}_{B,t}^\dagger \mathbf{\Sigma}_{W,t}\right) &= C \, \mathrm{tr}\left(\left(\overline{\mathbf{M}}_t \overline{\mathbf{M}}_t^\top\right)^\dagger \mathbf{\Sigma}_{W,t}\right) && \text{(Def. of } \mathbf{\Sigma}_{B,t}\text{)} \\
&= C \, \mathrm{tr}\left(\mathbf{\Sigma}_{W,t}^{\frac{1}{2}} \left(\overline{\mathbf{M}}_t \overline{\mathbf{M}}_t^\top\right)^\dagger \mathbf{\Sigma}_{W,t}^{\frac{1}{2}}\right) && \text{(Cyclic property of trace)} \\
&= C \sum_{j=1}^{C-1} s_j\left(\mathbf{\Sigma}_{W,t}^{\frac{1}{2}} \left(\overline{\mathbf{M}}_t \overline{\mathbf{M}}_t^\top\right)^\dagger \mathbf{\Sigma}_{W,t}^{\frac{1}{2}}\right) && \text{(Trace is sum of singular values)} \\
&= C \sum_{j=1}^{C-1} s_j^{-1}\left(\mathbf{\Sigma}_{W,t}^{-\frac{1}{2}} \left(\overline{\mathbf{M}}_t \overline{\mathbf{M}}_t^\top\right) \mathbf{\Sigma}_{W,t}^{-\frac{1}{2}}\right) && \text{(Def. of pseudoinverse)} \\
&= C \sum_{j=1}^{C-1} s_j^{-2}\left(\mathbf{\Sigma}_{W,t}^{-\frac{1}{2}} \overline{\mathbf{M}}_t\right) && \\
&= \sum_{j=1}^{C-1} \frac{1}{\omega_j^2(t)} \xrightarrow{t \to \infty} 0. && \text{(Corollary 1)}
\end{aligned}
$$

By Horn & Johnson (2012, Theorem 1.3.22),

$$\lambda_j(\mathbf{\Sigma}_{B,t}^\dagger \mathbf{\Sigma}_{W,t}) = \lambda_j(\mathbf{\Sigma}_{W,t}^{0.5} \mathbf{\Sigma}_{B,t}^\dagger \mathbf{\Sigma}_{W,t}^{0.5}) \geq 0,$$

where $\lambda_j(\cdot)$ denotes the $j$-th eigenvalue of its argument, and the last inequality follows from the positive-semidefiniteness of $\mathbf{\Sigma}_{W,t}^{0.5} \mathbf{\Sigma}_{B,t}^\dagger \mathbf{\Sigma}_{W,t}^{0.5}$. Since the trace is the sum of eigenvalues, the only way for the trace of $\mathbf{\Sigma}_{B,t}^\dagger \mathbf{\Sigma}_{W,t}$ to tend to zero is if all eigenvalues also tend to zero. All eigenvalues tending to zero implies the matrix itself tends to zero, i.e.

$$\lim_{t \to \infty} \mathbf{\Sigma}_{B,t}^\dagger \mathbf{\Sigma}_{W,t} = \mathbf{0},$$

which is the definition of **(NC1)** in Section 1.1.

Derivation of **(NC2)-(NC4)**: Recall that the SNR matrix has zero-mean columns and rank $C-1$ (see Definition 2). This, combined with the fact that the singular values converge to equality (second fact of Corollary 1) imply, by Lemma 7, that the renormalized class-means converge to a Simplex ETF i.e. **(NC2)** .

From Theorem 1 of Papyan, Han, and Donoho (2020), we then know that **(NC3)** and **(NC4)** follow from **(NC1)** and **(NC2)** on the central path.

Derivation of Equation 11: Combining the limit in Equation 10 with **(NC1)** proves the limit in Equation 11. $\qquad\square$

## E  RELATED WORKS EXAMINING NEURAL COLLAPSE

In this section, we discuss the contributions and limitations of seven recent works that propose and analyze theoretical abstractions of Neural Collapse. These works are only available in preprint, and may not yet be peer-reviewed. Thus, they might ultimately appear with very different claims or results. Additionally, works such as Poggio & Liao (2020a;b); Ergen & Pilanci (2020) also analyze behaviors other than NC; we will only discuss the parts relevant to Neural Collapse here.

### E.1 MIXON, PARSHALL, AND PI (2020)

Mixon et al. (2020) considered the unconstrained features model in Equation 2 (without weight decay) where, under gradient flow, $(\boldsymbol{W}, \boldsymbol{H})$ evolve according to a nonlinear ordinary differential equation (ODE). They followed a two-step strategy for studying Neural Collapse. First, they linearized the ODE—claiming nonlinear terms are negligible for models initialized near the origin—and proved the simplified ODE converges to a subspace of $(\boldsymbol{W}, \boldsymbol{H})$ satisfying **(NC1)** and **(NC3)** . Second, they proved that gradient flow, restricted to that subspace, converges to **(NC2)** .

The assumption of small weights and classifiers leading to the linearized ODE is not aligned with today's paradigm. Specifically, the most commonly used He initialization (He et al., 2015) is designed: (i) to create weights with non-negligible magnitude; and (ii) to preserve the magnitude of features, as they propagate throughout the layers of the network, exactly so that last-layer features would have non-negligible magnitude. Moreover, the analysis of Mixon et al. essentially assumes that **(NC1)** and **(NC3)** occur much sooner than **(NC2)** . However, from the experiments in both Papyan, Han, and Donoho (2020) and this paper, there is no empirical evidence that **(NC2)** happens slower than **(NC1)** and **(NC3)** in practice.

### E.2 LU AND STEINERBERGER (2020)

While the MSE loss provides a mathematically natural setting for analysis, the modern paradigm in multi-class classification with deep learning involves training with CE loss, which is more challenging to analyze than MSE.

Lu & Steinerberger (2020) studied the (one-example-per-class) unconstrained[22] features model with CE loss:

$$\min_{\boldsymbol{W}, \boldsymbol{M}} \text{CE}(\boldsymbol{W}, \boldsymbol{M}) \quad \text{s.t.} \quad \|\boldsymbol{w}_c\|_2 = \|\boldsymbol{\mu}_c\|_2 = 1.$$

Since under linear separability the CE loss can be driven arbitrarily close to zero, just by re-scaling the norms of $\boldsymbol{W}$ and $\boldsymbol{M}$, the authors further imposed a norm constraint on $\boldsymbol{w}_c$ and $\boldsymbol{\mu}_c$. Lu & Steinerberger observe that the global minimizer of this optimization problem is only achieved once $\boldsymbol{W}$ and $\boldsymbol{M}$ are the same Simplex ETF. This derivation is suggestive, but it does not identify closed-form dynamics which would get gradient flow to such a global minimizer, nor does it address the rate of convergence to Neural Collapse. Additionally, the constraint on $\boldsymbol{\mu}_c$ possesses no immediate or direct analogy to standard deep net training—where procedures often control the norm of the weights $\boldsymbol{W}$, but not features $\boldsymbol{H}$—nor class-means $\boldsymbol{M}$.

### E.3 E AND WOJTOWYTSCH (2020)

E & Wojtowytsch (2020) also consider the unconstrained features model[22] with CE loss,

$$\min_{\boldsymbol{W}, \boldsymbol{H}} \quad \text{CrossEntropy}(\boldsymbol{W}\boldsymbol{H}) \quad \text{s.t.} \quad \|\boldsymbol{W}\|_2 \leq 1, \|\boldsymbol{h}_{i,c}\|_2 \leq 1,$$

where they adopt a more technical, *spectral* norm constraint on $\boldsymbol{W}$ to specify their model.

Building on the results of Chizat & Bach (2018; 2020), E & Wojtowytsch also construct a simple counter-example showing that Neural Collapse need not occur in two-layer, infinite-width networks—which have been the focus of intense recent study in the theoretical deep learning community (Mei et al., 2018; Rotskoff & Vanden-Eijnden, 2018; Arora et al., 2019). Thus, E & Wojtowytsch's counterexample suggests the alternative perspective that, despite the expressiveness of infinite-width, two-layer networks, such abstractions do not capture key aspects of trained *deep* nets.

As with Lu & Steinerberger (2020), standard deep net training does not possess any direct analogies for constraining the norm of *features* (as opposed to weights)—nor are there any paradigmatic regularizations that correspond to controlling the *spectral* norm on $\boldsymbol{W}$. Moreover, the work does not characterize any closed-form dynamics or the rate of convergence to Neural Collapse.

---

[22]The model is unconstrained in the sense that the features are allowed to move directly with gradient flow and are not constrained to be the output of a forward pass—not in the sense that there are no constraints on the optimization problem.

### E.4 POGGIO & LIAO (2020A;B) (WITH BANBURSKI)

Distinguished from the simplified unconstrained features models in the previously mentioned works is the theoretical analysis of Poggio & Liao (2020a;b) (in a special section, co-authored with Andrzej Banburski).

The authors study deep homogeneous classification networks, with weight normalization layers, trained with stochastic gradient descent and weight decay. This is much closer to today's training paradigm, but the setting still differs from the one in which Neural Collapse has been empirically observed in Papyan, Han, and Donoho (2020) and in Section A of this paper. In particular, they replace batch normalization with weight normalization and consider deep homogeneous networks; homogeneous networks can not have bias vectors nor skip connections, which are present both in ResNet and DenseNet. Moreover, the work gives explicit descriptions of neither the dynamics nor the rate of convergence to Neural Collapse.

### E.5 ERGEN & PILANCI (2020)

While the above-described works tend to focus on either the used-in-practice CE loss or the theoretically-insightful MSE loss, Ergen & Pilanci (2020) observed that these are both instances of the general class of convex loss functions and, thus, one could derive insights from the classical convex analysis literature. Moreover, compared to Mixon et al. (2020); Lu & Steinerberger (2020); E & Wojtowytsch (2020), this work studies the optimization starting from the *second*-to-last layer features rather than the last-layer features. In particular, the authors use a strong-duality argument to show that NC emerges in the optimal solution of an equivalent proxy-model to the following optimization:

$$\min_{\boldsymbol{H}_{L-1}, \boldsymbol{W}_{L-1}, \boldsymbol{W}_L, \gamma, \alpha} \mathcal{L}\left(\boldsymbol{W}_L\left(\mathrm{BN}_{\gamma,\alpha}\left(\boldsymbol{W}_{L-1}\boldsymbol{H}_{L-1}\right)\right)_+, \boldsymbol{Y}\right) + \frac{\lambda}{2}\left(\|\gamma\|_2^2 + \|\alpha\|_2^2 + \|\boldsymbol{W}_L\|_F^2\right),$$

where $\mathcal{L}(\cdot)$ is a general convex loss, $\boldsymbol{H}_{L-1}$ are the second-to-last layer activations, $\boldsymbol{W}_{L-1}$ are the second-to-last layer weights, $\boldsymbol{W}_L$ are the network classifiers, $\boldsymbol{Y}$ are the training targets, $\lambda$ is a weight-decay parameter, $\mathrm{BN}_{\gamma,\alpha}(\cdot)$ is a batch-norm operator parameterized by $\alpha$ and $\gamma$, and $(\cdot)_+$ is a ReLU. The incorporation of batch-normalization and weight-decay ensures the existence of bounded, well-defined optimal solutions—serving a similar role to that of weight-normalization and weight decay in Poggio & Liao (2020a;b) as well as the norm constraints in the other aforementioned related works.

Since strong-duality only characterizes properties of the converged optimal solution of an optimization model, Ergen & Pilanci (2020) does not provide insights into the dynamics with which that solution is achieved which training.

### E.6 FANG, HE, LONG, AND SU (2021)

In Fang et al. (2021), the authors introduce the ($N$-)*layer-peeled model* in which one considers only the direct optimization of the $N$-th-to-last layer features of a deep net along with the weights that come after the $N$-th-to-last layer. The motivating philosophy is that, after raw inputs are passed through some initial number of layers, the overparameterization of those layers would allow us to effectively model the $N$-th-to-last layer features as freely-moving in some subset of Euclidean space. In this terminology, the concurrent works of Mixon et al. (2020); Lu & Steinerberger (2020); E & Wojtowytsch (2020) on the unconstrained (last-layer) features model could be considered instances of a 1-layer-peeled model; while the model of Ergen & Pilanci (2020) could be considered as a 2-layer-peeled model. This perspective is attractive because it gives a name and organization to a common modeling philosophy behind the above-described body of independent works. For comparison, the work of Poggio & Liao (2020a;b) is a non-example of layer-peeled modeling as it considers optimization on the weights of homogeneous deep nets and not the input features.

Fang et al. (2021) then analyzes a convex relaxation of the 1-layer-peeled model—with norm constraints on the weights and features—into a semidefinite program. Not only do the authors show that this model exhibits Neural Collapse in the canonical setting of balanced examples-per-class, but they also analyze the behavior of this model under *imbalanced classes*. While, in the imbalanced case, one would intuitively expect the Simplex ETF to "skew" to have bigger angles around over-represented classes and smaller angles around under-represented ones; Fang et al. (2021) identifies the

surprising phenomenon—named *minority collapse*—in their model where, when the imbalances pass a certain threshold, *the last-layer features and classifiers of the under-represented classes collapse to be exactly the same*. However, their work does not provide closed-form dynamics or rates at which collapse—in either the balanced or imbalance case—occurs.

### E.7    ZHU, DING, ZHOU, LI, YOU, SULAM, AND QU (2021)

In Zhu et al. (2021), the authors examine the following unconstrained features model:

$$\min_{\boldsymbol{W},\boldsymbol{H},\boldsymbol{b}} \text{CrossEntropy}\left(\boldsymbol{W}\boldsymbol{H} + \boldsymbol{b}, \boldsymbol{Y}\right) + \frac{\lambda_{\boldsymbol{W}}}{2}\left\|\boldsymbol{W}\right\|_F^2 + \frac{\lambda_{\boldsymbol{H}}}{2}\left\|\boldsymbol{H}\right\|_F^2 + \frac{\lambda_{\boldsymbol{b}}}{2}\left\|\boldsymbol{b}\right\|_F^2,$$

where $(\boldsymbol{W}, \boldsymbol{b})$ are the classifier weights and biases, $\boldsymbol{H}$ are the last layer features, and $(\lambda_{\boldsymbol{W}}, \lambda_{\boldsymbol{H}}, \lambda_{\boldsymbol{b}})$ are weight-decay parameters. On this model, the authors not only prove that all minima exhibit Neural Collapse but also that *all local minima are global minima*. In comparison to our current paper, Zhu et al. (2021) focus on characterizing the landscape of the loss and, thus, do not explore the dynamics and rate at which such minimizers of the loss are achieved.

Zhu et al. (2021) also make notable empirical contributions by conducting a series of experiments on the MNIST and CIFAR10 datasets trained on MLPs, ResNet18, and ResNet50. Their measurements give evidence for the following novel NC-related phenomena in deep classification networks:

1. **NC is algorithm independent:** NC emerges in realistic classification deep net training regardless of whether the algorithm is SGD, ADAM, or L-BFGS.
2. **NC occurs on random labels:** NC emerges even when the one-hot target vectors are completely shuffled.
3. **Width improves NC:** Increasing network width expedites NC when training with random labels.

The authors of Zhu et al. (2021) moreover conducted ablation experiments suggesting that the following substitutions can be made to deep neural net architectures without affecting performance:

1. **Weight-decay substitution:** Replacing (A) weight-decay on the norm of all network parameters with (B) weight-decay just on the norm of the last-layer features and classifiers.
2. **Classifier substitution:** Replacing (A) the last-layer classifiers that are trained with SGD with (B) Simplex ETF classifiers that are fixed throughout training.

While the authors only demonstrated these new behaviors on limited network-dataset combinations, these experiments indeed inspire interesting conjectures about the generalization behavior of deep nets as well as potential architecture design improvements; Zhu et al. (2021) discuss many of these conjectures and related open-questions in detail.

