# OpenReview forum: "Neural Collapse Under MSE Loss: Proximity to and Dynamics on the Central Path"
_ICLR.cc/2022/Conference — ICLR 2022 Oral_

### Official Review · Reviewer_9pr9 · 2021-11-02

**Correctness:** 4
**Technical Novelty And Significance:** 4
**Empirical Novelty And Significance:** 4
**Recommendation:** 8
**Confidence:** 4

**Main Review:**

Pros:

--The authors provided extensive experiments to show that NC occurs in DNNs with MSE loss.

--The authors provided motivation and explained the required conditions of the theorems and the limitations of their results. Overall, the conditions of the theorems do not look too restrictive and unrealistic.

--The appendix has an extensive overview of related work.

--In the paper, a new theoretical construct, central path, is proposed and analysed.

Cons:

--Overall, the paper is well-written, but there are some places that are not easy to follow, for example, Section 1.1. I would recommend the authors to state the problem and all the necessary notions first and only after this to introduce what NC is. The first sentence of the introduction also does not sound good.

--The authors investigate the NC phenomenon from the point of view of the last classifier layer only and do not consider how DNNs generate the features that are passed as input to this classifier. It is still not clear how the NC phenomena are connected to generalization. While MSE delivers similar performance to cross-entropy loss (CE), it is less used in classification settings than CE.


Additional comments:

--In Section 1.3 I would recommend adding references to datasets and architectures.

--I would recommend adding an explanation into Section 1.2 about how classification performed when the model is trained with MSE loss.

--Could you please clarify for me whether NC4 is an independent condition or it follows from (NC1-NC3)?

--Could you please clarify how condition (A2) is related to empirical observations?

**Summary Of The Paper:**

The authors of this paper proposed a theoretical explanation of Neural Collapse (NC) for the DNNs with MSE loss. In particular, the authors showed that MSE loss can be decomposed into a sum of terms that corresponds to NC conditions and proposed the theoretical model (central path) that explains why NC emerges in DNNs with MSE under some mild assumptions.

**Summary Of The Review:**

I would recommend accepting the paper. The paper further extends our theoretical understanding of deep learning. The result is new and supported by experiments.

---

> ### Author Response · Authors · 2021-11-14
> **Authors' Response to Reviewer "9pr9" (Part 2/2)**
>
> 4. *"While MSE delivers similar performance to cross-entropy loss (CE), it is less used in classification settings than CE."*
>     #
>    We agree that deep nets are typically trained with Cross-Entropy loss rather than MSE loss. At the same time, recent papers by many researchers consider the MSE situation—such as [Demirkaya, Chen, and Oymak, 2020], [Hui and Belkin, 2020], [Poggio and Liao, 2020a,b], [Mixon, Parshall, and Pi, 2020], and [Ergen and Pilanci, 2020]. Therefore, we feel it is worth noting that scholars’ attitudes about MSE *are* changing: Partly because it has been shown—by works such as [Demirkaya, Chen, and Oymak, 2020] and [Hui and Belkin, 2020]—that training with MSE can produce well-performing networks for image-classification tasks (even the more challenging ones such as CIFAR and ImageNet).
>     #
>    Moreover, there are insights one can glean from analyzing the MSE loss that currently go deeper than those possible with the more analytically-complex Cross-Entropy. These MSE-based findings possess scientific value since MSE-trained deep nets display many of the same empirical behaviors as Cross-Entropy-trained ones. In other words, the deeper results discovered from analyzing the MSE loss may constitute first-steps in eventually extending the derived theoretical insights to broader, more-complex settings.
>     #
> 5. *"In Section 1.3 I would recommend adding references to datasets and architectures."*
>     #
>    Thank you for this suggestion. Following the reviewer’s suggestion, we now add datasets and architecture references in Section 1.3.
>     #
> 6. *"I would recommend adding an explanation into Section 1.2 about how classification performed when the model is trained with MSE loss."*
>     #
>    Thank you for this suggestion. Our experiment in Table 1 (analogous to Table 1 of [Papyan, Han, Donoho 2020]) of the appendix shows that models trained with MSE loss perform comparably when compared to the models trained with CE loss. Following the reviewer’s suggestion, we now mentioned this fact explicitly in Section 1.2.
>     #
> 7. *"Could you please clarify for me whether NC4 is an independent condition or it follows from (NC1-NC3)?"*
>     #
>    Thank you for this question. NC4 indeed follows from NC1-3. This is explicitly shown within the proof of Theorem 1 in [Papyan, Han, and Donoho 2020] (starting from their Equation 7b).
>     #
> 8. *"Could you please clarify how condition (A2) is related to empirical observations?"*
>    #
>    In our discussion of (A2), we show that biases on the central path are effectively doing global mean subtraction. In our Figure 2 measurements, we showed that the deviation from the central path is negligible. Therefore, we see that an empirical network is effectively acting as a bias-less network where the features have been globally-mean-subtracted—leading to (A2).
>    #
>
> We again express our deep gratitude to the reviewer for their careful reading of the paper and positive affirmation of the quality of our work.  Given the above description of the relevance of MSE-based theoretical research, the potential insights from the unconstrained features model, as well as the revisions we made to improve the paper, might the reviewer consider raising their score even more to help this work gain acceptance?

---

> > ### Comment · Reviewer_9pr9 · 2021-11-24
> > **Comment**
> >
> > I would like to thank the authors for their clarifications. I believe it is a good paper and I would recommend accepting it.

---

> ### Author Response · Authors · 2021-11-14
> **Authors' Response to Reviewer "9pr9" (Part 1/2)**
>
> We thank the reviewer for their careful reading of this paper as well as their positive assessment of its relevance. This paper is the culmination of over a year’s worth of careful empirical and theoretical research, so we are very grateful for the reviewer’s validation of its quality.
>
> #
>
> Below, we give point-by-point responses to the reviewer’s comments and describe the revisions we made to address them. The revised paper has been uploaded to OpenReview. We would be happy to add additional clarifications and revisions to the paper to address any additional recommendations from the reviewer.
>
> 1. *"Overall, the paper is well-written, but there are some places that are not easy to follow, for example, Section 1.1. I would recommend the authors to state the problem and all the necessary notions first and only after this to introduce what NC is. The first sentence of the introduction also does not sound good."*
>     #
>    We thank the reviewer for this feedback. Following the reviewer’s feedback, we have rewritten Section 1.1 to introduce all necessary notions first before introducing NC. Additionally, we have rewritten the first sentence.
>     #
> 2. *"The authors investigate the NC phenomenon from the point of view of the last classifier layer only and do not consider how DNNs generate the features that are passed as input to this classifier."*
>     #
>    Yes, we agree with the reviewer that this work does not completely capture the multi-layer aspect of deep nets.
>     #
>    On the other hand, the unconstrained features model *does* give insights into the ability of deep networks to perform collapse because it models the interaction between deep net feature engineering and classifiers during training. In particular, the unconstrained features model is a mathematical model of feature engineering mechanisms that have unlimited flexibility. This flexibility is exactly motivated by the overparameterization and strong-nonlinearity induced by the stacking of many layers in modern deep nets. At the same time, the central path characterizes the situation where network classifiers are optimal relative to a fixed set of engineered features—which is empirically observed in our supplementary experiments on canonical deep nets and architectures.
>     #
>    This paper mathematically shows that the interaction of such a feature engineering mechanism with unlimited flexibility and an optimally-paired classifier leads to the Neural Collapse—with dynamics that can be expressed in closed form. Since realistic deep nets indeed possess both exceptional feature engineering flexibility and features-and-classifiers pairs that are effectively on the central path, the unconstrained features model on the central path is a well-motivated model for understanding the ability of deep nets to perform collapse.
>     #
> 3. *"It is still not clear how the NC phenomena are connected to generalization."*
>     #
>    In this paper, we show that the prevalent empirical practice of training past zero-error towards zero-loss—within the (experimentally-motivated) unconstrained features model on the central path—has various exact consequences including explicit closed-form expressions for the trajectory of the classifiers and training features. Only after developing these rigorous *descriptive* characterizations of Neural Collapse can researchers then deduce equally-rigorous *predictive* theorems regarding its connection to generalization.
>     #
>    More specifically, armed with this paper’s precise characterization of the trained classifier, we are now using—in on-going projects—linear algebra, functional analysis, and multivariate statistics tools to investigate the performance of the trained classifier on fresh data—i.e. generalization. We plan to document this in future, follow-up reports.
>     #

---

### Official Review · Reviewer_pKnQ · 2021-11-02

**Correctness:** 3
**Technical Novelty And Significance:** 3
**Empirical Novelty And Significance:** Not applicable
**Recommendation:** 8
**Confidence:** 4

**Main Review:**

**Pro**

\+ The submission is well motivated and coherently structured. (Empirical observation -> Splitting the loss function in summands -> Examining the gradient flow from the predominant summand)

\+ The theory is able to explain, some aspects, of the empirical observations by Papyan, Han and Donoho (2020) . It separates itself from the prior work, as it contains closed form solutions of the class means (of renormalized features).

\+ The submission is relevant, as understanding why and how optimization of current neural networks works is crucial for making informed improvements in future algorithms.

**Contra**

\- The main result (Section 3.3) is formalized in terms of the singular values of the SNR matrix. However, the interpretation is not obvious. I would advice to expand on the footnote 9 and move it to the main text as a brief discussion of the findings.

\- There is an additional assumption which is required for Theorem 3, which is not mentioned in the statement and the subsequent discussion, but somewhere later: The within class covariance $\Sigma_W$ is full rank. While one expects the assumption to be fulfilled, this assumption should still be stated more prominently.

\- There is a correctness issue, due to which I currently cannot give a higher score. The conclusions might not necessarily be wrong, but I am a bit wary. I hope, this can be resolved with the author response.

1. It is claimed that Corollary 1 and Eq. 11 imply neural collapse. So in particular, they imply (NC1): $\Sigma_W \to 0$. However, these corollaries are concluded from the dynamics specified in Equation 5, where the representations are always such that $\Sigma = I$ is constant.

2. The first limit in Corollary 2, i.e. the statement that the singular vectors  of SNR stay constant with respect to $t$, is reasoned for by the fact that $L(w(t))$ only depends on the singular values (and not the singular vectors) and so its gradient does not depend on the singular vectors. However, the dynamics in Eq 5 do not only depend on the gradient, but also on an projection operator, which might change the singular vectors.
Intuitively, the dynamics in Eq 5 correspond to gradient updates followed by a renormalization step, i.e. matrix multiplication such that $X$ has identity covariance. This multiplication might possibly chance the singular vectors.


*Questions and suggestions unrelated to score:*

It might be a good idea to reduce the notation in section 2, by considering only the case of zero bias and move the general case to the appendix. In Section 3, the setting is reduced to the case of no bias term anyway.

Is symmetry really required for equation 8 to hold, or does invertibility suffice?

Could you explain the terminology central path (Equation 4)? I guess, central refers to the least-squares optimality, but why is it a path? It does not appear to be one dimensional.

In appendix D, there is a rather informal discussion on the feature space as a fiber bundle. Can the renormalized gradient flow be formalized as a connection on this bundle?

There is a typo in Corollary 1. The second limit should depend on $c_3$ instead of $c_4$.

**Summary Of The Paper:**

Recently, Papyan, Han and Donoho (2020) have observed that training neural networks beyond zero training error leads to simplex arrangements of the features. The submission studies this phenomenon, called neural collapse, from a theoretical perspective, when training is done via minimizing the square loss.

In particular, it is shown that the square loss can be split, such that one summand corresponds to the quality of the features (quantified by the loss of the MSE classifier on them) and that the other summand corresponds to the quality of the classification layer (quantified by the its deviationfrom the least squares classifier). Furthermore, the first summand quantifies the closeness of the features to neural collapse.
In a second step, the authors investigate the gradient flow induced from the first summand and derive closed form solutions.

**Summary Of The Review:**

I am quite positive on the submission, however, there are two issues related to the proofs of the theorems. These issues might only be due to a misunderstanding on my part, but I am not sure. I hope the authors can clarify on this. Until then, I rate the submission as "5: marginally below the acceptance threshold".

**Post author response update**

The authors adressed the correctness issues.

Regarding 1. The authors confirmed, that Corollary 1, does *not* imply that NC1 ($\lim_{t\to\infty} \Sigma_{W,t} = 0$) and presented the remedy of replacing NC1 by a scale sensitive version, corresponding to the ratio of inner class variations to the between-class variations.  While this is a weaker result, it is still a wortwhile contribution and strongly connected to the observations by Papyan, Han and Donoho (2020).
In fact, the updated definition of NC1, is precisely the quantity measured in the experiments by Papyan, Han and Donoho.

Regarding 2. The authors added the missing proof to the appendix, that the singular vectors of SNR indeed stay constant when subject to the dynamics from Eq. 5.

With these two major issues out of the way I will adapt my recommendation to **8, accept**.

Last but not least, I want to thank the authors for their detailed and clear responses. I really appreciate, that you took the time and effort to already updating the manuscript with proofs for my initial and follow-up questions.

---

> ### Author Response · Authors · 2021-11-14
> **Authors' Response to Reviewer "pKnQ" (Part 2/2)**
>
> 5. *"It might be a good idea to reduce the notation in section 2, by considering only the case of zero bias and move the general case to the appendix. In Section 3, the setting is reduced to the case of no bias term anyway."*
>     #
>    We thank the reviewer for this suggestion. The bias-term in Section 2 and corresponding Figure 2 (measured on deep nets with bias) plays a key role in justifying our theoretical assumptions in Section 3. In particular, Figure 2 shows that real deep nets (with bias) converge to having weights and biases on the central path (defined with bias). In Section 3, to justify assumption (A2), we then explain that having biases on the central path is equivalent to having mean-subtracted features with no biases at all. Given the key role of the bias in justifying the relevance of our theory, we ask for the reviewer’s understanding for our choice to incorporate bias in Section 2.
>     #
>
> 6. *"Is symmetry really required for equation 8 to hold, or does invertibility suffice?"*
>     #
>    Thank you for this question. The reviewer is correct: Invertibility will indeed suffice for Equation 8 (which is now Equation 7 in the revision). Within this paper, we choose to consider only symmetric $A$ as we will later be interested exclusively in the case where $A = \Sigma_W$.
>     #
>
> 7. *"Could you explain the terminology central path (Equation 4)? I guess, central refers to the least-squares optimality, but why is it a path? It does not appear to be one dimensional."*
>     #
>    During optimization, the classifier-feature pair "traverses" a path where the classifier remains MSE-optimal relative to the features i.e. the classifier-feature tuple is "following a central path". Therefore, we call the entire set where this classifier-features optimality holds the "central path" so that any path landing within this set will then, rhetorically, be "on the central path".
>     #
>
> 8. *"In appendix D, there is a rather informal discussion on the feature space as a fiber bundle. Can the renormalized gradient flow be formalized as a connection on this bundle?"*
>     #
>    We thank the reviewer for proposing this idea: We find it fascinating. In the process of writing the paper, we have primarily thought of modeling the training dynamics as a flow on the features. We adopt only the language of differential geometry to specify the tools we use to complete the analysis. We agree it is interesting to consider whether the renormalized gradient flow might be formalized into a connection on the fiber bundles. An answer to this question is not immediate to us, but we are very curious to hear any ideas the reviewer might have in mind.
>     #
>
> 9. *"There is a typo in Corollary 1. The second limit should depend on c3 instead of c4."*
>     #
>    Thank you very much. This is indeed a typo and we have fixed it.
>     #
>
> We again express our gratitude to the reviewer for their dedicated time and effort in examining the content and proofs of this paper. Given our demonstration of the correctness of our proofs as well as the other clarifications above, might the reviewer consider raising their score for this paper and recommending it for acceptance?
>
> We would also be happy to provide additional clarifications to any details of the proofs that the reviewer thinks should be further elaborated upon.

---

> > ### Comment · Reviewer_pKnQ · 2021-11-18
> > **Reply to response**
> >
> > Thank you for your detailed response. My concerns were adressed and I will raise my score.
> >
> > Some remarks regarding the updated manuscript / your comments:
> >
> > - The (updated) argument for NC1 makes use of the divergence of the singular values of the SNR. Could your theory be adapted for settings, where this divergence cannot happen due to norm constraints or penalties on the features (for example induced by weight decay)?
> >
> > - Would you add some verbosity to the proof of corollary 2?
> >   *"Based on Equation 10, we deduce the within-class covariance, $\Sigma_W( \dots)$, becomes negligible compared to the between-class covariance. Therefore, the normalized matrix [...] undergoes activation collapse (NC1) "*
> >   Could you pair this with a calculation of the form $\lim_{t\to \infty} \Sigma_B^\dagger(t) \Sigma_{W_{LS}}(t) = \dots = 0$.
> >   Furthermore, could you provide a short proof or reference to *"The only C-column matrices possessing C−1 equal, non-zero singular values are Simplex ETFs"*.
> >
> > - Regarding 5. I was under the impression, that your theory still works for classification layers without bias. From your answer it seems, that this might not be the case. Could you clarify?
> >
> > - Regarding 8. Indeed, it's an interesting question. When reading the manuscript, it appeared to me, that this section was written already with the idea of a connection in mind. This is why I asked.
> >
> > - I think the added footnote 11 is rather confusing and from my point of view provides no benefit. Manifolds are ubiquitous in geometry and topology and therefore also appear in various contexts in machine learning.
> > Why the clarfification? For other mathematical objects such as vectors, functions or sets one does usually not point out that there are other papers using these objects for formalizing there arguments.
> > In fact, the only clarification needed here, is that $X$ is indeed a manifold (which boils down to a condition on the derivative of the covariance).
> > Furthermore, one could be more specific, as the gradient is not projected onto $X$, but on its tangent space.
> > I am aware, that the footnote was added in response to a question by Reviewer Jnw3, but still, I suggest to remove it.

---

> > > ### Author Response · Authors · 2021-11-20
> > > **Re: Reply to response**
> > >
> > > We are very grateful for the reviewer's decision to raise our score. Additionally, we thank the reviewer for these new, detailed comments on the updated paper. Based on the reviewer's new feedback, we have made additional revisions to the paper. The revision is uploaded to OpenReview.
> > >
> > > Below are point-by-point replies to the reviewer's new comments and suggestions:
> > >
> > > 1. *"The (updated) argument for NC1 makes use of the divergence of the singular vales of the SNR. Could your theory be adapted for settings, where this divergence cannot happen due to norm constraints or penalties on the features (for example induced by weight decay)?"*
> > >    #
> > >    We thank the reviewer for this thought-provoking question. As-is, there is no short, immediate adaptation of the SNR-divergence theory to the case where the norms of the features are constrained. With more derivations, we speculate that such an extension would involve (1) showing that the feature norm constraint effectively restricts the features to some mathematically-characterizable, bounded feasible region and (2) showing that the SNR would diverge until it reaches the boundary of such a region. We are definitely interested in proving such a result in future reports, but see no simple way of showing it within the scope of the current paper.
> > >    #
> > >
> > > 2. *"Would you add some verbosity to the proof of corollary 2? 'Based on Equation 10, we deduce the within-class covariance, $\Sigma_W$, becomes negligible compared to the between-class covariance. Therefore, the normalized matrix [...] undergoes activation collapse (NC1) ' Could you pair this with a calculation of the form $\lim_{t\to\infty}\Sigma_{B}^\dagger(t)\Sigma_{W}(t)=\dots=0.$"*
> > >    #
> > >    Thank you for this feedback. Based on the reviewer’s recommendation, we now provide a detailed derivation of $\lim_{t \to \infty}\Sigma_{B,t}^\dagger\Sigma_{W,t} = 0$ in the proof of Corollary 2 using only results from Corollary 1.
> > >    #
> > >
> > > 3. *"Furthermore, could you provide a short proof or reference to 'The only C-column matrices possessing C−1 equal, non-zero singular values are Simplex ETFs.'"*
> > >    #
> > >    We thank the reviewer for this suggestion. Based on the reviewer’s recommendation, we now explicitly prove this fact in Lemma 7 of Appendix D.6---prior to the proof of Corollary 2. Additionally, we include a reference to this result in the intuitive discussion after Corollary 1 of the main text.
> > >    #
> > >
> > > 4. *"Regarding 5. I was under the impression, that your theory still works for classification layers without bias. From your answer it seems, that this might not be the case. Could you clarify?"*
> > >    #
> > >    Thank you very much for this thought-provoking question. The reviewer indeed identified an interesting subtlety in the theory. A bias---which, as we show in (A2), performs a global mean subtraction on the features---is indeed needed to induce a Simplex ETF structure on the classifier and features. If there were no bias, the classifiers and features would not converge to a globally-mean-subtracted structure like the Simplex ETF. We speculate the limit would be instead an orthogonal matrix, although we have not formally gone-through such a derivation.
> > >
> > >    More formally, the Simplex ETF has a rank-1 nullspace due to the global-mean subtraction of the features caused by the bias. Removing the bias would prevent the global-mean subtraction and therefore eliminate the rank-1 nullspace. "Filling-in" that rank-1 nullspace of the Simplex ETF would result in an orthogonal matrix.
> > >    #
> > >
> > > 5. *"I think the added footnote 11 is rather confusing and from my point of view provides no benefit … In fact, the only clarification needed here, is that $\mathcal{X}$ is indeed a manifold (which boils down to a condition on the derivative of the covariance). Furthermore, one could be more specific, as the gradient is not projected onto $\mathcal{X}$, but on its tangent space."*
> > >    #
> > >    We thank the reviewer for this feedback. We have revised the text to say:
> > >    “...where the operator $\Pi_{\mathcal{T}_\mathcal{X}}X$ projects the gradient onto the tangent space of the manifold...”
> > >
> > >    For the footnote, to address both this feedback and that of Reviewer Jnw3, we have now shortened Footnote 11 to only state that $\mathcal{X}$ is the well-known Generalized Stiefel manifold with a relevant reference. All additional redundant text has been removed.
> > >    #
> > >
> > > We again thank the reviewer for their meticulous effort in evaluating this paper as well as for their many suggestions that improved its quality. We would be happy to provide additional clarifications for any additional feedback the reviewer may have.

---

> ### Author Response · Authors · 2021-11-14
> **Authors' Response to Reviewer "pKnQ" (Part 1/2)**
>
>  We thank the reviewer for their careful reading of this paper as well as the meticulous efforts dedicated to examining the proofs. Based on the reviewer’s questions, comments, and recommendations, we have made many revisions that significantly improved the quality of the paper. The revised paper has been uploaded with this submission. In particular, addressing the reviewer’s main concern, we show that our theoretical results are indeed correct by (1) presenting a more precise formalization of (NC1) than that used by [Papyan, Han, Donoho 2020] and (2) now proving explicitly in Appendix D.6 that the renormalized gradient flow does not change the left and right singular vectors of the renormalized SNR matrix.
>
> #
> Below, we provide point-by-point responses to the reviewer’s feedback and comments as well as the revisions we made to address them. We would be happy to add additional clarifications and revisions to the paper to address any additional recommendations from the reviewer.
>
> #
>
>  1. *"The main result (Section 3.3) is formalized in terms of the singular values of the SNR matrix. However, the interpretation is not obvious. I would advice to expand on the footnote 9 and move it to the main text as a brief discussion of the findings."*
>     #
>     Thank you for this suggestion. Following the reviewer’s suggestion, we moved footnote 9 to the main-text and elaborated upon the interpretation.
>     #
>
> 2. *"There is an additional assumption which is required for Theorem 3, which is not mentioned in the statement and the subsequent discussion, but somewhere later: The within-class covariance $\Sigma_W$ is full rank. While one expects the assumption to be fulfilled, this assumption should still be stated more prominently."*
>     #
>     Thank you for this feedback. We now explicitly state that $\Sigma_W$ is full-rank immediately preceding the statement of Theorem 3.
>     #
>
> 3. *"It is claimed that Corollary 1 and Eq. 11 imply neural collapse. So in particular, they imply (NC1): $\Sigma_W$→0. However, these corollaries are concluded from the dynamics specified in Equation 5, where the representations are always such that $\Sigma_W=I$ is constant."*
>     #
>    Thank you for pointing this out. We now state the definition of (NC1) more precisely as the convergence of $\Sigma_B^\dagger \Sigma_W$ to 0 with training. This is, in fact, the actual empirical quantity measured by [Papyan, Han, and Donoho, 2020] in their Figure 6 when demonstrating the occurrence of (NC1). It is also more intuitive as the classification performance is determined by the size of the "noise" (captured by $\Sigma_W$) relative to the size of the class-means (captured by $\Sigma_B$). We suspect that [Papyan, Han, and Donoho 2020] only used $\Sigma_W \to 0$ for rhetorical simplicity.
>     #
>    This more precise formalization of NC1 resolves the apparent discrepancy noted by the author. Indeed, we prove the non-zero singular values of the SNR matrix $\Sigma_W^{-0.5} M$ tend to infinity, which is equivalent to saying that $\Sigma_B^\dagger \Sigma_W$ tends to zero (because $\Sigma_B=MM^T/C$), which implies the revised definition of (NC1).
>     #
>
> 4. *"The first limit in Corollary 2, i.e. the statement that the singular vectors of SNR stay constant with respect to t, is reasoned for by the fact that $L(w(t))$ only depends on the singular values (and not the singular vectors) and so its gradient does not depend on the singular vectors. However, the dynamics in Eq 5 do not only depend on the gradient, but also on an projection operator, which might change the singular vectors.
>     Intuitively, the dynamics in Eq 5 correspond to gradient updates followed by a renormalization step, i.e. matrix multiplication such that $X$ has identity covariance. This multiplication might possibly chance the singular vectors."*
>     #
>     We thank the reviewer for their feedback and the meticulous reading of our supplementary material and its proofs. We now prove a new Lemma 6 (in Section D.6, immediately before the proof of Corollary 2) explicitly showing that the renormalized gradient flow will not change the singular vectors of the SNR matrix. We revised the proof of Corollary 2 to be more precise by invoking Lemma 6.

---

### Official Review · Reviewer_pHXk · 2021-11-02

**Correctness:** 3
**Technical Novelty And Significance:** 3
**Empirical Novelty And Significance:** 3
**Recommendation:** 6
**Confidence:** 3

**Main Review:**

Strengths:
- The paper empirically shows that NC also occurs when training deep networks for classification using the MSE loss on five canonical datasets and three backbone networks.
- The theoretical part of the paper to justify the NC phenomenon for the unconstrained features model seems rigorous. The decomposition of the loss function and how it is helpful in understanding NC is interesting.

Weaknesses:
- I think the paper contains too much information to be wised packed in a single 9-page conference paper. Specifically, for one of its main contribution, namely the empirical study of the NC with MSE loss, almost all the supportive experiments are deferred to the Appendix, while the main body of the paper only focuses on explaining the theoretical part. I have doubts on such practice (claim the contribution are two-fold while only mainly presenting one of them in the main paper). Note that the authors also admit in Section 1.3 that the empirical study is too long to be included in the paper.

- Some statements need more clarification. In the legend of Figure 2, the term "Lperp" should be referred to as "L^\perp" in the caption. Also, in the caption it says "early in the training, L^\perp becomes negligible compared to the dominant term LNC1"; however, I don't see this from the figure: for many of them, the L^\perp curve is on the above of LNC1 during the early phase of training. How to explain this?

**Summary Of The Paper:**

This paper studies the phenomenon of Neural Collapse (NC) and empirically shows that it occurs during the training of deep networks with the MSE loss. Then it theoretically analyzes NC with MSE loss by decomposing it and introducing the notion of central path. It shows a closed-form dynamics predicts NC in this setting.

**Summary Of The Review:**

The strength and weakness of the paper are very clear, as described above. I would give an overall score of marginally above the acceptance threshold based on its theoretical nature and serious study of the phenomenon.

---

> ### Author Response · Authors · 2021-11-14
> **Authors' Response to Reviewer "pHXk"**
>
> We thank the reviewer for their positive assessment of this paper. Based on the reviewer’s feedback, we have revised the paper to better clarify our contributions as well as the intended description of Figure 2. The revised paper has been uploaded to OpenReview.
>
> #
> Below, we provide point-by-point responses to the reviewer’s feedback. We would be happy to add additional clarifications and revisions to the paper to address any additional recommendations from the reviewer.
>
> #
>
>   1. *"I think the paper contains too much information to be wised packed in a single 9-page conference paper. Specifically, for one of its main contribution, namely the empirical study of the NC with MSE loss, almost all the supportive experiments are deferred to the Appendix, while the main body of the paper only focuses on explaining the theoretical part. I have doubts on such practice (claim the contribution are two-fold while only mainly presenting one of them in the main paper). Note that the authors also admit in Section 1.3 that the empirical study is too long to be included in the paper."*
>
>      #
>      We agree with the reviewer on this aspect. We had previously tried to publish the MSE experiments within a stand-alone paper, but reviewers felt that the empirical experiments themselves lacked novelty. Therefore, they are now a supplementary part of this theoretical work. In response to the reviewer’s feedback, we revised our abstract to clarify that the empirical experiments are a preliminary demonstration rather than a main contribution. We also removed the experiments from the bullet list of main contributions in Section 1.3, and only discuss it afterwards while clearly explaining that it is an supplementary deliverable.
>      #
>
>   2. *"Some statements need more clarification. In the legend of Figure 2, the term "Lperp" should be referred to as $L_{\text{LS}}^\perp$ in the caption."*
>
>      #
>      Thank you for pointing this out. We have fixed have fixed all typos in the legend.
>      #
>
>   3. *"Also, in the caption it says "early in the training, $L_{\text{LS}}^\perp$ becomes negligible compared to the dominant term LNC1"; however, I don't see this from the figure: for many of them, the $L_{\text{LS}}^\perp$ curve is on the above of LNC1 during the early phase of training. How to explain this?"*
>
>      #
>      Thank you for bringing this to our attention. What we intended to say is that the phenomenon where $L_{\text{LS}}^\perp$ is smaller than LNC1 begins to occur at an early epoch relative to the entire training process. We did not intend to say that this occurs during all early epochs of training. We have revised the wording to say "starting from an early epoch in training" rather than simply "early in training".
>      #
>
> We again thank the reviewer for their time and effort dedicated to reading and suggesting improvements to this paper. Given the clarifications and revisions described above, might the reviewer consider raising their score?

---

### Official Review · Reviewer_Jnw3 · 2021-11-03

**Correctness:** 3
**Technical Novelty And Significance:** 3
**Empirical Novelty And Significance:** 3
**Recommendation:** 6
**Confidence:** 4

**Main Review:**

The strengths of the paper:
- The phenomenon of Neural Collapse is certainly an interesting and thought-provoking and analyzing it further to understand its premises is certainly worthwhile of publication. In that sense, the contributions of this work are very relevant and welcoming.
- The experiments seem to corroborate that the intuitions/derivations of the paper are on the right track. I particularly like the idea of decomposing the loss in two terms, one that is the 'optimal loss' and a secondary loss that accumulates the remaining errors.
- I find the connection with the dynamics of learning a fascinating direction, and I think this is where research should focus on. It would be very interesting if indeed the dynamics of the learning, perhaps by examining the eigenvalues of the signal-to-noise ratio matrix, lead to Neural Collapse. However, I am not sure about this, because in all honesty it was hard to parse the respective subsection.
- I would spend more time on the central path, which I find an interesting concept. What does it really mean? Are the features H supposed to be 'fixed' or at least 'fixed within infinitesimal time steps'? I believe this is the largest and most interesting contribution, and a good proof why using the MSE is relevant in this context.

The weaknesses of the paper:
- A major weakness of the work is writing and structure. While very interesting ideas are in it, and it is clear that there is a worthwhile message, it is very hard to understand in precise detail the claims, so that a 'third' reader can derive their own insights. As one example, the intro is very technical, with lots of forward references, and extends till p 4. It creates a feeling of repetition and unclarity at the same time, eg, proposition 1 is basically defined twice. Lots of different concepts, terminologies, and ideas are mixed and the text often jumps from one place to another, even referencing later parts of the text that have not been read, assuming someone does read a paper sequentially. Another example is that not notation is not always clearly explained. For instance, what is the time t in equation 5? I suppose time steps during training. For the lack of writing clarity alone, I am not sure if the paper should be accepted, it would be a pity for the work itself.
- It is not clear (at all) what are the 'assumptions' that are made for the sake of the analysis.
-- For instance, equation 5 assumes that the features H (and thus X) are converged, that is the manifold of the identity-covariance features is fixed? Or, is it assumed that in infinitesimal training steps the features H (and the manifold H) remain roughly equal? In practice, what does it mean to have the loss L_{LS} for given features H, since the features H change during the training? How is this computed in the plots of figure 2?
-- In the sentence 'As LNC1 decreases, individual activations would need to tend to their corresponding class-mean', how is this conclusion derived? Do you assume W_{LS} to be fixed? Otherwise, W_{LS} can also reduce the loss, no (in fact, that is the point of learning)?
-- Although that is a point for the original NC paper, I think NC4 is self-evident.
- Focusing on the figure 2, it shows that NC2/3 is much much smaller. Does this mean that the model basically distributes features 'uniformly' early on (thus NC2/3 goes to zero fast) and from thereon, it tries to group/cluster each class features as much as possible? What about overfitting? How well does the models generalize if keeping the training till zero loss? Doesn't this contradict standard practices, like 'early stopping'? To put otherwise, what would happen if having small training sets, say 50 or 100 examples per class.
- Isn't the zero-global mean after bias b_LS an obvious conclusion, in that after subtracting the bias, the average is zero-mean? That is the point of the bias, no?
- How do you obtain the A^{-1} in equation 8? Is this part of some definition, or Linear Algebra? More generally, is equation (8) suppose to derive a result or to state it? How do you go from W(AH) to W(H) A^-1? I think you mean to say that the W operator is linear (matrix multiplication), so the multiplier A can go out, but how do you derive the A^-1?
- What is the intution behind the signal-to-noise ratio matrix in the off-diagonal elements? Class confusion so to speak?
- Section 3.3 is very involved and I am not sure I understand how the NC1-4 are derived. It is stated what we are to conclude from it, but no guidance is provided on why this is the case. I am not asking for the proof, but for the interpretation of the results. For instance, in equation (11) we have the \omega_max in the denominator, while #1 in Corollary 1 tells that the eigevalues go to infinity. Does this imply that the SNR divided by inifinity goes to 0? Generally, I am not sure what am I to take from this section.



**Summary Of The Paper:**

This paper extends the recent work on Neural Collapse, using Mean Squared Error (MSE) instead of CE, as MSE is easier for analysis. With this, the paper shows that the least square loss can be decomposed into one that corresponds to a so called 'central' path (namely a set of optimal tuples (W, b, H) given H for which there is an optimal loss), and a perpendicular loss. The paper shows that the perpendicular loss is much smaller than the optimal least square loss and thus Neural Collapse appears due to the fact that the optimizer focuses on the central path. Empirical studies confirm the findings.

**Summary Of The Review:**

All in all, the paper had very nice ideas, but the writing and presentation is suboptimal. This means that it is not ready yet for publication.

---

> ### Author Response · Authors · 2021-11-14
> **Authors' Response to Reviewer "Jnw3" (Part 3/3)**
>
>   12. *"How do you obtain the $A^{-1}$ in equation 8? Is this part of some definition, or Linear Algebra? More generally, is equation (8) suppose to derive a result or to state it? How do you go from $W(AH)$ to $W(H) A^{-1}$? I think you mean to say that the $W$ operator is linear (matrix multiplication), so the multiplier $A$ can go out, but how do you derive the $A^{-1}$?"*
>       #
>       Thank you for this question. Indeed, Equation 8 (which is now Equation 7 in the revision) derives from linear algebra and the definition of $W_{\text{LS}}$. It is intended to demonstrate the following result: the multiplication of the MSE-optimal classifier corresponding to the features $AH$, i.e. $W_{\text{LS}}(AH)$, by $AH$ is the same as the multiplication of the MSE-optimal classifier for the features $H$, i.e. $W_{\text{LS}}(H)$, by $H$. We have expanded the equation to clarify these facts.
>       #
>
>   13.   *"What is the intution behind the signal-to-noise ratio matrix in the off-diagonal elements? Class confusion so to speak?"*
>         #
>         Thank you for this question. The off-diagonal elements of $M^T \Sigma_W^{-1} M$ i.e., the Gram of the SNR matrix, is the class-confusion matrix. The off-diagonals of the SNR matrix itself does not possess any intuitive interpretation that we are aware of.
>         #
>
>   14.   *"Section 3.3 is very involved and I am not sure I understand how the NC1-4 are derived. It is stated what we are to conclude from it, but no guidance is provided on why this is the case. I am not asking for the proof, but for the interpretation of the results."*
>         #
>         Thank you for this feedback. An intuitive summary of the reasoning is as follows:
>         #
>         Fact #1 of Corollary 1 shows that the eigenvalue of the SNR matrix $\Sigma_W^{-0.5} M$ tend to infinity. As such, the within-class covariance $\Sigma_W$ becomes negligible compared to the class means $M$. This is exactly NC1.
>         #
>         Fact #3 of Corollary 1 (which is now Fact #2 in the revision) shows that all the (non-zero) eigenvalues of the SNR matrix are becoming equal. The only matrix which has equal (non-zero) eigenvalues is a Simplex Equiangular Tight Frame (ETF). Thus, this implies NC2.
>         #
>         Given NC1-2 on the central path, NC3-4 follow directly from Theorem 1 of [Papyan, Han, and Donoho 2020].  NC4 is also self-evident from NC1-NC3, as the reviewer already noted.
>         #
>         Based on the reviewer's feedback, we now explicitly describe this intuition in the revised Section D.6 after Corollary 1 and before Corollary 2. We thank the reviewer for this clarifying suggestion.
>
>  15. *"For instance, in equation (11) we have the $\omega_{\text{max}}$ in the denominator, while #1 in Corollary 1 tells that the eigevalues go to infinity. Does this imply that the SNR divided by inifinity goes to 0?"*
>      #
>      Thank you for this feedback. The entries of the SNR matrix tend to infinity at the same rate as omega_max. Once the SNR matrix is rescaled by $1/\omega_{\text{max}}$, we obtain that the rescaled matrix tends to a finite limit given by the singular vectors of the SNR matrix at initialization. This is the claim of Equation (11).
>      #
>
> We again reiterate our gratitude to the reviewer for the above detailed recommendations, we truly feel that they led to revisions that greatly improved the quality and clarity of the paper. Given these improvements, might the reviewer consider raising their score for this paper and recommending it for acceptance?

---

> > ### Comment · Reviewer_Jnw3 · 2021-11-18
> > **Thanks**
> >
> > Hi,
> >
> > first of all, thank you for your positive attitude and your detailed respose. Also, sorry for my late reply, I had a few deadlines in the meantime as well.
> >
> > The responses look alright. Since my critique about the work was mainly presentation, I will have to have a more detailed look about the updated manuscript. So, I cannot really tell my final recommendation, hopefully by the weekend. As far as the technical questions, however, I am covered.
> >
> > Cheers,

---

> > > ### Author Response · Authors · 2021-11-18
> > > **Re: Thanks**
> > >
> > > Thank you for this response! We greatly appreciate the reviewer's kind thoughtfulness in taking time during the busy week to provide us this update.
> > >
> > > We are happy to hear that the reviewer is satisfied with the technical details.
> > >
> > > We look forward to hearing the reviewer's final recommendation. We are also happy to make additional revisions on the paper's presentation based on any additional feedback.

---

> > > > ### Comment · Reviewer_Jnw3 · 2021-12-01
> > > > **Raised score**
> > > >
> > > > Hi,
> > > >
> > > > I decided to raise the score considering the effort put by the authors. Still, I think another iteration would benefit the message in the paper.

---

> ### Author Response · Authors · 2021-11-14
> **Authors' Response to Reviewer "Jnw3" (Part 2/3)**
>
>    5. *"In practice, what does it mean to have the loss $L_{\text{LS}}$ for given features $H$, since the features $H$ change during the training?"*
> #
>     Consider some fixed (last-layer) features $H$—for example at a fixed moment in time. Suppose, one were hypothetically allowed to optimize over the classifier $W$. Then, $L_{\text{LS}}$ is the smallest possible value achievable on the loss, $L$, when only varying $W$. The $W$ minimizing the loss for the given $H$ is $W_{\text{LS}}$, which depends on $H$. And the loss achieved for that given $H$ with $W_{\text{LS}}$ is $L_{\text{LS}}(H)$. Thank you for pointing out this potential confusion. In response to the reviewer’s feedback, we now explicitly clarify this intuition in the discussion of the Central Path after Equation (4).
> #
>  6. *"How is this computed in the plots of figure 2?"*
> #
>     $W_{\text{LS}}$ has a closed mathematical form, due to Webb and Lowe (1990), as a function of $H$ (Proposition 1). Therefore, in Figure 2, we can explicitly compute $W_{\text{LS}}$—and hence $L_{\text{LS}}$—using this closed form solution.
> #
>  7. *"In the sentence 'As LNC1 decreases, individual activations would need to tend to their corresponding class-mean', how is this conclusion derived? Do you assume $W_{\text{LS}}$ to be fixed? Otherwise, $W_{\text{LS}}$ can also reduce the loss, no (in fact, that is the point of learning)?"*
> #
>     We thank the reviewer for pointing out this subtlety. We have rewritten the relevant part of the text (the discussion below Theorem 2) to clarify this intuition. In particular, the intuition is as follows: First, observe that $L_{\text{NC2/3}}$ is a function of the class-means and MSE-optimal classifiers. Minimizing $L_{\text{NC2/3}}$ will push the (unextended) class-means and classifiers towards the same Simpex ETF matrix. Next, note that the within-class variation is independent of the means. Thus, despite the fact that classifiers are converging towards some (potentially large) ETF matrix, we can always reduce LNC1 by pushing $\Sigma_W$ towards 0 i.e. (NC1). Thus, to simultaneously minimize both components, (NC1)-(NC3) must all occur.
> #
>  8. *"Although that is a point for the original NC paper, I think NC4 is self-evident."*
> #
>     Yes, we agree.
> #
>  9. *"Focusing on the figure 2, it shows that NC2/3 is much much smaller. Does this mean that the model basically distributes features 'uniformly' early on (thus NC2/3 goes to zero fast) and from thereon, it tries to group/cluster each class features as much as possible?"*
> #
>     Yes, this intuition is indeed correct. To make this insight explicit to readers, we have added it to the caption of Figure 2. We thank the reviewer for this feedback.
> #
>  10. *"What about overfitting? How well does the models generalize if keeping the training till zero loss? Doesn't this contradict standard practices, like 'early stopping'? To put otherwise, what would happen if having small training sets, say 50 or 100 examples per class."*
>      #
>      The original NC paper (Papyan, Han, Donoho, 2020) showed empirically that training past zero error towards zero loss improves generalization. In our supplementary Table 1 in Appendix A, we also show analogous empirical findings for networks trained with MSE loss. This aligns with other related research works by [1] and [2] who have made similar observations. We have not investigated the effect of training sample size on this phenomenon and we agree it would be a very interesting direction for future research. We suspect that early stopping may indeed be useful in cases where sample size is small or the data is noisy, but, since we have not personally investigated such cases, we can not make any conclusive claims. We thank the reviewer for this thought provoking question and have added a new Footnote 2 into the main text to discuss early stopping.
>
>      #
>      [1] The Implicit Bias of Gradient Descent on Separable Data
>      [2] Reconciling modern machine-learning practice and the classical bias–variance trade-off
>      #
>   11. *"Isn't the zero-global mean after bias $b_{\text{LS}}$ an obvious conclusion, in that after subtracting the bias, the average is zero-mean? That is the point of the bias, no?"*
>       #
>       Yes, we completely agree with the reviewer. We only chose to give a detailed description of this fact in the text in order to clarify this to readers from a diverse variety of backgrounds.
>       #

---

> ### Author Response · Authors · 2021-11-14
> **Authors' Response to Reviewer "Jnw3" (Part 1/3)**
>
> We thank the reviewer for their careful and detailed reading of our paper. Based on the reviewer’s questions and recommendations, we have made many revisions that significantly improved the clarity of the paper. The new, revised paper has been uploaded to OpenReview.
>
> #
> Below, we give point-by-point responses to the reviewer’s comments and describe the revisions we made to address them. We would be happy to add additional clarifications and revisions to the paper to address any additional recommendations from the reviewer.
>
> #
>
>  1. *"I would spend more time on the central path, which I find an interesting concept. What does it really mean? Are the features H supposed to be 'fixed' or at least 'fixed within infinitesimal time steps'?"*
> #
>     We are very happy that the reviewer shares our interest in the central path abstraction. The reviewer’s intuition is correct. For a classifier-features pair to lie on the central path, the classifier in the pair has to exactly equal the MSE-optimal classifier corresponding to the features i.e. the optimal classifier that would result from fixing $H$ (within that infinitesimal timestep) and computing an MSE-optimal classifier for that $H$. We now explicitly present this interpretation after the introduction of the Central Path in Equation 4. We thank the reviewer for this clarifying feedback.
> #
>  2.   *"A major weakness of the work is writing and structure. While very interesting ideas are in it, and it is clear that there is a worthwhile message, it is very hard to understand in precise detail the claims, so that a 'third' reader can derive their own insights. As one example, the intro is very technical, with lots of forward references, and extends till p4. It creates a feeling of repetition and unclarity at the same time, eg, proposition 1 is basically defined twice. Lots of different concepts, terminologies, and ideas are mixed and the text often jumps from one place to another, even referencing later parts of the text that have not been read, assuming someone does read a paper sequentially."*
> #
>     We thank the reviewer for this feedback. Following this suggestion, we have abbreviated the introduction (to only approximately two pages) to clarify and reduce the repetition in the paper. In particular, in Section 1.3, we removed all technical descriptions, forward references, and discussions that overlap with later claims (besides a bullet-list summary).
> #
>  3.   *"Another example is that not notation is not always clearly explained. For instance, what is the time t in equation 5? I suppose time steps during training."*
> #
>     Thank you for this feedback. The variable $t$ represents a continuous moment in time during the training of the features and classifiers. Time is continuous since our theory examines gradient flow in continuous time rather than gradient descent in discrete time steps. In response to this feedback, we now explicitly define t immediately before its introduction in Equation (5).
> #
>  4. *"For instance, equation 5 assumes that the features $H$ (and thus $X$) are converged, that is the manifold of the identity-covariance features is fixed? Or, is it assumed that in infinitesimal training steps the features $H$ (and the manifold $H$) remain roughly equal?"*
> #
>     The word "manifold" possesses many meanings in the literature that could lead to confusion. Some papers consider a "manifold of features" (consisting of transformations of all potential images after passing through multiple deep net layers) which is evolving as the network parameters are optimized. We are *not* using the word manifold in this sense and are *not* claiming that a "manifold of features" has converged to an identity-covariance feature manifold. Instead, the manifold in our theory is the set of all matrices having identity within-class covariance. In other words, our assumption is that the features are renormalized to have an identity within-class covariance throughout the optimization of the deep network—and therefore always stay within this set (sometimes called a Generalized Stiefel Manifold). We thank the reviewer for pointing out this ambiguity and we now explicitly clarify this distinction in Footnote 10 of the text.

---

### Decision · Program_Chairs · 2022-01-20

**Decision:**

Accept (Oral)

**Comment:**

This paper extends the Neural Collapse (NC) phenomenon discovered by Papyan, Han and Donoho (2020) on deep learning image classifications with Cross Entropy (CE) loss, to the scenario with Mean Squared Error (MSE), that achieves similar performance to CE and favors deeper analysis. In particular, the paper shows that the least square loss can be decomposed orthogonally into a 'central' path as the optimal least square loss, and its perpendicular loss. Moreover, the paper shows by experiments that after the zero training error (Terminal Phase of Training, or TPT) the perpendicular loss is typically much smaller than the optimal least square loss, and the optimal least square loss is further decomposed into the NC1 loss which is the dominance and NC2/3 loss (even smaller than the perpendicular loss). Such a discovery with loss decomposition is very thought provoking to understand the training dynamics of deep neural networks.

Reviewers unanimously accept the paper, so is the final recommendation.